

# Fermionic tensor network methods

Quinten Mortier[1*], Lukas Devos[1], Lander Burgelman[1], Bram Vanhecke[2],
Nick Bultinck[1], Frank Verstraete[1,3], Jutho Haegeman[1] and Laurens Vanderstraeten[4]

**1** Department of Physics and Astronomy, Ghent University, Belgium
**2** University of Vienna, Faculty of Physics, Boltzmanngasse 5, 1090 Wien, Austria
**3** Department of Applied Mathematics and Theoretical Physics,
University of Cambridge, United Kingdom
**4** Center for Nonlinear Phenomena and Complex Systems,
Université Libre de Bruxelles, Belgium

* quinten.mortier@ugent.be

## Abstract

We show how fermionic statistics can be naturally incorporated in tensor networks on arbitrary graphs through the use of graded Hilbert spaces. This formalism allows the use of tensor network methods for fermionic lattice systems in a local way, avoiding the need of a Jordan-Wigner transformation or the explicit tracking of leg crossings by swap gates in 2D tensor networks. The graded Hilbert spaces can be readily integrated with other internal and lattice symmetries, and only require minor extensions to an existing tensor network software package. We review and benchmark the fermionic versions of common algorithms for matrix product states and projected entangled-pair states.

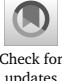

## Contents



# 1 Introduction

Crucial in the numerical progress was the development of a toolbox with tensor network methods for tackling quantum many-body systems. The density matrix renormalization group (DMRG) [10] was first introduced to efficiently simulate ground state properties of spin chains, a method that was later reformulated as a sweeping-like variational optimization of matrix product states (MPS). Building on the MPS formalism, ground state algorithms were later formulated for infinite systems directly. Examples are the time-evolving block decimation (TEBD) algorithm [11], the infinite DMRG (iDMRG) algorithm [12] or the variational optimization of uniform MPS (VUMPS) [13]. Going beyond ground state properties, MPS algorithms are used for approximately simulating real-time evolution, excited states, finite temperature states or open systems. The numerical efficiency of these MPS methods is staggering, to the extent that they are now also routinely used for simulating two-dimensional systems on large cylindrical geometries, despite the exponential scaling of the bond dimension as a function of the cylinder circumference.

The class of projected entangled-pair states (PEPS) [14, 15] was introduced for simulating quantum lattice systems in two dimensions directly, and the computational toolbox is in full development. The task of evaluating an expectation value for a given PEPS cannot be done exactly, and different approximation methods have been developed based on the corner transfer matrix renormalization group (CTMRG) [16–18] or boundary MPS [15, 19, 20]. Given these contraction routines, the variational optimization of PEPS can be performed using imaginary time evolution [15, 21, 22] or direct variational optimization [23, 24]. The extension of PEPS methods to capture excited states, time evolution or mixed states in two dimensions is actively pursued.

However, many (if not all) of these methods were originally formulated in the setting of spin systems. Here, we intend their application to fermions and therefore need either a reformulation of those fermions in terms of spins, or an adaption of the methods to accommodate the anti-commuting nature of the fermionic degrees of freedom. In one dimension, the former can be achieved via the Jordan-Wigner transformation [25]. However, in higher dimensions, an equally simple and effective generalization of this transformation does not exist, though there is recent progress [26] and specific implementations have been achieved for PEPS on finite two-dimensional lattices [27]. Hence, fermions need to be explicitly integrated into the formalism. Possible routes to do so have already been proposed, e.g. in Ref. [28] where the fermionic degrees of freedom are replaced by Grassmann numbers. For the resulting Grassmann tensor networks, interesting results have been obtained both numerically [29] and conceptually [30, 31]. An alternative to the Grassmann formalism, originally introduced in Refs. [32] and [33], writes the local tensors as a normally-ordered product of physical and virtual creation and annihilation operators acting on the vacuum [34–39]. To maintain the celebrated diagrammatic language of tensor networks, all these tensors are required to have even parity. The composite spaces on the other hand can have odd sectors such that crossings of the corresponding legs in a tensor diagram will yield additional minus signs. To keep track of these minus signs, one "resolves" crossings by replacing them with so-called swap gates [40]. With this technique many of the standard tensor network methods can be rephrased for fermions. However, to keep track of the minus signs, one has to be careful in the manipulation of tensor diagrams. Indeed, every additional crossing has to be resolved with a new swap gate. Despite this method requiring some bookkeeping in practice, it has been used quite successfully for simulating $t$-$J$ models [41–43] and Hubbard models [27, 39, 44–46] in two dimensions, including excited states [47], finite temperatures [48] and time-evolution [49].

In this work, we present an alternative approach based on earlier works by some of the authors [9, 50], where fermions are integrated in the tensor network formalism by constructing the latter from $\mathbb{Z}_2$-graded linear spaces, also called super vector spaces. In these earlier works, the intention was to classify fermionic topological phases in 1D and 2D and the discussion was primarily analytical. Here, we extend the formalism toward numerical implementation. We thus build fermionic MPS and PEPS and consider their interplay with the aforementioned methods. Our framework even supports the Jordan-Wigner transformation itself, opening up the route to its higher-dimensional extensions. Benchmark simulations have been performed in both one dimension (with a focus on excitations) and two dimensions (aimed at the reproduction of Gaussian correlation functions) with open source software libraries developed by some of the authors, both in MATLAB, by means of the `TensorTrack` package [51], as well as using the Julia programming language, through the combination of `TensorKit.jl` and `MPSKit.jl` [52, 53]. These provide efficient implementations of many (symmetric) tensor network algorithms, to which fermionic features have now been added.

## 2 Fermionic tensors

Before delving into specific fermionic tensor network states (fTNS) and their corresponding algorithms, we first introduce the concept of a $\mathbb{Z}_2$-graded tensor. We start from a list of desiderata for these tensors and demonstrate how the $\mathbb{Z}_2$-graded formalism elegantly implements them. We also provide a justification for the diagrammatic tensor network notation and conclude by discussing tensor properties such as hermiticity and unitarity. These properties serve as the foundation for various tensor manipulations and decompositions that we introduce in the subsequent sections.

### 2.1 Preliminaries

The wavefunction of an arbitrary fermionic many-body state is given by

$$|\Psi\rangle = \sum_{\sigma_1,\ldots,\sigma_N} \psi_{\sigma_1\cdots\sigma_N} \left(a_1^\dagger\right)^{\sigma_1} \cdots \left(a_N^\dagger\right)^{\sigma_N} |\Omega\rangle, \tag{1}$$

where $|\Omega\rangle$ is the Fock vacuum and $\{a_i^\dagger\}_{i=1,..,N}$ collects $N$ fermionic creation operators satisfying canonical anti-commutation relations. Note that the order chosen for the indices in this expression is important. Swapping $a_1^\dagger$ and $a_2^\dagger$, for instance, would introduce an extra parity factor $(-1)^{\sigma_1\sigma_2}$, thus altering the coefficients. Indeed, labeling a fermionic many-body state by means of coefficients like $\psi_{\sigma_1\cdots\sigma_N}$ will always entail choosing an ordering for the modes and, opposed to bosons, swapping two could also introduce a minus sign now. Furthermore, note that our physical state $|\Psi\rangle$ resides in the tensor product of single-particle local Hilbert spaces and therefore only consists of ket states/vectors. One could equally well consider the corresponding bra states/covectors in the dual spaces. Building tensor networks, one needs both and, as we will show in this section, the distinction between both will prove to be instrumental to construct a consistent tensor network methodology for fermions.

The idea behind a tensor network state is to provide an efficient parametrization of the coefficients $\psi_{\sigma_1\cdots\sigma_N}$ as their number grows exponentially with $N$. Conceptually, a clear way to arrive at this point starts from virtual modes. Therefore, consider an exemplary system with four physical fermionic modes, which we can represent schematically as

$$\psi_{\sigma_1\sigma_2\sigma_3\sigma_4} = \quad\psi\quad \tag{2}$$

To every physical degree of freedom we associate two virtual fermionic modes and arrange these in maximally entangled pairs to create

$$|\Psi_{\text{virt}}\rangle = \prod_e \frac{1}{\sqrt{2}}\left(1 + a_{e,1}^\dagger a_{e,2}^\dagger\right)|\Omega_{\text{virt}}\rangle = \tag{3}$$

where $e$ labels the edges along which the entangled pairs are oriented with the additional index discerning between between both sides of these edges (for instance, running through the loop in a counterclockwise manner). Next, we introduce a linear operator that maps the virtual state onto a physical many-body state. In a tensor network, this map is a product of local maps,

$$\mathcal{A}_j = \sum_{\sigma,\alpha_1,\ldots,\alpha_n} (A_j)_{\sigma\alpha_1\cdots\alpha_n} (a_j^\dagger)^\sigma (a_1)^{\alpha_1}\cdots(a_n)^{\alpha_n}, \tag{4}$$

where the creator $a_j^\dagger$ acts on physical mode $j$ while the annihilators act on the $n$ (in this case two) virtual degrees of freedom per site. The tensor network state is obtained by projecting back on the virtual vacuum state,

$$|\Psi\rangle = \langle\Omega_{\text{virt}}| \left( \prod_j \mathcal{A}_j \right) |\Psi_{\text{virt}}\rangle \, |\Omega_{\text{phys}}\rangle \, . \tag{5}$$

Schematically, $|\Psi\rangle$ can be represented as

$$|\Psi\rangle = \tag{6}$$

In the case where both the physical and virtual degrees of freedom are bosonic, this procedure gives rise to a tensor network representation for $|\Psi\rangle$ where the coefficients $\psi_{\sigma_1\cdots\sigma_N}$ can be computed by promoting the local $A_j$ to tensors and contracting them. However, in the case of fermionic modes, anti-commutation relations could generate a number of minus signs such that the state can no longer be obtained as a simple contraction of the local $A_j$ tensors. Indeed, in typical contraction routines, two tensors are contracted by first reordering their indices so that the contracted indices are neighboring. Next, they are multiplied (for instance by first reshaping the tensors into matrices and performing a matrix multiplication) after which the result is reshaped back and reordered as desired. For fermions these internal reorderings already generate minus signs and thus form a difference compared to bosons. Even more important is the fact that tensor network diagrams can display lines crossing each other, corresponding to the swapping of fermionic modes and thus the adding of parity factors upon calculating the contraction. As mentioned before, one possible way to keep track of these swapping operations is by replacing every crossing by a swap gate. Here, we would like an alternative that keeps track of the fermionic minus signs by incorporating them in a more local manner, within the tensor, in line with the general tensor network philosophy. I.e., we require the following properties:

- In the definition of the linear map $\mathcal{A}_j$ of a fermionic tensor, the coefficients of $A_j$ depend on the chosen linear order of the modes, and different orderings can be related by parity factors.

- We want the familiar diagrammatic notation of tensor networks to survive. One should thus be able to represent all relevant tensor network operations as diagrams where the composite blocks can be moved around freely. Note that this immediately requires the tensors to have an even total parity, i.e., with parity-preserving $\mathcal{A}_j$, such that we can swap their positions.

- We require the existence of a contraction procedure for the numerical computation of tensor contractions, i.e., a single-line command which is well defined and the result of which is independent of the order of its arguments.

More generally, these items indicate that the evaluation of a tensor network diagram can only depend on the topological properties of this diagram, and that these topological properties need to be completely determined by the linear order of the modes of the input tensors, the pairs of contracted modes, and the linear order of the output tensor. Note that this is not the

case for the swap gate approach where for instance

$$\text{(diagram)} \qquad \text{and} \qquad \text{(diagram)} \tag{7}$$

have identical leg orders (when ordering everything from left to right) and connections but where the self-crossing in the latter has to be resolved by a swap gate,

$$\overset{i_1 \quad i_2}{\underset{j_1 \quad j_2}{\times}} = (-1)^{|i_1||i_2|}\delta_{i_1 j_2}\delta_{i_2 j_1}, \tag{8}$$

with $|i|$ indicating the fermion parity of its corresponding leg, i.e., 0 for even and 1 for odd. The swap gate thus identifies legs but adds a minus sign when these legs all carry an odd fermion parity. As such, a self-crossing with $j_1 = j_2$ reduces to a parity factor $(-1)^{|i_1|}\delta_{i_1 i_2}$ and therefore conflicting results in Eq. (7).

A priori it is not clear whether a consistent fermionic formalism, satisfying the three constraints, even exists. Indeed, for more general anyonic degrees of freedom with non-symmetric braiding rules [54], there is simply no alternative to the swap-gate approach where every line crossing in the network diagram has to be resolved by hand, and a distinction between over- and undercrossing has to be made. The symmetric braiding rules for fermions, on the other hand, allow to satisfy these three desiderata, albeit in a non-trivial way. To illustrate this, consider the following type of diagram that we will also encounter in Sec. 3,

$$\text{(diagram)} \tag{9}$$

Here the numbers denote the order of the fermionic modes in the definition of the tensors, which is the same for $\rho$ and $\rho'$. If a contraction method as described above were to exist, it would be completely specified by the pairs of contracted modes, and the linear order of the output tensor.[1] I.e.,

- the third leg of $B$ has to be contracted with the first leg of $\rho$,

- the first leg of $A$ has to be contracted with the second leg of $\rho$,

- the second leg of $A$ has to be contracted with the second leg of $B$,

- the first leg of $B$ corresponds to the first leg of $\rho'$,

- the third leg of $A$ corresponds to the second leg of $\rho'$.

We could uniquely translate these statements to a linearized diagram by drawing the contracted tensors from left to right with all of their legs pointing downward and in the order

---

[1] Using the well-known NCON syntax [55], $\rho'$ would be calculated via

```
ρ' = contract(B, [-1,3,1], ρ, [1,2], A, [2,3,-2]).
```

Here, the repeated positive numbers indicate pairs of legs to be contracted and numbers themselves indicate the order in which to do so. The negative numbers on the other hand denote legs that should not be contracted with the numbers again referring to the order of the legs in the resulting tensor object.

of the tensor's definition. Next, contracted legs are connected horizontally and open legs are ordered from left to right in the order of the output tensor. In this way, we obtain

$$\lceil \rho' \rceil = \lceil B \quad \rho \quad A \rceil \tag{10}$$

By continuously deforming the diagram, we recover the intended. However, the order of the tensors in the contraction command should not matter. Therefore, swapping $A$ and $\rho$ and drawing its corresponding tensor diagram like before,[2] we obtain

$$\lceil \rho' \rceil = \lceil B \quad A \quad \rho \rceil \tag{11}$$

If we were to apply the swap-gate approach, the three crossings should be resolved,

$$\lceil \rho' \rceil = \lceil B \quad A \quad \rho \rceil \tag{12}$$

Subsequently deforming this by dragging the full (and even) $\rho$ tensor through the $A$ legs, we get

$$\lceil \rho' \rceil = \lceil B \quad \rho \quad A \rceil \tag{13}$$

Note that the self-crossing, corresponding to a fermionic parity factor, forms a crucial difference between Eq. (13) and Eq. (10) and that we are again left with two conflicting results. We conclude that, using the swap gate method, the linear order of the modes of the input tensors, the pairs of contracted modes, and the linear order of the output tensor, do not paint a complete picture. Indeed, one should make clear if legs are self-crossing or not. One possible way to do so is to uniquely select the diagram which is crossing-free, thus avoiding fermionic minus signs upon contraction. If we do want a crossing in the diagram, then it should be added explicitly via a swap gate tensor but upon invoking the contraction routine, every diagram has to be planar. This may seem drastic, but for MPS methods the typical connection graphs are planar to begin with, so that this procedure automatically selects the desired result. For our example, this would force us to use the first contraction while the second diagram would be precluded. Hence, we are immediately granted a self-consistency check when calling the contraction command.[3] While elegant for MPS, matters are not as simple for PEPS since typical connection graphs are no longer planar. One should thus write down the desired tensor diagram with inevitable crossings, introduce braiding tensors that resolve these (i.e., swap gates) and then use the planar contraction code to compute the result. Of course this invalidates our initial objective to encode the fermionic characteristics locally within the tensors.

Fortunately, there is a second option. Rather than forbidding (self-)crossings, we will keep track of them in a way that does not require any drastic changes to standard contraction routines, namely by introducing an orientation (arrow) to all tensor legs. This idea is not

---

[2]In the NCON syntax, this comes down to

$$\rho' = \texttt{contract(B, [-1,3,1], A, [2,3,-2], } \rho \texttt{, [1,2]).}$$

[3]`TensorKit.jl` supports resolving planar diagrams from an NCON-like input, provided such a planar configuration exists, using the `@planar` syntax. This convention is used to implement fermions and more general anyonic symmetries in MPS algorithms in the `MPSKit.jl` package [53].

new and has already been used in bosonic, globally symmetric tensor network codes [56]. For fermions, arrows can tell us exactly where self-crossings, and the corresponding fermionic minus signs, appear. Indeed, consider again Eq. (9) but where we now choose an orientation for all legs, so that the arrows on the horizontal (contracted) legs in Eq. (10) all point in the same direction, here to the left. I.e., we choose

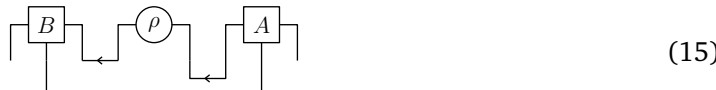

$$ \tag{14} $$

corresponding to

$$ \tag{15} $$

Using the same arrows, the second diagram becomes

$$ \tag{16} $$

Comparing the two cases, one finds that every arrow that changes direction will result in a self-crossing. By adding fermionic minus signs for all "opposite" arrows, it can then correct for the self crossings by adding the appropriate minus signs, so that both results agree. As many codes already have tensor objects with oriented legs, the fermionic contraction routine just has to compare the arrows to the order of the tensors in the function call. If it points to the left no extra minus signs are added, if it points to the right, we add a parity factor. While this recipe results in a method where Eq. (15) and Eq. (16) agree and thus satisfies all three requirements we put forward, it is not a priori clear that it corresponds to a natural and consistent implementation of fermionic multilinear algebra. Consider for instance the diagrammatic representation of a matrix,

$$ \tag{17} $$

and suppose we want to contract both legs. Precluding crossings, the only way to do so is

$$ \tag{18} $$

yielding $A_{ii} = \mathrm{tr}(A)$, the standard trace without any fermionic minus signs. However, with oriented legs, two possibilities exist. Either

$$ \tag{19} $$

yielding the standard trace again as the contracted arrow points to the left or

$$ \tag{20} $$

where the contracted arrow points to the right yielding a so-called *supertrace*, $A_{ii}(-1)^{|i|}$. As we will see in the next Section, this supertrace reappears when working with $\mathbb{Z}_2$-graded Hilbert spaces and we can interpret the direction of the arrows as a difference between bras and kets.

We conclude that by using arrows and incorporating them into the contraction routine as described, there is no longer a need to monitor fermionic leg crossings in tensor network diagrams. This aspect of bookkeeping is entirely delegated to the arrows, which are often already in place to incorporate symmetries, and represents the main advantage of our formalism over the swap gate approach.

## 2.2 Graded Hilbert spaces

In this section we introduce $\mathbb{Z}_2$-graded Hilbert spaces and their tensor products. We show that these form a natural mathematical framework incorporating the three requirements from the previous section by giving the legs an orientation, leading to a supertrace.

$\mathbb{Z}_2$-graded Hilbert spaces are (in our case finite) Hilbert spaces with the defining property that they decompose in two orthogonal sectors,

$$V = V^0 \oplus^\perp V^1, \tag{21}$$

referred to as even and odd, respectively. If a given state $|v\rangle$ only has support in one of the two sectors, it has a definite grading or parity which we denote by $|v|$. These vectors are called homogeneous, and inhomogeneous vectors can always be decomposed into a sum of homogeneous ones. One can always introduce an orthonormal basis $\{|i\rangle, i = 1, \ldots, \dim(V)\}$, with thus $\langle i|j\rangle = \delta_{ij}$, consisting solely of homogeneous vectors with parity $|i|$. When sorted, we have $|i| = 0$ for $i = 1, \ldots, \dim(V^0)$ and $|i| = 1$ for $i = \dim(V^0)+1, \ldots, \dim(V^0)+\dim(V^1) = \dim(V)$.

Now, consider homomorphisms between two $\mathbb{Z}_2$-graded Hilbert spaces $V$ and $W$. A homomorphism is defined to be even (odd) when it is parity-preserving (-changing) meaning that it maps $V^0$ into $W^0$ ($W^1$) and $V^1$ into $W^1$ ($W^0$). Noting that $\mathrm{Hom}(V,W)$ is a Hilbert space itself and that any homomorphism can be uniquely decomposed in a parity-preserving and a parity-changing part, we thus find that the homomorphisms constitute a graded Hilbert space themselves.[4] In particular, one can consider the dual space $V^* = \mathrm{Hom}(V,\mathbb{C})$ containing the linear forms on $V$ and use the inner product structure of $V$ to construct the canonical anti-linear isomorphism $V \to V^* : |v\rangle \mapsto \langle v|$ where $\langle v|(|w\rangle) = \langle v|w\rangle, \forall |v\rangle, |w\rangle \in V$. As the complex field is only trivially graded in the sense that it does not contain an odd sector ($\mathbb{C} = \mathbb{C} \oplus \{0\}$) and as the inner product between vectors with a different grading yields zero, one immediately obtains that $\langle v|$ has the same grading as $|v\rangle$ when it is homogeneous. The dual space $V^*$ thus inherits an identical grading from $V$.

In order to describe many-body states, we need to consider tensor products of such graded Hilbert spaces, e.g. $V_1 \otimes V_2$. Note that this space is graded as well when the parity of the natural basis states $|i\rangle_1 \otimes |j\rangle_2$ is defined as $|i| + |j| \mod 2$ (hence a $\mathbb{Z}_2$ grading) with $\{|i\rangle_1\}$ and $\{|j\rangle_2\}$ homogeneous basis sets in $V_1$ and $V_2$ respectively.[5] Furthermore, this space is still a Hilbert space in the sense that it has an inner product inherited from $V_1$ and $V_2$,

$$\langle |v_1\rangle \otimes |v_2\rangle, |w_1\rangle \otimes |w_2\rangle \rangle = \langle v_1|w_1\rangle \langle v_2|w_2\rangle, \tag{22}$$

and extended via sesquilinearity to the whole space. The same applies for $V_2 \otimes V_1$ and both spaces can be connected by the isomorphism

$$\mathcal{F} : V_1 \otimes V_2 \to V_2 \otimes V_1 : |i\rangle_1 \otimes |j\rangle_2 \mapsto (-1)^{|i||j|} |j\rangle_2 \otimes |i\rangle_1, \tag{23}$$

also called fermionic reordering.[6] Indeed, this swapping prescription encodes the fermionic anti-commutation relations. We will use the notation $\mathcal{F}$ very loosely and will refer to objects related by this isomorphism as being the same. This is possible since the precise order of subsequent applications of $\mathcal{F}$ is irrelevant and comparison of objects will always implicitly assume that their spaces are first brought into the same order.

---

[4]Because $V$ and $W$ are assumed finite-dimensional, the Hilbert-Schmidt inner product $\mathrm{tr}(A^\dagger B)$ is well defined for all homomorphisms $A$ and $B$, and yields zero if one of the two is parity-preserving and the other parity-changing.

[5]In this way, the grading of $V \otimes W^*$ is also compatible with that of $\mathrm{Hom}(V,W)$.

[6]In the mathematics literature on graded algebras, this is known as the Koszul sign rule. More generally, in the context of tensor categories, it is an instance of a braiding rule. In particular, it is a symmetric braiding rule because applying it twice yields the identity.

Finally, we also introduce a canonical contraction map

$$\mathcal{C}: V^* \otimes V \to \mathbb{C} : \langle i | \otimes | j \rangle \mapsto \mathcal{C}(\langle i | \otimes | j \rangle) = \langle i | j \rangle = \delta_{ij}. \tag{24}$$

Together with the reordering isomorphism, this can be extended to $V \otimes V^*$,

$$\mathcal{C}(|i\rangle \otimes \langle j|) = (-1)^{|i||j|} \mathcal{C}(\langle j | \otimes |i\rangle) = (-1)^{|i||j|} \delta_{ij}, \tag{25}$$

thus implementing the supertrace through the additional parity factor.[7] Note that this is the rigourous instantiation of the aforementioned orientation of the tensor legs, as we can associate a contracted leg with a left-pointing arrow with $\mathcal{C}$, and a right-pointing arrow with $\mathcal{C} \circ \mathcal{F}$, after 'linearizing' the diagram as in Eq. (15) and (16). We come back on this in Sec. 2.4.

## 2.3 Graded tensors and tensor operations

We now introduce fermionic tensors as objects living in tensor products of $\mathbb{Z}_2$-graded Hilbert spaces and their duals. Consider for instance, a fermionic tensor $A \in V_1 \otimes V_2 \otimes V_3^* \otimes V_4^*$, defined as

$$A = A_{\alpha\beta\gamma\delta} |\alpha\rangle |\beta\rangle \langle\gamma| \langle\delta|, \tag{26}$$

where, from this point on, we omit the $\otimes$ symbol for taking tensor products. We furthermore adopt the Einstein summation convention, where each index has a range associated with the basis and dimension of its associated vector space. The order of the indices in the above tensor is important. We can change this order using the swapping prescription $\mathcal{F}$, which leads to parity factors in the tensor elements, e.g.

$$\mathcal{F}_{12}(A) = \left((-1)^{|\alpha||\beta|} A_{\alpha\beta\gamma\delta}\right) |\beta\rangle |\alpha\rangle \langle\gamma| \langle\delta|. \tag{27}$$

As mentioned previously, we will typically omit the explicit notation of $\mathcal{F}$ and apply it as needed in order to bring tensor indices into a requested order or to evaluate contractions (as discussed below).

Henceforth, we restrict to tensors for which the global parity is even, i.e., if the total parity of the indices is odd, the corresponding element should be zero:

$$A_{\alpha\beta\gamma\delta} = 0, \quad \text{when} \quad |\alpha| + |\beta| + |\gamma| + |\delta| = 1 \mod 2. \tag{28}$$

This implies that we can swap any two tensors, a feature that will be crucial when introducing the diagrammatic notation later on where the order of the tensors remains unspecified. More explicitly, this implies that for any two fermionic tensors $A$ and $B$, $A \otimes B$ and $B \otimes A$ are identical when bringing the total set of indices in a compatible order.

The most important operation between two tensors is the contraction, which we can build from the elementary contraction map $\mathcal{C}$ in Eqs. (24) and (25) together with the necessary swaps $\mathcal{F}$ to bring contracted indices next to each other (which we will not denote explicitly as mentioned above). As a first example, suppose we are given two tensors,

$$B = B_{\alpha\beta\gamma} |\alpha\rangle |\beta\rangle \langle\gamma|, \qquad \text{and} \qquad C = C_{\delta\varepsilon} \langle\delta| \langle\varepsilon|, \tag{29}$$

---

[7]In the mathematics literature on tensor categories, the contraction map $\mathcal{C}$ is known as the evaluation map associated with left duality pairing of $V^*$ and $V$. The contraction map $\mathcal{C} \circ \mathcal{F}$ is then the corresponding right evaluation map. A right evaluation map can be constructed from the left evaluation map using unitarity, namely by taking the dagger of the corresponding coevaluation map. This would yield the standard trace and therefore preserve positivity. Alternatively, a consistent right evaluation map can be constructed from the left evaluation by combining it with the braiding (reordering) and a twist (self-crossing). Because of the rules outlined in the previous section, we do not want to consider twists in our formalism. For a symmetric braiding rule, it is indeed possible to make the twist trivial. However, this choice then results in a right evaluation map that is incompatible with unitarity, and indeed gives rise to a non-positive supertrace.

then we can compute the contraction of the second index of $B$ with the first index of $C$ as

$$
\begin{aligned}
\mathcal{C}_{\beta\delta}\big(B \otimes C\big) &= \mathcal{C}_{\beta\delta}\big(B_{\alpha\beta\gamma}C_{\delta\varepsilon}\,|\alpha\rangle\,|\beta\rangle\,\langle\gamma|\,\langle\delta|\,\langle\varepsilon|\,\big) \\
&= \mathcal{C}_{\beta\delta}\big((-1)^{|\beta||\gamma|}B_{\alpha\beta\gamma}C_{\delta\varepsilon}\,|\alpha\rangle\,\langle\gamma|\,|\beta\rangle\,\langle\delta|\,\langle\varepsilon|\,\big) \\
&= \left(\sum_{\beta}(-1)^{|\beta||\gamma|}B_{\alpha\beta\gamma}(-1)^{|\beta|}C_{\beta\varepsilon}\right)|\alpha\rangle\,\langle\gamma|\,\langle\varepsilon|\,,
\end{aligned}
\tag{30}
$$

where we first reorder $B$ internally, yielding a first parity factor, $(-1)^{|\beta||\gamma|}$, followed by a ket-bra contraction, $\mathcal{C}_{\beta\delta}(|\beta\rangle\,\langle\delta|) = (-1)^{|\beta|}\delta_{\beta\delta}$, i.e., a supertrace and thus a total of two parity factors.

Let us reflect on how we could numerically implement this contraction without diving too deep into the algorithmic details. As will be explained in Sec. 2.8, we typically store our tensors via their entries, $B_{\alpha\beta\gamma}$ and $C_{\delta\varepsilon}$, together with the characteristics of the corresponding spaces, e.g. dual or non-dual, dimension, fermionic or not, etc. The spaces are fermionic here so that reordering amounts to parity factors. We could hence perform the first step in the contraction explicitly by making the tensor $\tilde{B}_{\alpha\gamma\beta}\,|\alpha\rangle\,\langle\gamma|\,\langle\beta| = (-1)^{|\beta||\gamma|}B_{\alpha\beta\gamma}\,|\alpha\rangle\,\langle\gamma|\,\langle\beta|$. Now, the contracted indices are neighbouring so that we can contract by reshaping the entries $\tilde{B}_{\alpha\gamma\beta}$ into a matrix $\tilde{B}_{\nu\beta}$, through grouping $\alpha$ and $\gamma$ into a single index $\nu$, after which the entries for the resulting tensor can be obtained via simple matrix multiplication of $\tilde{B}_{\nu\beta}$ and $C_{\delta\varepsilon}$ and reshaping $\nu$ back to $\alpha$ and $\gamma$. Note that we also need the additional parity factor from the supertrace. Indeed, inspecting the contracted spaces we would find the non-dual $\beta$ space placed first and the dual $\delta$ space placed second and we thus add a parity factor for this ket-bra contraction. Fermionic tensor network libraries like TensorTrack and TensorKit operate exactly in this way: fermionic reording is handled within the tensor and the detection of supertraces happens via the dual/non-dual character of the contracted spaces. As such, appropriate parity factors are correctly supplemented to standard bosonic contraction routines.

As tensors are part of a Hilbert space, they also have an inner product. The inner product between two tensors $A$ and $B$ with the same structure (i.e., from the same tensor product space) simply amounts to the standard complex inner product of their components, due to the choice of orthonormal basis in each of the individual Hilbert spaces. For example, for two tensors $A, B \in V_1 \otimes V_2 \otimes V_3$, i.e.,

$$
A = A_{\alpha\beta\gamma}\,|\alpha\rangle\,|\beta\rangle\,|\gamma\rangle\,, \quad \text{and} \quad B = B_{\delta\varepsilon\zeta}\,|\delta\rangle\,|\varepsilon\rangle\,|\zeta\rangle\,,
\tag{31}
$$

the inner product is given by

$$
\langle A, B\rangle = A_{\alpha\beta\gamma}{}^{*}B_{\delta\varepsilon\zeta}\,\langle\alpha|\delta\rangle\,\langle\beta|\varepsilon\rangle\,\langle\gamma|\zeta\rangle = A_{\alpha\beta\gamma}{}^{*}B_{\alpha\beta\gamma}\,.
\tag{32}
$$

We would now like to obtain this expression using the contraction operation. Firstly, we extend the notion of conjugation from vectors to tensors, namely by again complex conjugating the components, transforming bras into kets and vice versa, and additionally reversing the tensor product order. For the tensor $A \in V_1 \otimes V_2 \otimes V_3$ from Eq. (31), we then find

$$
\bar{A} = A_{\alpha\beta\gamma}{}^{*}\,\langle\gamma|\,\langle\beta|\,\langle\alpha|\,,
\tag{33}
$$

so that $\bar{A} \in V_3^{*} \otimes V_2^{*} \otimes V_1^{*}$. This space is isomorphic to the dual space $(V_1 \otimes V_2 \otimes V_3)^{*}$, so that we can think of $\bar{A}$ as a linear functional whose application onto another tensor $B$ coincides with $\langle A, B\rangle$. Furthermore, this application can be realised by applying the contraction map, i.e., $\langle A, B\rangle = \mathcal{C}(\bar{A} \otimes B)$ where each index of $\bar{A}$ is contracted with the matching index of $B$.

However, this final identification fails when the tensors also contain bras. For example, if we define the tensors

$$
A = A_{\alpha\beta\gamma}\,|\alpha\rangle\,\langle\beta|\,\langle\gamma|\,, \quad \text{and} \quad B = B_{\delta\varepsilon\zeta}\,|\delta\rangle\,\langle\varepsilon|\,\langle\zeta|\,,
\tag{34}
$$

we find

$$
\begin{aligned}
\langle A, B \rangle &= A_{\alpha\beta\gamma}{}^{*} B_{\delta\varepsilon\zeta} \langle \alpha | \delta \rangle \langle \varepsilon | \beta \rangle \langle \zeta | \gamma \rangle \\
&= A_{\alpha\beta\gamma}{}^{*} | \gamma \rangle | \beta \rangle \langle \alpha | (-1)^{|\varepsilon|}(-1)^{|\zeta|} B_{\delta\varepsilon\zeta} | \delta \rangle \langle \varepsilon | \langle \zeta | ,
\end{aligned}
\tag{35}
$$

where we used fermionic reordering and the fact that $A$ and $B$ are even to go from the first to the second line.

While this result does not coincide with $\mathcal{C}(\overline{A} \otimes B)$, it can be written as a contraction if we introduce a new operation, $\mathcal{P}(B)$, which adds extra parity factors to the dual vectors (bra indices) of $B$, i.e.,

$$
\mathcal{P}(B) = (-1)^{|\varepsilon|}(-1)^{|\zeta|} B_{\delta\varepsilon\zeta} | \delta \rangle \langle \varepsilon | \langle \zeta | .
\tag{36}
$$

With this definition we indeed find $\langle A, B \rangle = \mathcal{C}(\overline{A}\mathcal{P}(B))$, giving the correct result for both Eq. (31) and Eq. (34). For the norm of a fermionic tensor, one similarly obtains

$$
||A|| = \sqrt{\langle A, A \rangle} = \sqrt{\mathcal{C}(\overline{A}\mathcal{P}(A))},
\tag{37}
$$

yielding the Frobenius norm of the tensor entries.

At this point, one might question why we did not include these parity factors in the definition of the conjugate. The main rationale behind this is that the current definition of tensor conjugation "commutes" with contractions, i.e., for any two tensors $A$ and $B$ that admit some contraction, it holds that $\mathcal{C}(\overline{A} \otimes \overline{B}) = \overline{\mathcal{C}(A \otimes B)}$ with $\mathcal{C}$ some contraction over compatible indices. This would no longer be true if we alter the definition of the conjugation to include the parity factors. Take for example the tensors $A$ and $B$ of Eq. (34) and suppose that we first conjugate them with the inclusion of a parity factor on the dual spaces, i.e.,

$$
(-1)^{|\beta|+|\gamma|} A_{\alpha\beta\gamma}{}^{*} | \gamma \rangle | \beta \rangle \langle \alpha | , \quad \text{and} \quad (-1)^{|\varepsilon|+|\zeta|} B_{\delta\varepsilon\zeta}{}^{*} | \zeta \rangle | \varepsilon \rangle \langle \delta | ,
\tag{38}
$$

and then contract the $\gamma$ and $\delta$ indices,

$$
(-1)^{|\alpha|}(-1)^{|\delta|} \delta_{\gamma\delta} A_{\alpha\beta\gamma}{}^{*} B_{\delta\varepsilon\zeta}{}^{*} | \zeta \rangle | \varepsilon \rangle | \beta \rangle \langle \alpha | .
\tag{39}
$$

Compare this to contracting first,

$$
\delta_{\gamma\delta} A_{\alpha\beta\gamma} B_{\delta\varepsilon\zeta} | \alpha \rangle \langle \beta | \langle \varepsilon | \langle \zeta | ,
\tag{40}
$$

and then conjugating with the inclusion of parity factors,

$$
(-1)^{|\beta|+|\varepsilon|+|\zeta|} \delta_{\gamma\delta} A_{\alpha\beta\gamma}{}^{*} B_{\delta\varepsilon\zeta}{}^{*} | \zeta \rangle | \varepsilon \rangle | \beta \rangle \langle \alpha | = (-1)^{|\alpha|} \delta_{\gamma\delta} A_{\alpha\beta\gamma}{}^{*} B_{\delta\varepsilon\zeta}{}^{*} | \zeta \rangle | \varepsilon \rangle | \beta \rangle \langle \alpha | .
\tag{41}
$$

These operations yield different results: the former contains an additional parity factor on the contracted leg. Indeed, the $\mathcal{P}$ operation does not "commute" in any natural way with contractions, so that $\mathcal{C}(\mathcal{P}(A) \otimes \mathcal{P}(B))$ and $\mathcal{P}(\mathcal{C}(A \otimes B))$ cannot generically be identified. The $\mathcal{P}$ operation will always be explicitly denoted, also in the diagrammatic notation as explained in the next subsection.

Furthermore, the need to apply $\mathcal{P}$ to the second argument in the contraction, in order to be able to identify the contraction with the natural application of the linear functional extends to arbitrary linear maps. As another example hereof, we consider the application of an operator,

$$
T : V \otimes V^{*} \to V \otimes V^{*} : | \beta_1 \rangle \langle \beta_2 | \mapsto T^{\beta_1 \beta_2}_{\alpha_1 \alpha_2} | \alpha_1 \rangle \langle \alpha_2 | .
\tag{42}
$$

This operator is naturally identified with the tensor $T^{\beta_1 \beta_2}_{\alpha_1 \alpha_2} | \alpha_1 \rangle \langle \alpha_2 | | \beta_2 \rangle \langle \beta_1 |$, where here and henceforth we will use superscript (subscript) indices for the (co)domain. The application of this operator to an argument $v = v_{\beta_1 \beta_2} | \beta_1 \rangle \langle \beta_2 | \in V \otimes V^{*}$ is equivalent to contracting the

tensor $T$ with the tensor $v$, supplemented with parity factors for the dual components of $v$, i.e., with $\mathcal{P}(v)$. Indeed,

$$
\begin{aligned}
T(v) &= T^{\beta_1\beta_2}_{\alpha_1\alpha_2} v_{\beta_1\beta_2} |\alpha_1\rangle \langle\alpha_2| \\
&= \mathcal{C}_{\beta\gamma}\Big( T^{\beta_1\beta_2}_{\alpha_1\alpha_2} |\alpha_1\rangle \langle\alpha_2| |\beta_2\rangle \langle\beta_1| (-1)^{|\gamma_2|} v_{\gamma_1\gamma_2} |\gamma_1\rangle \langle\gamma_2| \Big) \\
&= \mathcal{C}(T\,\mathcal{P}(v)) .
\end{aligned}
\tag{43}
$$

Note that with these definitions and conventions, the fermionic tensor network machinery, based on the elementary operations of taking tensor products ($\otimes$), reordering ($\mathcal{F}$), contracting ($\mathcal{C}$), conjugating ($\bar{\ }$) and placing parities on dual vectors ($\mathcal{P}$), satisfies the criteria we put forward in the previous section. Indeed,

- the ordering of (co)vectors in the definition of fermionic tensors is important and extra minus signs will appear when permuting them due to the swapping prescription $\mathcal{F}$,

- we required all tensors to be even so that they can be shuffled around, making the idea of contracting larger networks consistent and manageable,

- each index has a directionality, captured by its correspondence to a vector or a dual vector. This can be incorporated in a consistent contraction routine.

## 2.4 Fermionic tensor networks and diagrammatic notation

Fermionic tensors and their contractions can be represented using tensor network diagrams, just like their bosonic counterparts. For instance, we can depict the tensor $A$ from Eq. (26) with a diagram

$$
A = A_{\alpha\beta\gamma\delta} |\alpha\rangle |\beta\rangle \langle\gamma| \langle\delta| =
\tag{44}
$$

where each leg corresponds to a (dual) vector space labeled with the same Greek index as in Eq. (26) and where the arrows indicate whether we have vectors (outgoing) or dual vectors (incoming). Note where we draw each leg on the diagram does not matter. Indeed, according to our established formalism, a tensor network is completely defined by specifying which indices are contracted. The arrangement of the corresponding lines in the diagram thus has no real meaning. What is important though, is that upon defining the tensor it is clear which leg corresponds to which space in the algebraic expression, necessarily having a certain order. Therefore, we will adhere to the convention of ordering tensor legs counterclockwise in the diagrams to match the order in their defining algebraic expressions, unless explicitly stated otherwise.

As always, contractions between different tensor indices are represented by connecting the corresponding legs. For example, we can represent the contraction in Eq. (30) as

$$
\begin{aligned}
\mathcal{C}_{\beta\delta}\big(B \otimes C\big) &= \\
&= \mathcal{C}_{\beta\delta}\big( B_{\alpha\beta\gamma} C_{\delta\varepsilon} |\alpha\rangle |\beta\rangle \langle\gamma| \langle\delta| \langle\varepsilon| \big) \\
&= \left( \sum_\beta (-1)^{|\beta||\gamma|} B_{\alpha\beta\gamma} (-1)^{|\beta|} C_{\beta\varepsilon} \right) |\alpha\rangle \langle\gamma| \langle\varepsilon| .
\end{aligned}
\tag{45}
$$

Note that in this example the resulting indices are indeed not in counterclockwise order when compared to the algebraic expression. Furthermore, note that upon linearizing the diagram as in Eq. (15), we would find the connected arrow to point to the right, indicating that a parity factor will be added in the contraction. Indeed, we expect a supertrace here.

Taking the conjugate of a fermionic tensor, all vectors are turned into dual vectors and vice-versa, reversing the order of the tensor product. Diagrammatically, this implies that all arrows have to be flipped. I.e., if we have a tensor

$$A = A_{\alpha\beta\gamma} |\alpha\rangle \langle\beta| \langle\gamma| = \quad , \tag{46}$$

its conjugate can be depicted as

$$\bar{A} = A_{\alpha\beta\gamma}{}^* |\gamma\rangle |\beta\rangle \langle\alpha| = \quad . \tag{47}$$

However, comparing the algebraic expression with the diagram, the legs are no longer ordered counterclockwise. To retain the convention, one could mirror the diagram across an arbitrary axis, for instance yielding

$$\bar{A} = A_{\alpha\beta\gamma}{}^* |\gamma\rangle |\beta\rangle \langle\alpha| = \quad . \tag{48}$$

We will typically order tensor legs in this way and thus draw mirrored conjugate tensors. However, note that it is only a convention.

The final operation from the previous subsection, namely $\mathcal{P}(A)$, can be obtained by contracting every incoming leg of a tensor with a corresponding $P$ (parity) tensor, given by

$$\alpha \dashleftarrow P \leftarrow \beta \; = (-1)^{|\alpha|} \delta_\alpha^\beta |\alpha\rangle \langle\beta| \,. \tag{49}$$

For the quadratic norm of $A$, one then obtains

$$|\langle A, A\rangle|^2 = \mathcal{C}(\bar{A}\mathcal{P}(A)) = \quad \tag{50}$$

Let us elaborate on this parity tensor. First, note that it differs only slightly from the unit tensor, defined as

$$\alpha \dashleftarrow I \leftarrow \beta \; = \delta_\alpha^\beta |\alpha\rangle \langle\beta| \,. \tag{51}$$

Indeed, internally permuting the bra and ket in $P$ results in a tensor with the same entries as $I$ but when comparing them with their composite (co)vectors in the same order, their entries do differ and the difference exactly corresponds to the desired parity factors. Another fundamental property of the parity tensor is that it can be pulled through any even tensor, as a result

$$\mathcal{P}(A) = \quad = \quad \tag{52}$$

Hence, we could equally well calculate the quadratic norm of $A$ by using a single $P$ on the outgoing leg of $A$ in Eq. (50) instead of two on the incoming legs. While this would reduce the number of parity tensors in this particular case, it would not hold true for any extended tensor network where $\bar{A}$ is not solely contracted with $A$. In fact, adopting an alternative convention could introduce even more parities than the ones we eliminated here.

## 2.5 Linear maps, adjoints and hermiticity

When a tensor encodes a specific linear map $T$ that needs to be decomposed (see Sec. 2.7) or appears in an eigenvalue or linear problem, it can be useful to adopt an additional convention, whereby the legs corresponding to the (co)domain are placed to the right (left) or top (bottom) of the tensor. The arguments will thus be appended from the right or from the top. For example, the application of $T$ in Eq. (43) translates to

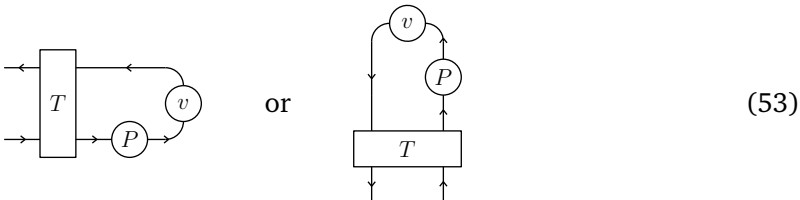

$$\text{or} \qquad\qquad (53)$$

With this convention in mind, we now elaborate on different properties associated with linear maps between or on Hilbert spaces, such as unitarity or hermiticity, as these also play a crucial role in typical tensor network algorithms. Because of the appearance of parity operators in the diagrammatic representation of the inner product of fermionic tensors Eq.(35), the diagrammatic representation of such properties needs to be adjusted accordingly. To do this, we first deduce the diagrammatic notation for applying the adjoint or Hermitian conjugation of an arbitrary fermionic linear map $T$. We will demonstrate this for $T : V_1 \otimes V_2^* \otimes V_3 \to W_1 \otimes W_2^*$, isomorphic to the tensor

$$T = T^{\beta_1\beta_2\beta_3}_{\alpha_1\alpha_2} |\alpha_1\rangle\langle\alpha_2|\langle\beta_3||\beta_2\rangle\langle\beta_1| = \qquad\qquad (54)$$

but the results can readily be extended to arbitrary linear maps. The adjoint of the linear map, $T^\dagger : W_1 \otimes W_2^* \to V_1 \otimes V_2^* \otimes V_3$ is defined by the property that $\langle v, T^\dagger w\rangle = \langle Tv, w\rangle$ should hold $\forall v \in V_1 \otimes V_2^* \otimes V_3$ and $\forall w \in W_1 \otimes W_2^*$. As discussed in the previous section, we can express this as a contraction of the corresponding $\mathbb{Z}_2$-graded tensors with $P$ tensors to correct for unwanted supertraces. Indeed,

$$T(v) = \mathcal{C}(T\,\mathcal{P}(v)) = \qquad\qquad (55)$$

and

$$T^\dagger(w) = \mathcal{C}(T^\dagger\,\mathcal{P}(w)) = \qquad\qquad (56)$$

The defining property for $T^\dagger$ can thus be rephrased as

$$\mathcal{C}(\bar{v}\,\mathcal{P}(T^\dagger(w))) = \mathcal{C}(\overline{T(v)}\,\mathcal{P}(w)), \qquad\qquad (57)$$

or equivalently

$$\text{(diagram)} = \text{(diagram)} \tag{58}$$

since

$$\overline{T(v)} = \text{(diagram)} \tag{59}$$

when we draw the conjugate tensors by mirroring across the domain-codomain axis. We conclude that applying $T^\dagger$ corresponds to contraction with $\bar{T}$. In this way, the (co)domain arrows on the right (left) of the original tensor are now on the other side. Note again that the reflection of $\bar{T}$ does not change the tensor. It is rather a diagrammatic operation, redrawing the legs, to retain the counterclockwise ordering and the meaning of the right (left) side as (co)domain of the homomorphism. In symbols, we can relabel $\alpha \leftrightarrow \beta$ so that the $\beta$ $(\alpha)$ indices keep labeling the (co)domain,

$$\bar{T} = T^{\beta_1 \beta_2 \beta_3 *}_{\alpha_1 \alpha_2} |\beta_1\rangle \langle\beta_2| |\beta_3\rangle |\alpha_2\rangle \langle\alpha_1| = T^{\alpha_1 \alpha_2 \alpha_3 *}_{\beta_1 \beta_2} |\alpha_1\rangle \langle\alpha_2| |\alpha_3\rangle |\beta_2\rangle \langle\beta_1|, \tag{60}$$

showing that the components of the tensor isomorphic to $T^\dagger$ are given by $T^{\alpha_1 \alpha_2 \alpha_3 *}_{\beta_1 \beta_2}$ when those of $T$ are given by $T^{\beta_1 \beta_2 \beta_3}_{\alpha_1 \alpha_2}$, i.e., the standard result.

Building on this, an arbitrary linear operator, $T : V_1 \otimes V_2^* \otimes \cdots \otimes V_n \to V_1 \otimes V_2^* \otimes \cdots \otimes V_n$, corresponding to the tensor

$$T = T^{\beta_1 \beta_2 \cdots \beta_n}_{\alpha_1 \alpha_2 \cdots \alpha_n} |\alpha_1\rangle \langle\alpha_2| \cdots |\alpha_n\rangle \langle\beta_n| \cdots |\beta_2\rangle \langle\beta_1| = \text{(diagram)} \tag{61}$$

is Hermitian when $T = T^\dagger$ or diagrammatically,

$$\text{(diagram)} = \text{(diagram)} \tag{62}$$

Note that we slightly changed the rectangular shape of the tensor to easily discern between $T$ and $\bar{T}$. In components we get

$$T^{\beta_1 \beta_2 \cdots \beta_n}_{\alpha_1 \alpha_2 \cdots \alpha_n} = T^{\alpha_1 \alpha_2 \cdots \alpha_n *}_{\beta_1 \beta_2 \cdots \beta_n}, \tag{63}$$

which, grouping the $\alpha$ and $\beta$ indices, comes down to $T^{\boldsymbol\beta}_{\boldsymbol\alpha} = T^{\boldsymbol\alpha *}_{\boldsymbol\beta}$. Introducing the notation $T^{\dagger \boldsymbol\beta}_{\phantom{\dagger}\boldsymbol\alpha}$ for the latter, we obtain the standard hermiticity condition for the tensor elements, $T^{\boldsymbol\beta}_{\boldsymbol\alpha} = T^{\dagger \boldsymbol\beta}_{\phantom{\dagger}\boldsymbol\alpha}$.

Analogously, an arbitrary linear map $T : W_1 \otimes W_2 \otimes \cdots \otimes W_n^* \to V_1 \otimes V_2^* \otimes \cdots \otimes V_m$,

$$T = T^{\beta_1 \beta_2 \cdots \beta_n}_{\alpha_1 \alpha_2 \cdots \alpha_m} |\alpha_1\rangle \langle\alpha_2| \cdots |\alpha_m\rangle |\beta_n\rangle \cdots \langle\beta_2| \langle\beta_1| = \text{(diagram)} \tag{64}$$

is defined to be left-isometric when application of $T$ followed by $T^\dagger$ onto an arbitrary tensor $v \in V_1 \otimes V_2 \otimes \cdots \otimes V_n^*$ leaves it invariant. We can again reformulate this as a contraction. $T$ is left-isometric when $T^\dagger(T(v)) = \mathcal{C}(\bar{T}\mathcal{P}(\mathcal{C}(T\mathcal{P}(v)))) = v$, or diagrammatically

$$ \tag{65} $$

such that a left-isometry is characterized by

$$ \tag{66} $$

Note the appearance of a $P$ tensor in the right-hand side of the equation, which might be unexpected. However, as this $P$ tensor has its outgoing leg arrow (which was associated to its first index) on the right, it means that this $P$ tensor actually contributes a factor

$$(-1)^{|\beta_n|}\delta_{\alpha_n,\beta_n}|\beta_n\rangle\langle\alpha_n| = \delta_{\alpha_n,\beta_n}\langle\alpha_n||\beta_n\rangle . \tag{67}$$

Similarly, $T$ is right-isometric when application of $T^\dagger$ followed by $T$ on an arbitrary tensor $v \in V_1 \otimes V_2^* \otimes \cdots \otimes V_n$ yields $v$ again, i.e., $T(T^\dagger(v)) = \mathcal{C}(T\mathcal{P}(\mathcal{C}(\bar{T}\mathcal{P}(v)))) = v$, or diagrammatically

$$ \tag{68} $$

We conclude that $T$ is a right-isometry if

$$ \tag{69} $$

In case of a unitary operator, both conditions apply. These isometry conditions can again be expressed in components yielding standard results: $T$ is left- (right-)isometric when $T^{\dagger\beta}_{\ \alpha}T^{\gamma}_{\ \beta} = \delta^{\gamma}_{\alpha}$ ($T^{\beta}_{\alpha}T^{\dagger\gamma}_{\ \beta} = \delta^{\gamma}_{\alpha}$), where, because of the how the $P$ tensors appear in these diagrams, no actual signs arise.

## 2.6 Tensor index manipulations

In order to express some more involved tensor properties (e.g. hermiticity of fermionic Matrix Product Operators (MPOs) (see Sec. 6.2 and Appendix A.2)) or to derive tensor decompositions (see Sec. 2.7), unitary index manipulations such as fusing legs and flipping arrows will be indispensable. Again, we will perform these operations by relating them to a contraction with an appropriate tensor. Therefore, we start by introducing the so-called fusers, representing a unitary map between two legs (corresponding to the spaces $V_1$ and $V_2$) and a single leg (corresponding to $V_1 \otimes V_2$) with a higher dimension. Associating a tensor

$$F = F^{\beta_1}_{\alpha_1\alpha_2}|\alpha_1\rangle|\alpha_2\rangle\langle\beta_1| = \tag{70}$$

to this map, we thus have that

$$
\cdots = \cdots \quad \text{and} \quad \cdots = \cdots \tag{71}
$$

Dual vectors can also be incorporated, e.g.

$$
F = F^{\beta_1}_{\alpha_1\alpha_2} \, |\alpha_1\rangle \, \langle\alpha_2| \, |\beta_1\rangle = \quad \begin{array}{c} \alpha_1 \\ \alpha_2 \end{array} \!\!\!\!\! \boxed{F} \!\!\! \longrightarrow \beta_1 \;, \tag{72}
$$

with $F$ unitary so that

$$
\cdots = \cdots \tag{73}
$$

and

$$
\cdots = \cdots \tag{74}
$$

As in the previous section, the appearance of $P$ tensors in these diagrams should be correctly interpreted, because the arrows indicate that the first (second) index of $P$ is the right (left) leg in each of the $P$ tensors in the above two diagrams. E.g. in Eq. (73) the second leg of the $P$ tensor from Eq. (49) should be contracted with the third leg of $F$ while the $P$'s first leg should be contracted with $\bar{F}$'s first leg. In symbols the left side of Eq. (73) hence becomes

$$
\mathcal{C}_{\gamma\beta_1}\Big(\mathcal{C}_{\gamma'\alpha_1'}\Big(F^{\beta_1}_{\alpha_1\alpha_2} \, |\alpha_1\rangle \, \langle\alpha_2| \, |\beta_1\rangle \, (-1)^{|\gamma|}\delta^{\gamma}_{\gamma'} \, |\gamma'\rangle \, \langle\gamma| \, F^{\alpha_1'}_{\beta_1'\beta_2'}{}^* \, \langle\alpha_1'| \, |\beta_2'\rangle \, \langle\beta_1'|\Big)\Big)
$$

$$
= F^{\gamma}_{\alpha_1\alpha_2} F^{\dagger\beta_1'\beta_2'}_{\gamma} \, |\alpha_1\rangle \, \langle\alpha_2| \, |\beta_2'\rangle \, \langle\beta_1'| \tag{75}
$$

$$
= \delta^{\beta_1'}_{\alpha_1}\delta^{\beta_2'}_{\alpha_2} \, |\alpha_1\rangle \, \langle\alpha_2| \, |\beta_2'\rangle \, \langle\beta_1'|
$$

$$
= \delta^{\beta_1'}_{\alpha_1} \, |\alpha_1\rangle \, \langle\beta_1'| \quad \delta^{\beta_2'}_{\alpha_2} \, \langle\alpha_2| \, |\beta_2'\rangle \;,
$$

where from the second to the third line we used that $F^{\gamma}_{\alpha} F^{\dagger\beta}_{\gamma} = \delta^{\beta}_{\alpha}$. The final result is consistent with the right side of Eq. (73), again because of the internal permutation of the indices of the $P$ tensor. Note that doing the same for an identity tensor would exactly lead to a parity factor.

Another useful type of unitary tensors are flippers. Consider for instance the linear map $f : V \to V^* : |\alpha\rangle \mapsto f^{\beta}_{\alpha} \, \langle\beta|$ with associated tensor

$$
f = f^{\beta}_{\alpha} \, \langle\alpha| \, \langle\beta| = \quad \alpha \longrightarrow \!\!\!\!\! \boxed{f} \!\!\! \longleftarrow \beta \;, \tag{76}
$$

where we choose the orientation of the triangular shape according to the arrow direction of the domain. Now, let $f$ be unitary, satisfying

$$
\cdots = \cdots \quad \text{and} \quad \cdots = \cdots \tag{77}
$$

e.g. by choosing $f^{\beta}_{\alpha} = \delta^{\beta}_{\alpha}$. The application of $f$ to a leg of a tensor then corresponds to flipping the arrow of this leg, i.e., to going from a vector to a dual vector. Similarly, one could use $f^{\dagger}$ to unitarily flip incoming legs. To illustrate this, consider again the $A$ tensor from Eq. (26). We could flip its first arrow by applying $f$ which corresponds to contracting with the second leg of the $f$ tensor. Additionally, we could flip its fourth leg by applying $f^{\dagger}$. Application of this

operator corresponds to contraction with the second leg of the $\bar{f}$ tensor but now also including a $P$ tensor in between. Consequently, the flipped object is given by

$$\tag{78}$$

Note that the contraction of $\bar{f}$ and $P$ is unitary as well. Therefore, one could equally well just contract with $\bar{f}$ to unitarily flip the fourth leg of $A$. More generally any invertible transformation could be utilized, yielding non-unitary flippers. However, these would spoil the orthogonality of the decompositions defined in the following section, such that it is often more convenient to restrict to unitary flippers.

## 2.7 Tensor decompositions

Typical tensor network algorithms heavily rely on tensor versions of matrix decompositions such as QR, SVD and the polar decomposition. To get these for fermionic tensors, consider again an arbitrary tensor map

$$A = A^{\beta_1 \beta_2 \cdots \beta_n}_{\alpha_1 \alpha_2 \cdots \alpha_m} |\alpha_1\rangle \langle\alpha_2| \cdots |\alpha_m\rangle |\beta_n\rangle \cdots \langle\beta_2| \langle\beta_1| = \qquad \tag{79}$$

One could interpret its matrix representation $A^{\boldsymbol{\beta}}_{\boldsymbol{\alpha}}$ as a fused version of $A$ where all the domain legs on the right and all the codomain legs on the left are fused together (using some specific choice for the fusers), i.e.,

$$A^{\boldsymbol{\beta}}_{\boldsymbol{\alpha}} |\boldsymbol{\alpha}\rangle \langle\boldsymbol{\beta}| = \qquad \tag{80}$$

thus yielding a two-leg tensor, i.e., a matrix. As this two-leg object is even, it only consists of a non-zero even-even and odd-odd sector. In other words, it is a block-diagonal matrix. If we assume a single even and odd basis vector for each of the composite spaces of $A$, $A^{\boldsymbol{\beta}}_{\boldsymbol{\alpha}}$ is just a $m \times n$ matrix in both these sectors. On these matrix blocks one can, for instance, perform a QR (LQ) decomposition, $A^{\boldsymbol{\beta}}_{\boldsymbol{\alpha}} = Q^{\gamma}_{\boldsymbol{\alpha}} R^{\boldsymbol{\beta}}_{\gamma}$ ($A^{\boldsymbol{\beta}}_{\boldsymbol{\alpha}} = L^{\gamma}_{\boldsymbol{\alpha}} Q^{\boldsymbol{\beta}}_{\gamma}$). Here, $Q$ is a left- (right-)isometric matrix while $R$ ($L$) is an upper- (lower-)triangular, square matrix as long as $m \geq n$ ($m \leq n$). When $m < n$ ($m > n$), $Q$ is square and unitary while $R$ ($L$) consists of a upper- (lower-)triangular part supplemented with additional dense columns (rows). We can again promote these matrices to two-leg tensors so that the fused $A$ tensor decomposes as

$$\qquad = \qquad \text{and} \qquad \qquad = \qquad \tag{81}$$

with

$$\qquad = \qquad \text{and} \qquad \qquad = \qquad \tag{82}$$

for *QR* and *LQ* decomposition respectively. Now, we can split the outer legs in the decomposition in a way consistent with how we fused them before to obtain a tensorial version of the *QR* (*LQ*) decomposition for *A*. We illustrate this for the *QR* decomposition of a tensor

$$A = A^{\beta_1\beta_2}_{\alpha_1\alpha_2} |\alpha_1\rangle \langle\alpha_2| \langle\beta_2||\beta_1\rangle = \quad \tag{83}$$

Transforming *A* to a two-leg object is possible with

$$\tag{84}$$

where we choose

$$F = (-1)^{|\alpha_2|} |\alpha_1\rangle \langle\alpha_2| \langle\boldsymbol{\alpha}| = $$
$$\tilde{F} = (-1)^{|\beta_1|} \langle\beta_1||\beta_2\rangle \langle\boldsymbol{\beta}| = \tag{85}$$

for the fusers. An immediate consequence is that

$$\tag{86}$$

Note that other unitary fusers would yield an identical result. However, this choice was made so that the entries of the two-leg tensor are exactly $A^{\boldsymbol{\alpha}}_{\boldsymbol{\beta}}$. Performing the *QR* decomposition on the two-leg *A*, we obtain

$$\tag{87}$$

where *Q* is left-isometric and thus

$$\tag{88}$$

Splitting the outer legs in the decomposition is possible in the same way as in Eq. (86), yielding

$$\tag{89}$$

i.e., a full tensorial version of the decomposition. While it is possible to split the inner leg, it typically remains fused as splitting has no well-defined purpose here. Note that the multi-leg *Q* is still left-isometric since

$$\tag{90}$$

We conclude that *QR* (*LQ*) decomposition of an arbitrary fermionic tensor yields a left- (right-) isometric *Q* with the same leg structure on the left (right) and a *R* (*L*) with the same leg structure on the right (left) as the original tensor.

The polar decomposition can be introduced along the same lines. Here, $A_\alpha^\beta = Q_\alpha^\gamma R_\gamma^\beta$ ($A_\alpha^\beta = R_\alpha^\gamma Q_\gamma^\beta$) when $m \geq n$ ($m \leq n$) with $R$ a positive semi-definite and Hermitian matrix while $Q$ is left- (right-)isometric. In this case, it does make sense to split the inner leg in the same way as the outer leg of $R$ as this allows the resulting $R$ tensor to be Hermitian in the fermionic tensor sense.

Finally, we consider the singular-value decomposition $A_\alpha^\beta = U_\alpha^\gamma S_\gamma^{\gamma'} V_{\gamma'}^\beta$ where $U$ is left-isometric, $V$ is right-isometric and $S$ is diagonal and collects the non-zero singular values of $A_\beta^\alpha$. Again, we promote the $U, V$ and $S$ matrices to two-leg fermionic tensors and split the outer legs in the same way as the original tensor while leaving the inner legs fused, yielding

$$\text{(diagram)} \tag{91}$$

where $U$ and $V$ are left- and right-isometries, respectively, in the fermionic tensor sense.

## 2.8 Implementation and combination with other symmetries in tensor networks

At this point we have defined fermionic tensors as objects living in graded Hilbert spaces, as well as contractions, permutations and decompositions of these objects. In numerical tensor network algorithms, we will store tensors as a certain data structure, and implement these tensor manipulations as efficient operations on these structures. Let us explain here how we can translate this formalism of graded tensor networks into an efficient computer code, again without going into the details of the implementation itself.

Fortunately, we can integrate the properties of a graded tensor straightforwardly within the framework of tensor networks with global symmetries. Consider, for example, again the three-leg tensor

$$A = A_{\alpha\beta\gamma} |\alpha\rangle \langle\beta| \langle\gamma| .$$

On each leg/index, this tensor has two parity sectors with a given dimension, so we can draw up a table that lists all different sectors that lead to global even parity of the tensor:

| $\alpha$ | $\beta$ | $\gamma$ |
|---|---|---|
| 0 | 0 | 0 |
| 1 | 1 | 0 |
| 1 | 0 | 1 |
| 0 | 1 | 1 |

$$\tag{92}$$

Corresponding to each row in this table or "fusion tree", there is a numerical array or "tensor block" with the dimensions of the corresponding sectors. Additionally, we have to keep track of the direction of the arrow on each leg. This data structure is similar to how one would define a tensor with a global (non-graded) $\mathbb{Z}_2$ symmetry. The fermionic nature of the graded tensor comes in when implementing tensor operations. For example, the permutation of this same tensor,

$$A = (-1)^{|\alpha||\beta|} A_{\alpha\beta\gamma} \langle\beta| |\alpha\rangle \langle\gamma| , \tag{93}$$

needs to be implemented by explicitly including the minus signs within the different blocks. As explained before, similar minus signs appear when contracting kets with bra's or while adding $P$'s to determine a certain tensor decomposition.[8]

---

[8]When placing parity tensors, the corresponding minus signs immediately apply to complete matrix blocks, i.e., for certain sets of fusion trees, and can therefore be kept as simple flags. The typical addition of $P$ tensors to fermionic tensor network contractions therefore only results in a minimal overhead.

The data structure for a graded tensor thus is very similar to the structure of tensors with other (non-graded) global tensor symmetries such as $\mathbb{Z}_N$, U(1) or SU($N$). As a result, it can be straightforwardly combined by taking a direct product with these other symmetries. This also yields a direct product between representations. I.e., the irreducible representations (irreps) of the group symmetries will receive an additional fermion parity label. Which combinations of group irreps and parities are possible are dictated by the physics. A straightforward example is the combination of the fermionic grading (denoted as $f\mathbb{Z}_2$) with global fermion number conservation (i.e., a U(1) symmetry) leading to a direct product structure $f\mathbb{Z}_2 \otimes$ U(1). On each leg of a tensor with this symmetry structure, we will have different sectors such as $(0,0)$, $(0,2)$, …, and $(1,1)$, $(1,3)$, …, where the first label indicates the fermion parity, and the second label the U(1) charge or number. These particular combinations indicate that states with odd fermion parity also carry an odd U(1) charge, corresponding to the common case where the elementary unit of charge in the system is carried by fermionic degrees of freedom.

As a more involved example, we consider a system of spin-$\frac{1}{2}$ fermions on a lattice, where we want to combine the graded fermionic sector with the SU(2) spin rotation symmetry, leading to a direct product structure $f\mathbb{Z}_2 \otimes$ SU(2). On every site we have a four-dimensional Hilbert space spanned by the following states:

|  | $f\mathbb{Z}_2$ | SU(2) |
|---|---|---|
| $|0\rangle$ | 0 | 0 |
| $|\uparrow\downarrow\rangle$ | 0 | 0 |
| $\{|\uparrow\rangle, |\downarrow\rangle\}$ | 1 | 1/2 |

where we have explicitly noted the charge sector of each state.

As we mentioned before, the relevant symmetries and their charge sectors are determined by the physics of the problem under consideration by means of the Hamiltonian. Hence, a fermionic model should have a fermionic tensor representing its Hamiltonian. Consider for instance the simple model of spinless fermions hopping on a 1D chain of length $N$,

$$H = \sum_{i=1}^{N-1} -t\left(a_i^\dagger a_{i+1} + \text{h.c.}\right) = \sum_{i=1}^{N-1} h_i\,. \tag{94}$$

The local two-site terms $h_i$ are even and can therefore be written as

$$\begin{aligned}
&\quad\; = (h_i)_{\alpha_i \alpha_{i+1}}^{\beta_i \beta_{i+1}} |\alpha_i\rangle_i |\alpha_{i+1}\rangle_{i+1} \langle\beta_{i+1}|_{i+1} \langle\beta_i|_i \\
&\quad\; = -t\,|1\rangle_i |0\rangle_{i+1} \langle 1|_{i+1} \langle 0|_i - t\,|0\rangle_i |1\rangle_{i+1} \langle 0|_{i+1} \langle 1|_i\,.
\end{aligned} \tag{95}$$

This is a fermionic tensor described by our formalism and we could imagine approximating the ground state of $H$ by an MPS through the application of standard MPS optimization methods like DMRG (see Sec. 3.3). However, suppose that rather than expressing $H$ through $h_i$, we would like to find tensors for the separate creation and annihilation operators. At first sight, this looks impossible. Indeed, these are parity changing operators and therefore correspond to odd tensors that we did not allow for in our framework. However, we can circumvent this problem by given the tensors representing these operators an auxiliary parity-odd space (leg), which we denote by a bra with a round bracket (wiggly line). We could for instance define

$$\qquad = |1\rangle \langle 0| (1|\,, \qquad \text{and} \qquad = |1\rangle |0\rangle \langle 1|\,. \tag{96}$$

Contracting these through their auxiliary leg and multiplying by $-t$, we obtain the first term in $h_i$. Analogously, we could construct

$$\boxed{a^\dagger} = |1)|1\rangle\langle 0|\,, \qquad \text{and} \qquad \boxed{a} = |0\rangle\langle 1|(1|\,, \tag{97}$$

yielding the second term in $h_i$ after contraction. More generally, we could put any invertible tensor and its inverse on the auxiliary leg and make an infinite set of consistent $a$ and $a^\dagger$ tensors. Their exact form does not matter, the fact that they give rise to a correct $h_i$ and thus $H$ does. Note that one could also make pairing terms by combining the two different creation/annihilation operators from Eq. (96) and Eq. (97). Similarly, one can build higher-order interacting terms as well.

Now let us return to the more involved example of the combination of $f\mathbb{Z}_2$ and SU(2) symmetry with basis vectors as listed in the table above. The creation operator that acts on these spaces is again a parity changing operator (i.e., it adds a fermion) but now the creation of a single fermion with either up- or downwards spin also does not retain the SU(2) charge. Both of these issues can again be resolved by adding an odd auxiliary leg with a spin-$\frac{1}{2}$ charge, i.e.,

$$\boxed{a^\dagger}^\sigma = \delta_{\sigma,\uparrow}\big(|\uparrow\rangle\langle 0| + |\uparrow\downarrow\rangle\langle\downarrow|\big)(\uparrow| + \delta_{\sigma,\downarrow}\big(|\downarrow\rangle\langle 0| - |\uparrow\downarrow\rangle\langle\uparrow|\big)(\downarrow|\,. \tag{98}$$

Here, the ancilla essentially "carries" both the $\sigma$ spin label, as well as the parity, and ensures the total charge of the tensor is spin-0, and has even fermionic parity. In this tensor, the physical spaces are graded with an even sector spanned by $|0\rangle$ and $|\uparrow\downarrow\rangle$, both parity even states and each transforming as a spin-0 irrep, and an odd sector spanned by $|\uparrow\rangle$ and $|\downarrow\rangle$, transforming together as a spin-$\frac{1}{2}$ irrep. The virtual space is spanned by $(\uparrow|$ and $(\downarrow|$, which also transform together as a spin-$\frac{1}{2}$ irrep with odd fermion parity. We conclude that the fermionic grading is clearly compatible with the SU(2) symmetry, and we can write out the full, symmetry-preserving $4\times 4\times 2$ tensor as

$$\begin{pmatrix} 0 & 0 & 0 & 0 \\ 0 & 0 & 0 & 1 \\ \hline 1 & 0 & 0 & 0 \\ 0 & 0 & 0 & 0 \end{pmatrix}\otimes\begin{pmatrix}1 & 0\end{pmatrix} + \begin{pmatrix} 0 & 0 & 0 & 0 \\ 0 & 0 & -1 & 0 \\ \hline 0 & 0 & 0 & 0 \\ 1 & 0 & 0 & 0 \end{pmatrix}\otimes\begin{pmatrix}0 & 1\end{pmatrix}\,, \tag{99}$$

corresponding to the basis order $\{|0\rangle, |\uparrow\downarrow\rangle; |\uparrow\rangle, |\downarrow\rangle\}$ in the physical space in both the domain and codomain. The second tensor product factor in both terms is associated with the $\sigma$ line, and is denoted as a row vector, because the $\sigma$ line was drawn as part of the domain of the linear map associated with the tensor (even though this is simply a matter of choice).

The two states $\{|0\rangle, |\uparrow\downarrow\rangle\}$ are not distinguished by symmetry, and thus need to be labelled by a two-dimensional degeneracy label. On the other hand, the states $\{|\uparrow\rangle, |\downarrow\rangle\}$ make up a single spin-1/2 multiplet. As in the non-fermionic case, the fact that this is a symmetric tensor implies that it is partially fixed by the Clebsch-Gordan coefficients $C_J^{j_1,j_2}$ (according to the Wigner-Eckart theorem), with the actual degrees of freedom associated with the degeneracy labels.

Indeed, this tensor can be written as

$$
\begin{aligned}
&= C_{\frac{1}{2}}^{0,\frac{1}{2}} \quad \otimes \quad \begin{pmatrix} 1 & 0 \end{pmatrix} \quad + \quad C_0^{\frac{1}{2},\frac{1}{2}} \quad \otimes \quad \begin{pmatrix} 0 \\ -\sqrt{2} \end{pmatrix}\\
&= \quad \otimes \quad \begin{pmatrix} 1 & 0 \end{pmatrix} \quad + \quad \otimes \quad \begin{pmatrix} 0 \\ -\sqrt{2} \end{pmatrix},
\end{aligned}
\tag{100}
$$

where the additional tensor factors are now associated with the two-dimensional degeneracy of the $(e,0)$ space, which appears in the domain for the first term, and in the codomain for the second term. The equality between the non-zero subblocks in Eq. (99) and Eq. (100) can then be established, using

$$
C_{\frac{1}{2}}^{0,\frac{1}{2}} = |{\uparrow}\rangle\,\langle{\uparrow}|\,\langle 0| + |{\downarrow}\rangle\,\langle{\downarrow}|\,\langle 0| , \quad \text{and} \quad C_0^{\frac{1}{2},\frac{1}{2}} = |0\rangle\,[\langle{\downarrow}|\,\langle{\uparrow}| - \langle{\uparrow}|\,\langle{\downarrow}|]/\sqrt{2}. \tag{101}
$$

This shows that the only difference for fermions is the incorporation of an additional label denoting the parity for each of the irreps.

Similarly, by taking the adjoint of $a^\dagger$ and denoting adjoint Clebsch-Gordan coefficients diagrammatically, we obtain

$$
= \quad \otimes \quad \begin{pmatrix} 1 \\ 0 \end{pmatrix} \quad + \quad \otimes \quad \begin{pmatrix} 0 & -\sqrt{2} \end{pmatrix}, \tag{102}
$$

i.e., a symmetric tensor that can be used to represent the annihilation operator. Indeed, contracting $a^\dagger$ and $a$, we can make use of the unitarity of the Clebsch-Gordan coefficients and for instance build the SU(2) symmetric number operator

$$
\begin{aligned}
&= (o,\tfrac{1}{2}) \quad (o,\tfrac{1}{2}) \quad \otimes \quad \begin{pmatrix} 0 & 0 \\ 0 & 2 \end{pmatrix} \quad + \quad (e,0) \quad (o,\tfrac{1}{2}) \quad \otimes \quad 1\\
&= \quad \otimes \quad \begin{pmatrix} 0 & 0 \\ 0 & 2 \end{pmatrix} \quad + \quad \otimes \quad 1\\
&= 2\,|{\uparrow}{\downarrow}\rangle\,\langle{\uparrow}{\downarrow}| + |{\uparrow}\rangle\,\langle{\uparrow}| + |{\downarrow}\rangle\,\langle{\downarrow}|\\
&= a_\sigma^\dagger a_\sigma .
\end{aligned}
\tag{103}
$$

## 2.9 Summary

In this chapter we introduced a fermionic tensor network formalism based on $\mathbb{Z}_2$-graded Hilbert spaces, decomposing in orthogonal even and odd sectors. Taking the tensor product

of these spaces and their duals we get objects like

$$A = A_{\alpha\beta\gamma} |\alpha\rangle \langle\beta| |\gamma\rangle = \quad , \tag{104}$$

where each of the (dual) spaces has a basis of ket (bra) vectors with a definite grading, e.g. $|\alpha|$ is the grading of the $|\alpha\rangle$ basis vector of the first space in $A$. The net grading of a fermionic tensor is even, i.e., tensor entries for net odd parity combinations are zero. The diagrammatic representation of fermionic tensors matches a leg with a space, while the arrows indicate wether the space is dual with an outgoing arrow or non-dual with an incoming arrow. Furthermore, we typically adopt the convention to draw the legs counterclockwise when compared to the algebraic expression.

On these fermionic tensor objects we define some elementary operations:

- $\otimes$: takes the tensor product of two fermionic tensors, e.g.

$$A \otimes B = A_{\alpha\beta\gamma} |\alpha\rangle \langle\beta| |\gamma\rangle B_{\delta\varepsilon} \langle\delta| \langle\varepsilon| , \tag{105}$$

with $A$ as in Eq. (104) and $B = B_{\delta\varepsilon} \langle\delta| \langle\varepsilon|$.

- $\mathcal{F}$: fermionic reordering, swaps two or more spaces by introducing an appropriate parity factor, e.g.

$$\mathcal{F}_{\alpha\beta} \left( A_{\alpha\beta\gamma} |\alpha\rangle \langle\beta| |\gamma\rangle \right) = (-1)^{|\alpha||\beta|} A_{\alpha\beta\gamma} \langle\beta| |\alpha\rangle |\gamma\rangle . \tag{106}$$

We identify a tensor with its internally reordered version and therefore typically do not write down $\mathcal{F}$ explicitly.

- $\mathcal{C}$: tensor contraction of two or more consistent pairs of spaces (they have to be each others dual) with the addition of a supertrace in case of a ket-bra contraction, e.g.

$$\mathcal{C}_{\gamma\delta}(A \otimes B) = \quad$$

$$\tag{107}$$

$$= \mathcal{C}_{\gamma\delta} \left( A_{\alpha\beta\gamma} |\alpha\rangle \langle\beta| |\gamma\rangle B_{\delta\varepsilon} \langle\delta| \langle\varepsilon| \right)$$

$$= A_{\alpha\beta\gamma}(-1)^{|\gamma|} B_{\gamma\varepsilon} |\alpha\rangle \langle\beta| \langle\varepsilon| .$$

- $\bar{\phantom{A}}$: conjugation of a tensor, takes the complex conjugate of all tensor entries, maps each space to its dual and reverses their order, e.g.

$$\bar{A} = A_{\alpha\beta\gamma}^{\,*} \langle\gamma| |\beta\rangle \langle\alpha| = \quad \tag{108}$$

Diagrammatically, we thus adhere to the convention of mirroring the tensor to maintain counterclockwise ordering.

- $\mathcal{P}$: places a parity tensor on all incoming (bra) legs of a tensor, e.g.

$$\mathcal{P}(A) = (-1)^{|\beta|} A_{\alpha\beta\gamma} |\alpha\rangle \langle\beta| |\gamma\rangle = \quad . \tag{109}$$

The need for the $\mathcal{P}$ operation lies in the fundamental difference between a contraction (evaluation map) and an inner product in our formalism as demonstrated in Eq. (35). As a result, every time we perform or use an algebraic operation involving inner products, we have to be careful and place $P$ tensors to make sure that the tensor network diagram, based on contractions, corresponds to the intended algebraic operation. In particular we mention again inner products, the application of operators (Eq. (43) and Eq. (53)), the definition of unitarity (Eq. (66) and Eq. (69)) and its implications for operations like fusing (e.g. Eq. (73) and Eq. (74)) and flipping legs (Eq. (77)) as well as tensor decompositions based on these concepts like QR and SVD.

The main advantage of our formalism is that once a correct diagram is constructed, tensors and their legs can be moved around freely, even if this results in additional crossings. The fermionic bookkeeping is entirely delegated to internal reorderings and the direction of connected arrows. As a result, a consistent contraction routine can be devised, solely based on the pairs of connected legs and the leg order in input and output tensors. Moreover, this formalism can be integrated with relative ease into existing tensor network libraries as these typically support directed legs already and because fermionic $\mathbb{Z}_2$ symmetry nicely complements other symmetries.

## 3 Fermionic MPS

In this section we explain how to use the formalism of $\mathbb{Z}_2$-graded vector spaces for parametrizing and manipulating finite and infinite fermionic matrix product states (fMPS). We start by introducing the fMPS parametrization of a general 1D fermionic quantum state, after which we discuss the canonical form and how to apply it for computing expectation values. As a first illustration, we outline how the DMRG algorithm [10] for finding ground states of finite fermionic systems fits in this formalism. We then move to the thermodynamic limit, where we introduce the tangent space for the uniform fMPS manifold and discuss how it can be combined with the previous ingredients to arrive at the VUMPS [13] and TDVP [57,58] algorithms for uniform fMPS as well as the quasiparticle excitation ansatz [59,60]. We do not intend to re-derive these concepts in full detail but rather want to highlight where the fermionic nature of the composite tensors induces differences as compared to their bosonic counterparts.

### 3.1 General form

Conceptually, an MPS is introduced by associating two virtual, fermionic degrees of freedom to every site in a 1D chain. In a first step, these virtual modes are arranged in neighboring, maximally entangled states as

$$|\Psi_{\text{virt}}\rangle = \prod_e \left(1 + a^\dagger_{e,l} a^\dagger_{e,r}\right) |\Omega_{\text{virt}}\rangle$$

$$= \sim\!\!\circ\!\circ\!\!\sim\!\!\circ\!\circ\!\!\sim\!\!\circ\!\circ\!\!\sim \tag{110}$$

where the additional $l$ and $r$ labels now discern between the left and right sides of the entangled pairs. Next, we introduce a linear map, projecting the two virtual modes on each site onto a single physical mode,

$$\mathcal{A}_j = \left(A_j\right)_{\alpha_1 \sigma \alpha_2} (a_{j,l})^{\alpha_1} (a^\dagger_j)^\sigma (a_{j,r})^{\alpha_2} . \tag{111}$$

Note the slight alterations in the ordering and normalization compared to Eq. (3) and Eq. (4). This form is functionally equivalent, but will lead to fewer reorderings and scalar factors upon contracting multiple MPS tensors. The map $\mathcal{A}_j$ is such that it is fermion parity preserving, i.e.,

$\left(A_j\right)_{\alpha_1 \sigma \alpha_2} = 0$ if $\sigma + \alpha_1 + \alpha_2 \mod 2 \neq 0$. When projected back onto the virtual vacuum, this yields a fermionic MPS,

$$|\Psi(\mathbf{A})\rangle = \langle \Omega_{\text{virt}}| \prod_j \mathcal{A}_j |\Psi_{\text{virt}}\rangle |\Omega_{\text{phys}}\rangle . \tag{112}$$

In this expression all factors have an even fermion parity. Hence, when they do not contain the same operators, they can be commuted. Therefore, a factor

$$\langle 0| (a_r)^{\alpha_r} (a_l)^{\alpha_l} \left(1 + a_l^\dagger a_r^\dagger\right) |0\rangle = \delta^{\alpha_l \alpha_r} , \tag{113}$$

is obtained for each virtual link in the chain, yielding

$$|\Psi(\mathbf{A})\rangle = \left(\cdots (A_1)_{\alpha_1 \sigma_1 \alpha_2} (A_2)_{\alpha_2 \sigma_2 \alpha_3} \cdots\right) \cdots |\sigma_1\rangle |\sigma_2\rangle \cdots , \tag{114}$$

for an infinite chain. We can reinterpret this matrix product as a virtual contraction,

$$|\Psi(\mathbf{A})\rangle = \mathcal{C}_{\text{virt}} \left( \prod_j \left( (A_j)_{\alpha_j \beta_j \gamma_j} |\alpha_j\rangle |\beta_j\rangle \langle\gamma_j| \right) \right) , \tag{115}$$

where the contracted $\mathbb{Z}_2$-graded MPS tensors are defined and labeled as

$$\alpha \leftarrow \boxed{A_j} \leftarrow \gamma \quad = (A_j)_{\alpha\beta\gamma} |\alpha\rangle |\beta\rangle \langle\gamma| . \tag{116}$$
$$\downarrow \beta$$

For a finite system of length $N$ we can additionally define left and right boundary vectors

$$\boxed{A_1} \leftarrow \gamma \quad = (A_1)_{\beta\gamma} |\beta\rangle \langle\gamma| , \qquad \text{and} \qquad \alpha \leftarrow \boxed{A_N} \quad = (A_N)_{\alpha\beta} |\alpha\rangle |\beta\rangle , \tag{117}$$

yielding a finite fMPS of the form

$$|\Psi(\mathbf{A})\rangle = \boxed{A_1} \leftarrow \boxed{A_2} \leftarrow \cdots \leftarrow \boxed{A_{N-1}} \leftarrow \boxed{A_N} \tag{118}$$

When considering infinite systems with translation invariance, it is often natural to choose the $A_j$ equal on each site, yielding a uniform fMPS with diagrammatic representation

$$|\Psi(A)\rangle = \leftarrow \boxed{A} \leftarrow \boxed{A} \leftarrow \boxed{A} \leftarrow \boxed{A} \leftarrow \boxed{A} \leftarrow \tag{119}$$

As each leg in these diagrams corresponds to a single fermionic mode, both the virtual and physical (dual) vector spaces have dimension two. Evidently, one could introduce more virtual modes per site, say $\chi$, yielding a virtual space with bond dimension $D = 2^\chi$. In the same way the local physical space can be enlarged, containing $f$ fermionic modes and thus having a dimension $d = 2^f$. Finally, one could generalize the MPS construction even further by allowing any natural number for the bond dimension and subdividing the $D$ basis vectors in an even (odd) subset of $D_e(D_o)$ homogeneous basis vectors with $D = D_e + D_o$ [9].

## 3.2 Gauge fixing

As is the case for their bosonic counterparts, fermionic MPS contain gauge freedom, meaning that a transformation

$$\text{—}\boxed{A_j}\text{—} \quad \rightarrow \quad \text{—}(X_{j-1}^{-1})\text{—}\boxed{A_j}\text{—}(X_j)\text{—} \tag{120}$$

leaves $|\Psi(\mathbf{A})\rangle$ invariant for any set of $X_j = (X_j)_\alpha^\beta |\alpha\rangle \langle\beta|$ with all $(X_j)_\alpha^\beta$ invertible.

For infinite systems, one can always use a translation-invariant gauge transform to bring a uniform fMPS $|\Psi(A)\rangle$ in the center-site gauge

$$|\Psi(A)\rangle = \text{—}\boxed{A^l}\text{—}\boxed{A^l}\text{—}\boxed{A^c}\text{—}\boxed{A^r}\text{—}\boxed{A^r}\text{—} \tag{121}$$

e.g. by subsequent QR (LQ) or polar decompositions of $A$, where we define $\{A^l, A^r, C, A^c\}$ by the relations

$$\text{—}\boxed{A^c}\text{—} = \text{—}\boxed{A^l}\text{—}(C)\text{—} = \text{—}(C)\text{—}\boxed{A^r}\text{—} \tag{122}$$

and the left- and right-orthonormality conditions

$$\boxed{A^l} \quad = \quad (I) \quad , \qquad \boxed{A^r}(P) \quad = \quad (P) \tag{123}$$

expressing that $A^l$ is left-isometric from its outgoing to its incoming legs while $A^r$ is right-isometric from its left to its physical and right leg. Indeed, Eq. (123) is equivalent to

$$\text{—}\boxed{\bar{A}^l}\text{—}\boxed{A^l}\text{—} = \text{—}(I)\text{—} , \qquad \text{—}\boxed{A^r}\text{—}\boxed{\bar{A}^r}\text{—}(P) = \text{—}(I)\text{—} \tag{124}$$

showing that when placing the $P$ on the physical leg of $A^r$, the right fixed-point is an identity as well. However, we did not adhere to the left-right conventions of the previous section in Eq. (123) as we intend to homogenize the diagrams for fermionic and the standard bosonic tensor network methods as much as possible. As is the case for bosonic MPS, the singular values of $C$ constitute the entanglement spectrum of the state.

Utilizing the center-site gauge, the expectation value of a local operator, isomorphic to a two-leg tensor $O$, can easily be evaluated as

$$\langle\Psi(A)|O|\Psi(A)\rangle = \boxed{A^l}\boxed{A^l}\boxed{A^c}\boxed{A^r}\boxed{A^r}(O)\boxed{\bar{A}^l}\boxed{\bar{A}^l}\boxed{\bar{A}^c}\boxed{\bar{A}^r}\boxed{\bar{A}^r} = \boxed{A^c}(O)(P)\boxed{\bar{A}^c} , \tag{125}$$

where we mirrored across the horizontal to draw the conjugate MPS tensors. For the norm of a fermionic MPS, we obtain

$$\langle\Psi(A)|\Psi(A)\rangle = \boxed{A^c}(P)\boxed{\bar{A}^c} = (C)(P)(\bar{C}) , \tag{126}$$

such that for a normalized MPS, $A^c$ and $C$ are normalized to unity.

The above discussion is readily applicable to finite systems, where we can now freely move around the location of the center-site within the system by means of the appropriate orthogonal factorizations. In particular, Eqs. (123), (125) and (126) remain valid when taking into account the explicit site-dependence, and we have additional orthogonality conditions for the boundary vectors given by

$$\begin{array}{cc} \boxed{A_1^l} & \\ \Big\downarrow & = \quad I \\ \boxed{\bar{A}_1^l} & \end{array} \quad \text{and} \quad \begin{array}{cc} \boxed{A_N^r} \\ \Big\downarrow \quad = \quad P \\ \boxed{\bar{A}_N^r} \end{array} \,. \tag{127}$$

Furthermore, the entanglement spectrum now depends on where the entanglement cut is performed.

## 3.3 Fermionic DMRG algorithm

Before moving on to a more in-depth discussion of tangent space algorithms for uniform systems, we give a first illustration of the fermionic formalism in the context of a finite system. Suppose we want to find a finite fMPS $|\Psi(\mathbf{A})\rangle$ that minimizes the energy

$$E = \frac{\langle\Psi(\mathbf{A})|H|\Psi(\mathbf{A})\rangle}{\langle\Psi(\mathbf{A})|\Psi(\mathbf{A})\rangle}\,, \tag{128}$$

with respect to some fermionic Hamiltonian $H$.

Here, we will always represent such a 1D Hamiltonian as a Matrix Product Operator (MPO) of the form

$$H = \boxed{O_1} \!\leftarrow\! \boxed{O_2} \!\leftarrow\! \cdots \!\leftarrow\! \boxed{O_{N-1}} \!\leftarrow\! \boxed{O_N} \tag{129}$$

For systems with finite-range or exponentially decaying interactions, one can always find such an MPO representation using a finite virtual bond dimension independent of the system size [61–64]. The local MPO tensor $O_j$ is most naturally represented as a matrix of local operators that is indexed by the virtual levels on its horizontal legs, of the general form

$$\leftarrow\!\boxed{O_j}\!\leftarrow \;=\; \begin{pmatrix} \cdots\!\boxed{\hat{\mathbb{1}}_j}\!\cdots & \cdots\!\boxed{\hat{C}_j}\!\leftarrow & \cdots\!\boxed{\hat{D}_j}\!\cdots \\[2mm] & \cdots\!\boxed{\hat{A}_j}\!\leftarrow & \leftarrow\!\boxed{\hat{B}_j}\!\cdots \\[2mm] & & \cdots\!\boxed{\hat{\mathbb{1}}_j}\!\cdots \end{pmatrix}\,, \tag{130}$$

where the dashed lines are trivial. Upon contraction, these MPO tensors can generate any 1D interaction, where each local tensor can be interpreted as a finite state machine that dictates how the different terms in the original Hamiltonian are passed through that site. Note that the individual entries of each local MPO can have non-trivial virtual indices themselves, which is essential in order to correctly handle the internal symmetries of the Hamiltonian terms. This can for example be seen from the specific form of the symmetric creation operator defined in Eq. (98), and will be discussed in more detail for the examples in Sec. 4.1 and Sec. 4.2. For a more general discussion on how to construct MPO Hamiltonians we again refer to Refs. [61–64].

Given an MPO Hamiltonian, the ground state search problem can be interpreted as the task of trying to find the fMPS eigenvector of $H$ with the smallest real eigenvalue $E$. The DMRG algorithm [10] gives an iterative procedure for finding an fMPS eigenvector that satisfies

$$\tag{131}$$

At each iteration the algorithm proceeds by gauging the state around a site $j$, performing a local update on the fMPS tensors at site $j$ and inserting these back into the state, after which the gauge center is moved to a next site. Repeating this procedure while sweeping $j$ back and forth across the system eventually gives the desired approximate eigenstate with minimal energy eigenvalue.

In the two-site formulation of the algorithm, the local update proceeds by replacing the MPS tensor at site $j$ and its neighbor to the right by $X_j$, yielding

$$|\Psi(\mathbf{A}, X_j)\rangle = \tag{132}$$

Here, $X_j$ is determined by minimizing the resulting energy

$$E(X_j) = \frac{\langle \Psi(\mathbf{A}, X_j)|H|\Psi(\mathbf{A}, X_j)\rangle}{\langle \Psi(\mathbf{A}, X_j)|\Psi(\mathbf{A}, X_j)\rangle}, \tag{133}$$

where

$$\langle \Psi(\mathbf{A}, X_j)|H|\Psi(\mathbf{A}, X_j)\rangle = \tag{134}$$

with the environments $G_j^l$ and $G_j^r$ defined in the usual way as

$$\text{and} \tag{135}$$

and boundary environments given by

$$\text{and} \tag{136}$$

The norm on the other hand can be expressed as

$$\langle \Psi(\mathbf{A}, X_j)|\Psi(\mathbf{A}, X_j)\rangle = \tag{137}$$

Differentiating $E(X_j)$ w.r.t. $\bar{X}_j$ and setting the result equal to zero we find $X_j$ to be the eigenvector with smallest real eigenvalue of a local effective Hamiltonian which acts on a tensor $X_j$ as

$$\tag{138}$$

Note the presence of the extra $P$ tensor on the right-most open leg, which is the only difference compared to the usual bosonic formulation of the algorithm. All other fermionic behavior is dealt with by using oriented legs and a corresponding, consistent contraction routine.

The local eigenvalue problem of Eq. (138) is seeded with the MPS tensors at the end of the previous update,

$$\tag{139}$$

After finding the local fixed point $X_j'$, this tensor is decomposed and truncated into a new set of local fMPS tensors of a desired bond dimension using the fermionic SVD (Eq. (91)),

$$\tag{140}$$

after which $A_j^{c}{}'$ and $A_{j+1}^{r}{}'$ are inserted back into the state. We can then move the center gauge to the next site, and repeat this in a sweeping pattern until the algorithm converges.

## 3.4 Tangent space

Next we move to the thermodynamic limit, and start by considering the manifold of uniform fMPS more carefully. Differentiating a uniform fMPS $|\Psi(A)\rangle$ w.r.t. the entries in $A$ and taking superpositions of the result, one obtains a general tangent vector for the MPS manifold in $|\Psi(A)\rangle$,

$$|\Phi(B;A)\rangle = \sum_j \tag{141}$$

parametrized by the tensor $B$. Gauge transformations can be performed and absorbed in $B$ so that a general tangent vector can equally well be expressed as

$$|\Phi(B;A)\rangle = \sum_j \tag{142}$$

Moreover, the multiplicative gauge invariance for $A$ translates to an additive gauge invariance for $B$,

$$\tag{143}$$

We can remove this gauge invariance by defining canonical forms for $|\Phi(B;A)\rangle$, e.g. by imposing

$$0 = \quad (144)$$

However, one can only impose this condition when the tangent vector is orthogonal to the MPS as it implies that

$$\langle \Psi(A)|\Phi(B;A)\rangle = |\mathbb{Z}| \quad = 0. \quad (145)$$

In the following we will see that, conveniently, these are the most relevant tangent vectors so that we can safely work with this gauge condition, implying that $B$ is an element of the right null-space of $\bar{A}^l$ and can thus be written as

$$\quad = \quad (146)$$

Here, $V^l$ describes an basis for the null-space and therefore has dimension $(d-1)D$ on its incoming leg. $X$ (i.e., a different $X$ from the one we used before), on the other hand, essentially encodes a linear combination of these basis elements and thus contains the only free parameters in $B$. Furthermore, we orthonormalize $V^l$ in the same way as $A^l$ (Eq. (123)) so that

$$\quad + \quad = \quad (147)$$

due to completeness. Analogously, a right-canonical form can be derived,

$$\quad = \quad (148)$$

where $V^r$ encodes a basis for the left null-space of $\bar{A}^r$ and is normalized in the same way as $A^r$. When brought in this right-canonical form,

$$\quad = 0. \quad (149)$$

An important feature of the canonical form for tangent vectors is that the overlap of two such vectors can easily be computed as

$$\langle \Phi(B_2;A)|\Phi(B_1;A)\rangle = |\mathbb{Z}| \quad = |\mathbb{Z}| \quad , \quad (150)$$

whenever $B_1$ and $B_2$ are both in the left, or both in the right, canonical form.

## 3.5 Fermionic VUMPS and TDVP algorithms

Tangent vectors can be utilized to derive the tangent space projector $P_{|\Psi(A)\rangle}$ in (and orthogonal to) a point, $|\Psi(A)\rangle$, on the MPS manifold, as explained in Appendix A.1. This operator plays a key role in establishing both the VUMPS and TDVP algorithms. For the former, application of the variational principle to the MPS manifold to find the $|\Psi(A)\rangle$ with the lowest energy for a Hamiltonian $H$, requires the minimisation of the energy expectation value,

$$E = \frac{\langle \Psi(A)|H|\Psi(A)\rangle}{\langle \Psi(A)|\Psi(A)\rangle} \,, \tag{151}$$

w.r.t. the variational parameters in $|\Psi(A)\rangle$. The variatonal minimim is characterised by differentiating w.r.t. the (complex conjugates of) the variational parameters and give rise to the condition

$$P_{|\Psi(A)\rangle}(H - E)|\Psi(A)\rangle = 0 \,, \tag{152}$$

where the second term drops out as $P_{|\Psi(A)\rangle}$ projects orthogonally to $|\Psi(A)\rangle$. Note that this is not problematic as the equation is automatically satisfied in this direction. Indeed, $E$ is exactly defined so that $\langle \Psi(A)|(H-E)|\Psi(A)\rangle = 0$. However, this term is typically retained in order to fix divergences due to the extensive nature of $H$. For TDVP, assuming that an MPS state remains on the MPS manifold through time-evolution and can therefore be described with a time-dependent tensor $A(t)$, substitution in the time-dependent Schrödinger equation yields

$$|\Phi(\dot{A};A)\rangle = -iH|\Psi(A)\rangle \,. \tag{153}$$

Here, the left-hand side is a tangent vector of the MPS manifold in $|\Psi(A)\rangle$ while the right-hand side is not. Nonetheless, one can approximate the latter by projecting it down on the tangent space, yielding the TDVP equation,

$$|\Phi(\dot{A};A)\rangle = -iP_{|\Psi(A)\rangle}H|\Psi(A)\rangle \,. \tag{154}$$

Again the lack of support for $P_{|\Psi(A)\rangle}$ in the direction of $|\Psi(A)\rangle$ does not pose a problem as we do not want to change its norm or phase.

In the following we again represent $H$ as a, now uniform, MPO

$$H = \begin{array}{c}\text{---}\boxed{O}\text{---}\boxed{O}\text{---}\boxed{O}\text{---}\boxed{O}\text{---}\boxed{O}\text{---}\end{array} \tag{155}$$

with a local tensor $O$ of the form in Eq. (130). Making use of this MPO form we can rewrite the quantity $P_{|\Psi(A)\rangle}H|\Psi(A)\rangle$, which plays an essential role in both the VUMPS and TDVP algorithms. As explained in Appendices A.1 and A.2, for a Hamiltonian of the form of Eq. (155), this tangent vector can be written in the form of Eq. (142) with either

$$\boxed{B} \;=\; \boxed{A^{c\prime}} \;-\; \boxed{A^l}\text{---}\boxed{C'} \tag{156}$$

i.e., the left gauge, or

$$\boxed{B} \;=\; \boxed{A^{c\prime}} \;-\; \boxed{C'}\text{---}\boxed{A^r} \tag{157}$$

the right gauge, where

$$
\text{[diagram]} = \lambda^{N_l+N_r-1} \text{[diagram]} = E \text{[diagram]} \tag{158}
$$

and

$$
\text{[diagram]} = \lambda^{N_l+N_r} \text{[diagram]} = E \text{[diagram]} \tag{159}
$$

and where both choices for $B$ are related by an additive gauge transform with $X = C'$. Here, $N_l$ $(N_r)$ is the number of sites where $A$ was left- (right-)canonicalized and the uniform environments $G^l$ and $G^r$ are given by the dominant left and right eigenvectors of the MPS transfer matrices as

$$
\text{[diagram]} = \lambda \text{[diagram]} \quad \text{and} \quad \text{[diagram]} = \lambda \text{[diagram]} \tag{160}
$$

Note that these two eigenvalues have to be identical due to the canonicalization constraints. The determination of the environments is generally an eigenvalue problem, but it simplifies to a linear problem when $H$ is (quasi-)local and can be represented as an MPO in Schur form [13,62,63]. Of course, the right hand side of Eqs. (158) and (159) is in general divergent since both $N_l$ and $N_r$ are infinite here. However, it is clear that if we normalize the environments as

$$
\text{[diagram]} = 1, \tag{161}
$$

$E = \lambda^{N_l+N_r} = \lambda^{|\mathbb{Z}|}$ so that $E$ is either infinite when $\lambda > 1$, zero when $\lambda < 1$ or 1 when $\lambda = 1$. Furthermore, the dominant eigenvalue of the $A^c$ eigenvalue problem equals $\lambda$ while for the $C$ eigenvalue problem we find eigenvalue 1. In case of a (quasi-)local Hamiltonian, the finite energy density can also be extracted from the linear problem for the environments by taking into account the Schur form of $O$ [13]. Important is that this calculation requires the left and right fixed-points of the MPS transfer matrix (without an MPO tensor in between). Using the center-site gauge, these are $I$ and $P$ as opposed to only $I$ in the bosonic case.

Once computed, $A^{c'}$ and $C'$ and hence also the expression for $P_{|\Psi(A)\rangle}H|\Psi(A)\rangle$ in terms of a $B$ tensor can be used to express the TDVP equation as $\dot{A} = -iB$. For instance combining this with the Euler method, one obtains $A(t+dt) = A(t) - i\,dt\,\dot{A}(t)$. After gauge fixing and normalization by rescaling the new $A(t+dt)$, this can be repeated for further time steps. Alternatively, more

involved time-evolution methods can be utilized. In the VUMPS algorithm, on the other hand, we want that $|\Phi(B;A)\rangle = P_{|\Psi(A)\rangle} H |\Psi(A)\rangle = 0$ or thus $B = 0$ so that

$$-\boxed{A^{c\prime}}\leftarrow \;=\; -(C')\leftarrow\boxed{A^r}\leftarrow \;=\; -\boxed{A^l}\leftarrow(C')\leftarrow \qquad (162)$$

while Eq. (122) also still applies. As the gauge transformation that relates $A^l$ and $A^r$ is unique up to a factor (for injective MPS), this implies that $C$ and $C'$ have to be proportional, resulting in the VUMPS fixed-point equations, $A^{c\prime} \propto A^c$ and $C' \propto C$, that together with the gauge-fixing conditions describe the variational minimum. After solving these, one still has to perform an SVD or polar decomposition to derive new $A^l$ and $A^r$ in line with the obtained $A^c$ and $C$ [13]. Afterwards, the same steps can be repeated until a variational minimum is obtained. More variations and extensions of this algorithm exist, e.g. the integration of unit cells [65], but will not be considered here.

## 3.6  Quasiparticle excitations

The tangent space described by Eq. (142) was found by taking derivatives of the translation-invariant MPS Ansatz w.r.t. its parameters. However, a more general tangent space can be obtained by starting from a fermionic MPS with site-dependent tensors,

$$|\Psi(\boldsymbol{A})\rangle = -\boxed{A_{j-1}}\leftarrow\boxed{A_j}\leftarrow\boxed{A_{j+1}}\leftarrow \qquad (163)$$

and differentiating it w.r.t. its parameters. In this way, one obtains the general tangent vector

$$|\Phi(\boldsymbol{B};\boldsymbol{A})\rangle = \sum_j \; -\boxed{A_{j-1}}\leftarrow\boxed{B_j}\leftarrow\boxed{A_{j+1}}\leftarrow \qquad (164)$$

with a site-dependent $B_j$ tensor which can be used to describe non-translation-invariant excitations on top of an MPS. However, when both the Hamiltonian and its ground state are translation invariant, we can label the excited states by their momentum $p$ and again take the $A$ tensors to be identical. The momentum $p$ sector of the tangent space can then be targeted by setting $B_j = e^{ipj}B$ so that

$$|\Phi_p(B;A)\rangle = \sum_j e^{ipj} -\boxed{A}\leftarrow\boxed{B}\leftarrow\boxed{A}\leftarrow \qquad (165)$$

These states can be regarded as elementary excitations on top of uniform fermionic MPS that are, in the spirit of the single-mode approximation, local perturbations in a momentum superposition. As was the case for the $p = 0$ tangent vectors in Eq. (142), we can absorb gauge transformations for the $A$ tensors in $B$ so that

$$|\Phi_p(B;A)\rangle = \sum_j e^{ipj} -\boxed{A^l}\leftarrow\boxed{B}\leftarrow\boxed{A^r}\leftarrow \qquad (166)$$

This leads to an additive gauge transformation,



$$\text{(diagram)} \rightarrow \text{(diagram)} + \text{(diagram)} - e^{ip} \text{(diagram)} , \tag{167}$$

that can again be removed by imposing either Eq. (144) or Eq. (149), i.e., the left, respectively right, gauge condition for $|\Phi_p(B;A)\rangle$, resulting in identical canonical forms for the $B$ tensor as in the $p = 0$ case (Eq. (146) and Eq. (148)). For the overlap between two $|\Phi_p(B;A)\rangle$ states, one obtains

$$\langle\Phi_q(B_2;A)|\Phi_p(B_1;A)\rangle = \sum_j e^{i(p-q)j} \text{(diagram)}$$
$$= 2\pi\delta(p-q)\text{(diagram)} = 2\pi\delta(p-q)\text{(diagram)} , \tag{168}$$

where we used that only terms with $B$ tensors on the same sites are non-zero due to the gauge condition.

In order to calculate $B$ tensors corresponding to the elementary excitations of a certain Hamiltonian, we apply the variational principle to $|\Phi_p(B;A)\rangle$. Therefore, we differentiate the momentum-dependent excitation energy,

$$\omega = \frac{\langle\Phi_p(B;A)|(H-E)|\Phi_p(B;A)\rangle}{\langle\Phi_p(B;A)|\Phi_p(B;A)\rangle} , \tag{169}$$

w.r.t. the complex conjugate of the parameters in $B$, i.e., $X$ from Eq. (146) or Eq. (148). As $|\Phi_p(B;A)\rangle$ is linear in $X$ (and $B$), the numerator and denominator are quadratic in these tensors, resulting in a generalized eigenvalue problem after differentiation,

$$\frac{\partial}{\partial\overline{X}}\Big(\langle\Phi_p(B;A)|\Big)(H-E)|\Phi_p(B;A)\rangle = \omega\frac{\partial}{\partial\overline{X}}\Big(\langle\Phi_p(B;A)|\Big)|\Phi_p(B;A)\rangle , \tag{170}$$

with the same holding for differentiation w.r.t. (the conjugate of) $B$. Enforcing the left or right gauge, this reduces to an ordinary eigenvalue problem as the right-hand side of the equation becomes particularly easy to evaluate.

As an illustration, we work out the eigenvalue equation for $B$,

$$\sum_{jj'} e^{ip(j-j')} \text{(diagram } H-E\text{)} = \omega|\mathbb{Z}| \text{(diagram } B\text{)} \tag{171}$$

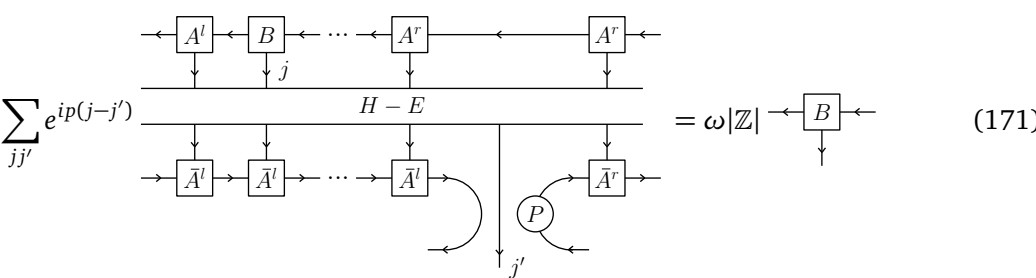

for a Hamiltonian with an MPO representation. Invoking translation invariance, we obtain a single sum over the separation $n = j' - j - 1$ between the $j$ and $j'$ indices, multiplied by $|\mathbb{Z}|$ so that this factor drops out on both sides. Finally, we make use of the fact that $H$ is an MPO as

in Eq. (155) to find

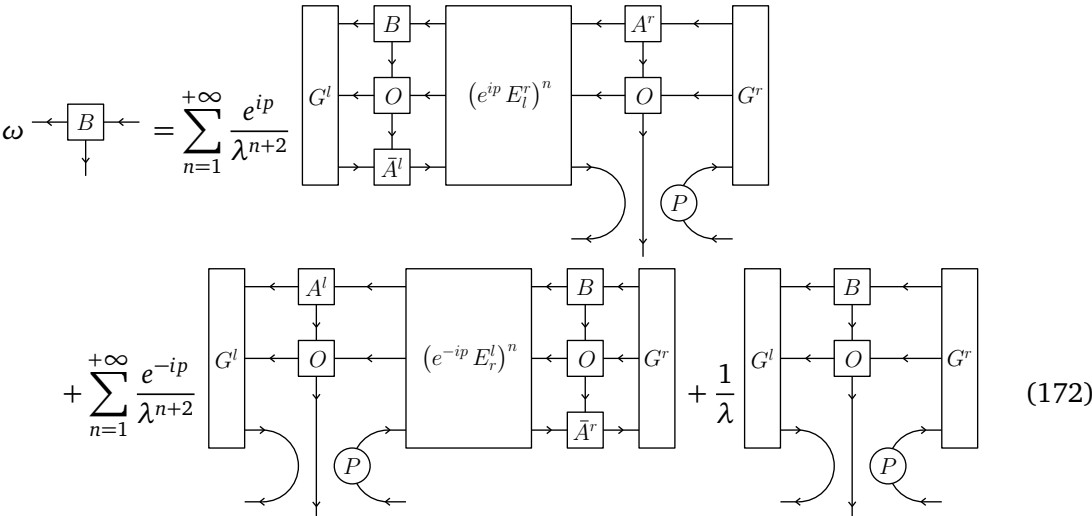

$$\omega \,\, \Big[ B \Big] \,\, = \sum_{n=1}^{+\infty} \frac{e^{ip}}{\lambda^{n+2}} \left[ \cdots \right] + \sum_{n=1}^{+\infty} \frac{e^{-ip}}{\lambda^{n+2}} \left[ \cdots \right] + \frac{1}{\lambda} \left[ \cdots \right] \tag{172}$$

Here, $G^l$ and $G^r$ are the left and right environments, i.e., the dominant eigenvectors of $E_l^l$, respectively $E_r^r$ with $\lambda = 1$ their common eigenvalue (see Appendix A.2). Furthermore, $E_r^l$ and $E_l^r$ are the mixed transfer matrices,

$$E_r^l = \begin{bmatrix} A^l \\ O \\ \bar{A}^r \end{bmatrix} \quad \text{and} \quad E_l^r = \begin{bmatrix} A^r \\ O \\ \bar{A}^l \end{bmatrix} \tag{173}$$

The geometric sums hereof in Eq. (172) can be expressed as a function of their respective dominant eigenvectors so that the eigenvalue equation can be solved for the excitation spectrum. Though this yields $D^2(d-1)$ solutions, only those corresponding to the smallest eigenvalues have physical meaning. Indeed, for a given value of the momentum, one typically finds a limited number of excitations living on an isolated branch in the spectrum. All the other solutions are organized in continuous bands corresponding to scattering states. It is not expected that all of these states are approximated well with the quasiparticle Ansatz. We also note that we did not take into account additional symmetries here. However, in doing so it can become necessary for $B$ to have a non-trivial charge in order to target the lowest excitations. This is typically done by giving $B$ an auxiliary leg with a non-trivial charge (e.g. see Sec. 4.1).

For the case of finite systems, the quasiparticle excitations can be computed in much the same way [66]. In this case momentum is not a conserved quantity, so the quasiparticle Ansatz consists of a general tangent vector of the form Eq. (164) which similarly admits a left and right gauge of the form Eq. (144) or Eq. (149) respectively. Differentiating the excitation energy Eq. (169) with respect to the variational parameters of the local excitation tensors and imposing the appropriate gauge constraints again yields an ordinary eigenvalue equation. This problem is even more straightforward to solve than in the infinite case, since evaluating the action of the effective Hamiltonian in a finite system does not require any elaborate resummation schemes.

## 4 Applications in 1D

We benchmark the fMPS methods by reproducing analytic results on the ground state energy and lowest excited states for the 1D Hubbard model and the Kitaev chain. The former, being

integrable admits an exact solution with the Bethe Ansatz while the latter is quadratic and can thus be solved by a Bogoliubov transformation. Not only can fMPS reproduce the energy densities quantitatively, qualitative features like spin-charge separation and gapless edge modes can be resolved as well.

## 4.1 Spin-charge separation in the Hubbard model

Consider the fermionic Hubbard model on an infinite 1D chain,

$$H = -t \sum_{\langle i,j \rangle} \sum_{\sigma} \left( a_{i,\sigma}^{\dagger} a_{j,\sigma} + \text{h.c.} \right) + U \sum_{i} n_{i,\uparrow} n_{i,\downarrow}, \tag{174}$$

at half-filling and for $U/t = 1, 3, 10$. This interacting model can be solved exactly using the Bethe Ansatz, making it an ideal benchmark case. Additionally, MPS techniques have already been applied to this model, albeit by first using a Jordan-Wigner transformation to transform the fermions to spins [67]. In Sec. 2.8 we have already laid out how to represent spin-1/2 fermions in our language by combining the fermionic grading with the SU(2) symmetry. Here, we extend this by also including the U(1) charge conservation. Using fermionic MPS with this $f\mathbb{Z}_2 \otimes \text{SU}(2) \otimes \text{U}(1)$ structure, we can approximate the ground state in an optimal way.

To impose half filling by we shift the U(1) charges down by 1, and then make use of the fact that the algorithms will automatically optimize in the globally trivial sector with shifted U(1) charge zero. Consequently, the local Hilbert space on each site will be comprised of the following states:

$$\begin{array}{c|c|c|c}
 & f\mathbb{Z}_2 & \text{SU}(2) & \text{shifted U}(1) \\
\hline
|0\rangle & 0 & 0 & -1 \\
|\uparrow\downarrow\rangle & 0 & 0 & 1 \\
\{|\uparrow\rangle, |\downarrow\rangle\} & 1 & 1/2 & 0
\end{array} \tag{175}$$

The sectors for each virtual leg can be split into two groups, which do not mix due to the $f\mathbb{Z}_2$ grading of the physical charges, which is also respected by the SU(2) sectors (integer and half-integer) and U(1) sectors (even and odd). As no symmetries can be broken due to the Mermin-Wagner-Hohenberg-Coleman theorem [68–70], the resulting MPS has to realize a two-site pattern alternating between two sets of charges for the virtual legs. Indeed, the left virtual spaces on even sites, for instance, contain two sets of irreps

$$\begin{array}{|c|c|c|}
\hline
f\mathbb{Z}_2 & \text{SU}(2) & \text{shifted U}(1) \\
\hline
0 & 0 & \pm 1 \\
0 & 1 & \pm 1 \\
0 & 0 & \pm 3 \\
0 & 1 & \pm 3 \\
\vdots & \vdots & \vdots \\
\hline
\end{array}
\qquad
\begin{array}{|c|c|c|}
\hline
f\mathbb{Z}_2 & \text{SU}(2) & \text{shifted U}(1) \\
\hline
1 & 1/2 & 0 \\
1 & 3/2 & 0 \\
1 & 1/2 & \pm 2 \\
1 & 3/2 & \pm 2 \\
\vdots & \vdots & \vdots \\
\hline
\end{array} \tag{176}$$

whereas on the odd sites we have

$$\begin{array}{|c|c|c|}
\hline
f\mathbb{Z}_2 & \text{SU}(2) & \text{shifted U}(1) \\
\hline
0 & 0 & 0 \\
0 & 1 & 0 \\
0 & 0 & \pm 2 \\
0 & 1 & \pm 2 \\
\vdots & \vdots & \vdots \\
\hline
\end{array}
\qquad
\begin{array}{|c|c|c|}
\hline
f\mathbb{Z}_2 & \text{SU}(2) & \text{shifted U}(1) \\
\hline
1 & 1/2 & \pm 1 \\
1 & 3/2 & \pm 1 \\
1 & 1/2 & \pm 3 \\
1 & 3/2 & \pm 3 \\
\vdots & \vdots & \vdots \\
\hline
\end{array} \tag{177}$$

The MPS will thus have a unit cell of two tensors,

$$
|\Psi(A_1, A_2)\rangle = \quad\overset{c_1}{\longleftarrow}\boxed{A_1}\overset{c_2}{\longleftarrow}\boxed{A_2}\overset{c_1}{\longleftarrow}\boxed{A_1}\overset{c_2}{\longleftarrow}\boxed{A_2}\overset{c_1}{\longleftarrow}
$$
$$
\quad (0,\tfrac{1}{2},0)\quad (0,\tfrac{1}{2},0)\quad (0,\tfrac{1}{2},0)\quad (0,\tfrac{1}{2},0)
$$
(178)

where the virtual charges alternate between these two sets, labeled by $c_1$ and $c_2$.

Starting from the tensor representations of the creation and annihilation operators in Sec. 2.8 (but now supplemented with their corresponding U(1) charge sectors), we construct the MPO representation of $H$ with the local MPO tensor

$$
\boxed{O_j} = \begin{pmatrix} 1 & -ta_j^\dagger & ta_j & Un_{j,\uparrow}n_{j,\downarrow} \\ 0 & 0 & 0 & a_j \\ 0 & 0 & 0 & a_j^\dagger \\ 0 & 0 & 0 & 1 \end{pmatrix}.
$$
(179)

In particular, the operators within this matrix are of the form introduced in Eq. (100) and Eq. (102). Note however that special care must be taken to pair up creation and annihilation operators with the right virtual arrows, i.e., creation and annihilation operators in the first row for instance have an incoming auxiliary leg on the right, while their counterparts in the last column need to have a flipped outgoing auxiliary leg on the left. Crucial is that their entries should be such that, when combined, they yield the correct two-body terms appearing in the Hamiltonian. Therefore, one could equally well first build an SU(2) symmetric hopping term from the basic operators in Eq. (100) and Eq. (102),

$$
\boxed{\quad h \quad} = \boxed{a^\dagger}\!\!\sim\!\!\boxed{a}
$$
(180)

sum it with its Hermitian conjugate and then apply an SVD to this two-body operator to split it in a left and right part,

$$
\boxed{\quad h \quad} + \boxed{\quad \bar{h} \quad} = \boxed{L}\longleftarrow\boxed{R}
$$
(181)

Note that these parts only have a non-trivial, bond dimension 4, virtual leg on the right (left) side. For the MPO, we then obtain the more compact form

$$
\boxed{O_j} = \begin{pmatrix} 1 & L_j & Un_{j,\uparrow}n_{j,\downarrow} \\ 0 & 0 & R_j \\ 0 & 0 & 1 \end{pmatrix}.
$$
(182)

The on-site interacting Hubbard term can easily be constructed in the local basis and has trivial auxiliary legs on both sides.

Combining all of the above, we obtain fMPS representations for the ground state directly in the thermodynamic limit, using the VUMPS algorithm as described in Sec. 3.5. The energy densities obtained in this way can then be compared to the exact Bethe Ansatz results [71]. Even with relatively low bond dimensions, in correspondence to Schmidt values down to $10^{-3}$, we obtain an agreement in the ground state energy density with over three digits of accuracy for all three parameter choices. The corresponding entanglement spectra are displayed in Fig. 1 for entanglement cuts to the left of $A_1$ and $A_2$, with the difference in the two sets of virtual charges.

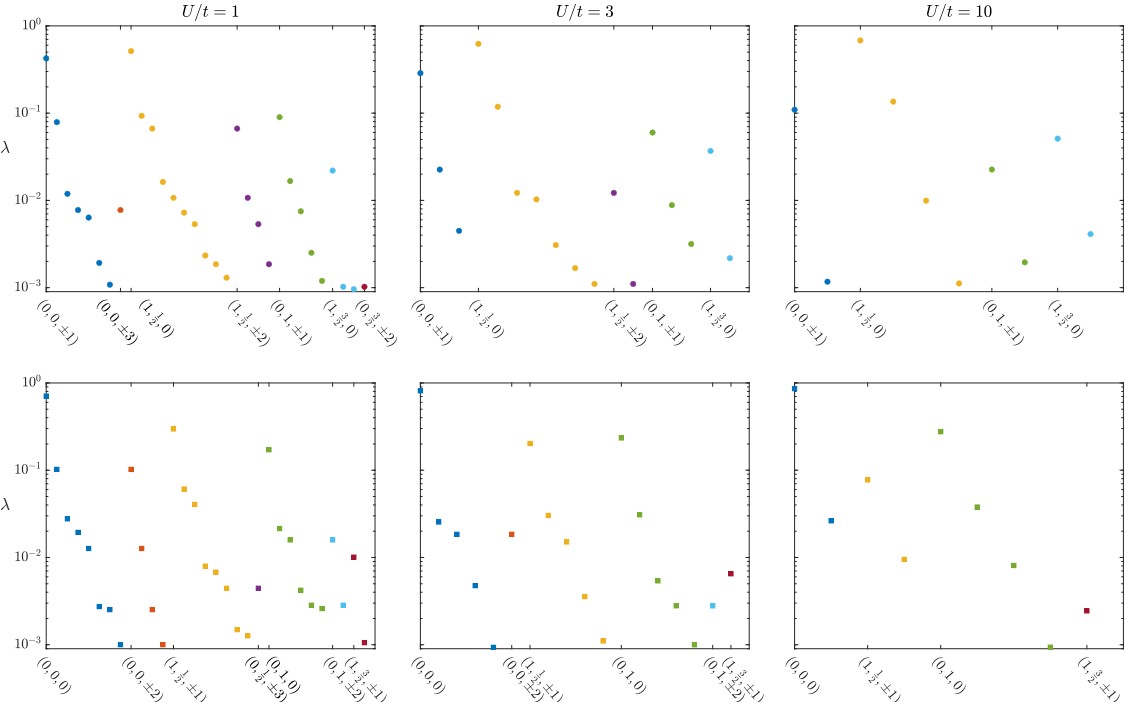

Figure 1: Symmetry-resolved entanglement spectra for the fMPS approximation of ground state of the 1D Hubbard model on an infinite chain at different values of $U/t$. Entanglement cuts are performed to the left of $A_1$ (top row) and $A_2$ (bottom row), yielding virtual charges $c_1$ (dots) and $c_2$ (squares) in the two aforementioned separate sets.

As is well known, for any finite value of $U/t$ the 1-D Hubbard model is a Mott insulator at half filling. As $U/t$ increases, we expect that charge fluctuations in the ground state will decrease; in the limit of large $U$, the charge fluctuations in the ground state will be completely frozen and the resulting spin state will reduce to the ground state of the spin-1/2 Heisenberg model. This trend is seen in the entanglement spectra in Fig. 1, as the weight of the non-trivial U(1) charge sectors are suppressed for large $U/t$, whereas the non-trivial SU(2) sectors with zero charge stay more or less constant.

In order to obtain quasiparticle excitations, multiple options arise as a result of the two-site pattern of the ground state. The $B$ tensors can follow this pattern,

$$|\Phi_p^k(B_1, B_2)\rangle = \sum_{m \text{ even}} e^{ipm}\Bigg( \underset{\substack{m-2 \quad m-1 \quad m \quad m+1 \quad m+2 \quad m+3}}{\overset{k}{\cdots A_1 \leftarrow A_2 \leftarrow B_1 \leftarrow A_2 \leftarrow A_1 \leftarrow A_2 \cdots}} $$

$$+ \underset{\substack{m-2 \quad m-1 \quad m \quad m+1 \quad m+2 \quad m+3}}{\overset{k}{\cdots A_1 \leftarrow A_2 \leftarrow A_1 \leftarrow B_2 \leftarrow A_1 \leftarrow A_2 \cdots}} \Bigg), \tag{183}$$

which requires that the irreps for its auxiliary leg are given by $(0,0,0),(1,\frac{1}{2},\pm 1),\ldots$, i.e., a trivial excitation, an electronic excitation, etc. These are all "physical" excitations in the sense that they have non-zero overlaps with physical operators (i.e., operators that are made by combinations of fermionic creation and annihilation operators) acting on the ground state.

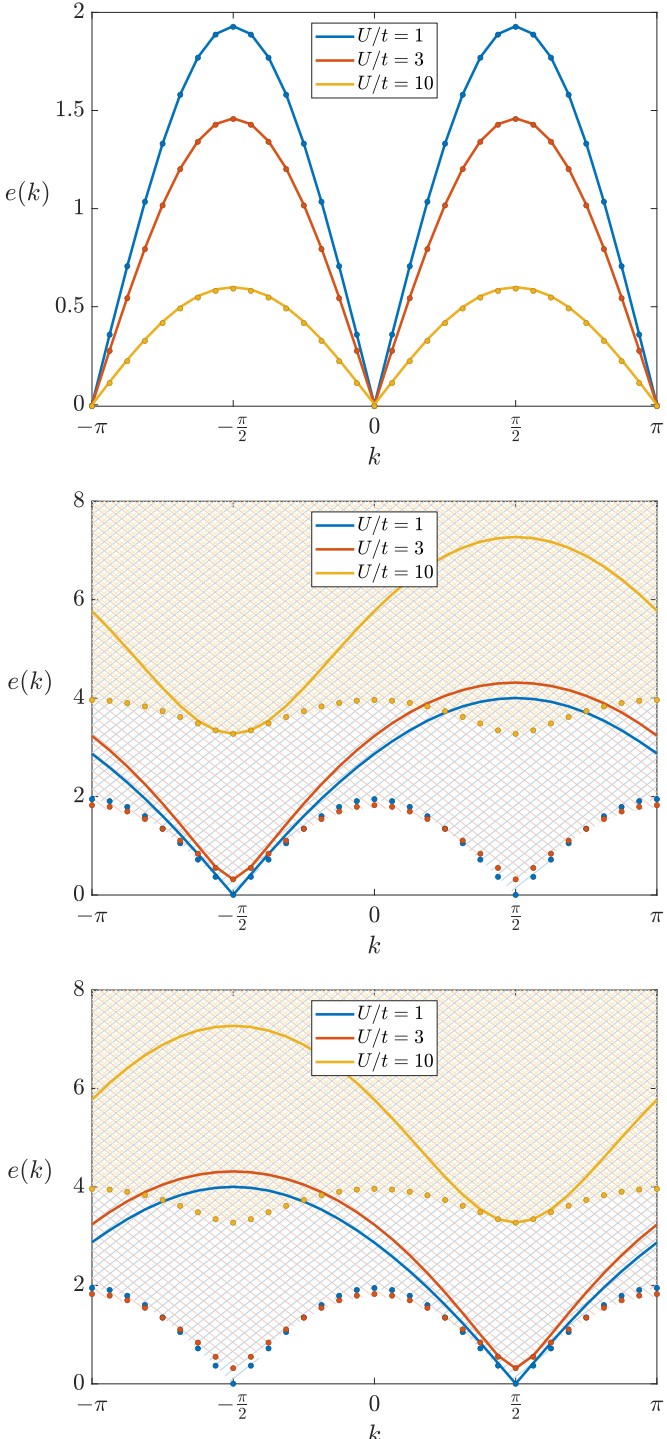

Figure 2: Dispersion relations for the infinite 1D Hubbard model at half-filling for different values of $U/t$. Solid lines correspond to the exact Bethe Ansatz dispersion relations [71] while circular markers display results obtained with the excitation Ansatz for spinons(top), holons (middle) and anti-holons (bottom). The exact bond dimensions used for these calculations are $D = 32$ for $U/t = 10$, $D = 55$ for $U/t = 3$ and $D = 116$ for $U/t = 1$.

Alternatively, the $B$ tensors can disrupt the two-site pattern,

$$|\Xi_p^k(B)\rangle = \sum_{m \text{ even}} e^{ipm} \quad (184)$$

effectively creating domain wall excitations between the two equivalent ground state possibilities. In this case, the allowed irreps are $(1, 0, \pm 1)$, $(0, \frac{1}{2}, 0)$, ... The former correspond to charged particles without spin that are either a U(1) charge higher or lower than single occupancy, i.e., a spinless hole-like or electron-like excitation. These are the so-called holons and anti-holons (also called chargons). The second possibility corresponds to a neutral particle with a spin, i.e., a spinon. These topological excitations are the elementary excitations in this system, demonstrating the principle of spin-charge separation.

In Fig. 2 we compare the dispersion relations obtained via the quasiparticle Ansatz (filled circles) with those that can be computed using the Bethe Ansatz [71] (full lines). For the spinon excitations we immediately have perfect agreement between numerical and analytical results, while for the (anti-)holon excitations we can only reproduce the low energy region of the analytical results, around momentum $\pm \frac{\pi}{2}$, while for other momenta we obtain states that are lower in energy than the predicted analytical results. This can be explained as the quasiparticle Ansatz just targets the lowest energy states, which need not be elementary. If we also consider scattering states, we can create a continuum of states with total momentum and energy given by the sum of the single-particle states. In particular, the combination of two spinons and a single (anti-)holon leads to an energy band as shown by the shaded regions in Fig. 2, and we see that the quasiparticle Ansatz agrees perfectly with their lower bounds.

## 4.2 Gapless edge modes in the Kitaev chain

Consider the 1D Kitaev chain of length $N$,

$$H = -t \sum_{i=1}^{N-1} \left( a_i^\dagger a_{i+1} + \text{h.c.} \right) - \mu \sum_{i=1}^{N} a_i^\dagger a_i - \Delta \sum_{i=1}^{N-1} \left( a_i^\dagger a_{i+1}^\dagger + \text{h.c.} \right). \quad (185)$$

This Hamiltonian can be rewritten as

$$H = \frac{1}{2} \sum \Upsilon^\dagger H_{\text{BdG}} \Upsilon + E, \quad (186)$$

with $E$ an energy offset, $H_{\text{BdG}}$ the Bogoliubov-de Gennes Hamiltonian and with the creation and annihilation operators gathered in the Nambu spinor, $\Upsilon = \left( a_1 \cdots a_N \quad a_1^\dagger \cdots a_N^\dagger \right)^T$. $\Upsilon$ and $\Upsilon^\dagger$ contain the same operators so that

$$(\sigma^x \otimes \mathbb{1}_N) H_{\text{BdG}}^T (\sigma^x \otimes \mathbb{1}_N) = -H_{\text{BdG}}. \quad (187)$$

Consequently, $H_{\text{BdG}}$ has the following substructure

$$H_{\text{BdG}} = \begin{pmatrix} \Xi & \Delta \\ -\overline{\Delta} & -\Xi^T \end{pmatrix}, \quad (188)$$

with the $N \times N$ circulant parameter matrices given by

$$\Xi = \Xi^\dagger = - \begin{pmatrix} \mu & t & & & \\ t & \mu & t & & \\ & t & \mu & \ddots & \\ & & \ddots & \ddots & t \\ & & & t & \mu \end{pmatrix}, \quad \Delta = -\Delta^T = - \begin{pmatrix} 0 & -\Delta & & & \\ \Delta & 0 & -\Delta & & \\ & \Delta & 0 & \ddots & \\ & & \ddots & \ddots & -\Delta \\ & & & \Delta & 0 \end{pmatrix}. \quad (189)$$

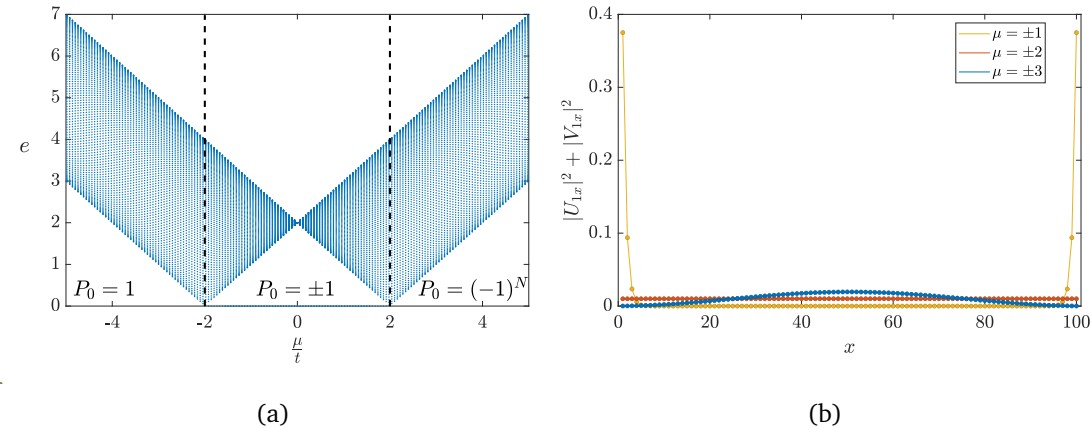

Figure 3: (a) Single particle energy spectra and ground state parity $P_0$ as a function of $\mu/t$. (b) Expansion coefficients for the Bogoliubov mode with the lowest energy. Full lines are used for theoretical values, filled circles for the fMPS results. The results in both panels are obtained for a chain of length $N = 100$ with $\Delta/t = 1$.

Furthermore, the energy offset is $E = -\mu\frac{N}{2}$. We can now perform a Bogoliubov transformation, $\Upsilon \to \tilde{\Upsilon} = \begin{pmatrix} U & V \\ \overline{V} & \overline{U} \end{pmatrix} \Upsilon$, diagonalizing this Hamiltonian,

$$H = \sum_{i=1}^{N} e_i \tilde{a}_i^\dagger \tilde{a}_i - \frac{1}{2} \sum_{i=1}^{N} e_i + E, \tag{190}$$

with $e_i$ the positive and ascending single-particle energies. The ground state is given by the vacuum of the Bogoliubov modes, $|\psi_0\rangle = |\tilde{0}\rangle$, with energy $E_0 = -\frac{1}{2}\sum_{i=1}^{N}(e_i + \mu)$. In Fig. 3 (a) these single-particle energy spectra are displayed as a function of $\mu/t$ for $N = 100$ and $\Delta/t = 1$, and show the existence of a gapless excitation, $e_1 \approx 0$, for $-2t \lesssim \mu \lesssim 2t$. This is the well-known topological phase with a twofold ground state degeneracy. Examining the magnitude of the expansion coefficients of the corresponding Bogoliubov mode, $\tilde{a}_1$, as a function of the original ones (i.e., $|U_{1x}|^2 + |V_{1x}|^2$), we note that these are peaked near the edges of the chain, i.e., we observe a gapless edge mode (Fig. 3 (b)).

We compare the local energy density for specific points in, and between both phases, for the ground state as well as for the first excited states. The latter can be constructed from $|\psi_0\rangle$ by acting with the $a_i^\dagger$ with the lowest energies $e_i$ on the ground state, thus realizing $|\psi_i\rangle = a_i^\dagger|\tilde{0}\rangle$ with for instance $i = 1, 2, 3$ for the three lowest odd parity excitations. Indeed, these states have an opposite parity with respect to the ground state, and in particular for even $N$ yield odd parity excitations.

We can apply our fMPS algorithms for the Kitaev chain Hamiltonian by constructing an MPO representation as described in Eq. (129), which leads to the following explicit form:

$$O_j = \begin{pmatrix} 1 & -ta_j^\dagger & ta_j & -\Delta a_j^\dagger & \Delta a_j & -\mu a_j^\dagger a_j \\ 0 & 0 & 0 & 0 & 0 & a_j \\ 0 & 0 & 0 & 0 & 0 & a_j^\dagger \\ 0 & 0 & 0 & 0 & 0 & a_j^\dagger \\ 0 & 0 & 0 & 0 & 0 & a_j \\ 0 & 0 & 0 & 0 & 0 & 1 \end{pmatrix}. \tag{191}$$

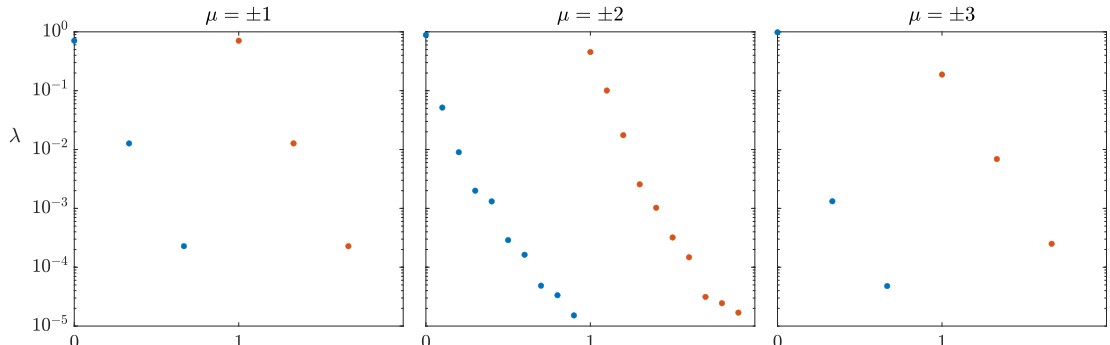

Figure 4: Mid-chain fMPS entanglement spectra for the Kitaev chain of length $N = 100$ within the topological ($\mu = \pm 1$, left) and trivial ($\mu = \pm 3$, right) phase as well as on the boundary between both ($\mu = \pm 2$, middle). Note that in the topological phase, Schmidt values are equal in both parity sectors.

This can again be rewritten in a more compact form, similar to Eq. (182) but now with bond dimension 2 and appropriate boundary tensors. Optimizing fMPS for the ground state by means of the DMRG algorithm, we then obtain an almost exact agreement with the theoretical energy values already for small bond dimensions, in correspondence to Schmidt values down to $10^{-5}$. The mid-chain entanglement spectra are displayed in Fig. 4 and feature identical Schmidt values in both sectors within the topological phase.

We compare the theoretical energy densities with the ones obtained with fMPS using DMRG for the ground state and the quasiparticle Ansatz for the lowest excited states. In Fig. 5 we show both the energy densities and the energy density difference with the ground state, again for a chain of length $N = 100$ with $\Delta/t = 1$. Using lines for the theoretical values, and filled circles for the numerical values, we show that these values are in perfect agreement. With the fMPS approximations, one can also determine the expansion coefficients of the first Bogoliubov mode using that $U_{1x} = \langle \psi_1 | a_i^\dagger | \psi_0 \rangle$ and $V_{1x} = \langle \psi_1 | a_i | \psi_0 \rangle$. The resulting coefficients are compared to their theoretical value in Fig. 3 (b) and clearly illustrate that our formalism is able to capture qualitative as well as quantitative features.

## 5 Jordan & Wigner

At this stage, many a reader has probably been wondering about the relation between fermionic tensor networks and the Jordan-Wigner (JW) transformation. The JW transformation was introduced to demonstrate that 1D systems of $N$ fermionic modes can be embedded into a standard Hilbert space endowed with a standard tensor product. This is a priori not obvious as fermions have a non-trivial braiding. The construction provides a mapping between a complete basis of fermionic operators and a basis of spins, but in a non-local way. This non-locality of the JW transformation is the hallmark of a duality transformation, in which an algebra of operators is mapped to an equivalent algebra (i.e., an algebra with the same structure factors) of operators, acting on a different Hilbert space. A crucial feature of such duality transformations is that they are completely determined by the symmetries of the dual systems [72,73]. These symmetries do not necessarily have to be represented in the same way, but only in a so-called Morita-equivalent way; local symmetric operators are always mapped to equivalent local symmetric operators. Non-symmetric local operators, however, are mapped to non-local operators. When this symmetry is the simplest one possible, namely a global $\mathbb{Z}_2$ symmetry (for instance, all symmetric observables commute with $\otimes_{n=1}^{N} \sigma_n^z$), then two possible duality transformations are the Kramers-Wannier (KW) duality and the Jordan-Wigner transformation.



Figure 5: (left) Energy density $e(x)$ of the ground state and first excited, odd parity states and (right) energy density relative to the ground state for the Kitaev chain of length $N = 100$, with $\Delta/t = 1$ and for different values of $\mu$. Filled circles correspond to numerical values obtained by fMPS, while lines represent theoretical values.

Although this is standard text-book material, let us revisit the JW transformation more closely. Given a set of $N$ fermionic creation and annihilation operators $a_n^\dagger, a_n$ ($n \in \{1, .., N\}$), satisfying the canonical anti-commutation relations

$$\{a_n^\dagger, a_m\} = \delta_{nm}, \qquad \{a_n, a_m\} = 0, \tag{192}$$

and a set of $N$ spins with raising and lowering operators $\sigma_n^\pm = \frac{1}{2}\left(\sigma_n^x \pm i\sigma_n^y\right)$, satisfying the canonical commutation relations

$$[\sigma_n^+, \sigma_m^+] = [\sigma_n^-, \sigma_m^-] = 0, \qquad [\sigma_n^+, \sigma_m^-] = \delta_{nm}\sigma_n^z, \tag{193}$$

we can easily check that both algebras can be related by the following mapping

$$
\begin{aligned}
\sigma_n^+ &= \exp\left(i\pi \sum_{l=1}^{n-1} a_l^\dagger a_l\right) a_n^\dagger, & a_n^\dagger &= \left(\prod_{l=1}^{n-1} \sigma_l^z\right)\sigma_n^+, \\
\sigma_n^- &= \exp\left(i\pi \sum_{l=1}^{n-1} a_l^\dagger a_l\right) a_n, & a_n &= \left(\prod_{l=1}^{n-1} \sigma_l^z\right)\sigma_n^-,
\end{aligned}
\tag{194}
$$

so that $a_n^\dagger a_n = \sigma_n^+ \sigma_n^- = \frac{1}{2}\left(1 + \sigma_n^z\right)$. As local Hamiltonians of fermions always involve an even number of local creation/annihilation operators, the JW mapping from fermions to spins will yield a local $\mathbb{Z}_2$-invariant Hamiltonian of spins.

A natural question is whether we can find an explicit operator $U_{\mathrm{JW}}$, also called an intertwiner, which explicitly maps any operator of spins $O_s$ (also non-symmetric) to an equivalent (possibly non-local) operator of fermions $O_f$,

$$U_{\mathrm{JW}} O_s = O_f U_{\mathrm{JW}}. \tag{195}$$

This intertwiner necessarily must be an isometry, as the spectra of dual theories must be the same. Furthermore, due to the non-local features of JW it should not be possible to build up this isometry from a constant-depth quantum circuit. As it turns out, this intertwiner can easily be written down in the form of a matrix product operator of bond dimension 2. The same holds true for the Kramers-Wannier duality as well as for the composition of both. Because this last one is a bit simpler to analyze, we will first discuss it first, referring to Refs. [26,72] for further details.

The composition of the Jordan-Wigner transformation and Kramers-Wannier duality can be written down as an MPO built up from the CNOT-like fully fermionic tensors

$$G = \sum_{\alpha\beta f} G_{\alpha\beta}^f |\alpha)|f)(\beta| = \sum_{\alpha+\beta=f \bmod 2} |\alpha)|f)(\beta| = \quad {}^{\alpha} \!\leftarrow\! \overset{\phantom{f}}{\boxed{G}} \!\leftarrow\! {}^{\beta}_{\downarrow f} \quad = \quad {}^{\alpha} \!\leftarrow\! \overset{\phantom{f}}{\oplus} \!\leftarrow\! {}^{\beta}_{\downarrow f}, \tag{196}$$

and from the mixed-type, GHZ-like tensors

$$
\begin{aligned}
H = \sum_{\alpha\beta s} H_{\alpha\beta}^s |\alpha)\langle s|(\beta| &= \sum_{\alpha\beta s} \delta_{\alpha s}\delta_{\beta s}|\alpha)\langle s|(\beta| = \quad {}^{\alpha} \!\leftarrow\! \overset{\overset{s}{|}}{\boxed{H}} \!\leftarrow\! {}^{\beta} \quad = \quad {}^{\alpha} \!\leftarrow\! \overset{\overset{s}{|}}{\bullet} \!\leftarrow\! {}^{\beta} \\
&= |s)\langle s|(s|,
\end{aligned}
\tag{197}
$$

where round brackets and legs with arrows correspond to $\mathbb{Z}_2$-graded fermionic spaces while standard brackets and legs without arrows are used for bosonic spaces with $|\!\downarrow\rangle = |0\rangle$ and

$|\uparrow\rangle = |1\rangle$. These two tensors with even fermion parity are alternated to construct the following MPO

$$U^i = \qquad\qquad\qquad\qquad\qquad\qquad\qquad\qquad\qquad\qquad \tag{198}$$

The chain is closed with the fermionic Pauli operators

$$\sigma_f^i = \sum_{\alpha\beta} \sigma_{\alpha\beta}^i |\alpha)(\beta| = \quad \alpha \rightarrow \boxed{\sigma_f^i} \leftarrow \beta \;, \tag{199}$$

including the identity matrix for $i = 1$ and being odd when $i = x, y$.[9] Indeed, a single odd tensor does not tamper with the premises of our formalism because it can still be shuffled around freely as all other tensors are even. Furthermore, note that rather than a single MPO, we constructed a set of four MPO's. This is necessary to provide for a complete map between the full spin and fermion Hilbert spaces. To see this, we explicitly contract the virtual legs where special care has to be taken when contracting the first and the last virtual fermion. Indeed, closing the loop introduces an additional parity factor and thus $\sigma^z$ in the resulting matrix product for an even MPO ($i = 1, z$).[10] This does not happen for the odd case ($i = x, y$). We obtain

$$U^i = \sum_{f_1, s_1, f_2, s_2, \cdots} \text{Tr}\left(\tilde{G}^{f_1} \tilde{H}^{s_1} \tilde{G}^{f_2} \tilde{H}^{s_2} \cdots \tilde{G}^{f_N} \tilde{H}^{s_N} \sigma^{\phi(i)}\right) |f_1\rangle |f_2\rangle \cdots |f_N\rangle \langle s_N| \cdots \langle s_2| \langle s_1| \,, \tag{200}$$

with

$$\tilde{G}^f = \sum_{\alpha+\beta=f \mod 2} |\alpha\rangle \langle\beta| \quad \text{and} \quad \tilde{H}^s = |s\rangle \langle s| \,, \tag{201}$$

now usual (non-graded) matrices and $\phi(1) = z, \phi(z) = 1, \phi(x) = x, \phi(y) = y$. Equivalently, we could write

$$U^i = \qquad\qquad\qquad\qquad\qquad\qquad\qquad\qquad\qquad\qquad \tag{202}$$

where the virtual degrees of freedom are no longer fermionic and where $\sigma_f^i$ is replaced by $\sigma^{\phi(i)}$. An important note is that the physical fermion legs do remain fermionic. Therefore, the CNOT's remain mixed-type, having both an even and an odd part now. As a result, their ordering is important and fermionic minus signs will appear when permuting the fermionic legs. As long as the order of these legs does not change, this problem does not pose itself and we can simply work with the bosonized versions of the GHZ and CNOT tensors having the well-known properties that

$$\tag{203}$$

---

[9]Instead of $\sigma_f^y$, one could equally well use $\sigma_f^x \sigma_f^z = -i\sigma_f^y$. For the usual Jordan-Wigner transformation we will typically make this choice.

[10]After contracting the inner virtual modes, one has to drag the final virtual mode all through the physical modes (amounting to an even state) as well as through the fist virtual mode, yielding an additional parity factor. For more details, see Ref. [9].

These can be used to show that for a spin state $|\psi_s\rangle$ with $\mathbb{Z}_2$ charge $\pm 1$, i.e., $\bigotimes_n \sigma_n^x |\psi_s\rangle = \pm |\psi_s\rangle$, the corresponding fermion state satisfies

$$|\psi_f\rangle = U^i |\psi_s\rangle$$

$$= \pm \quad \text{(204)}$$

For $i = z, x$ we thus obtain $|\psi_f\rangle = \pm|\psi_f\rangle$, showing that states with $\mathbb{Z}_2$ charge $-1$ are mapped to zero. For $i = 1, y$ on the other hand, $\sigma^x \sigma^{\phi(i)} \sigma^x = -\sigma^{\phi(i)}$ resulting in $|\psi_f\rangle = \mp|\psi_f\rangle$ and thus a zero projection for $|\psi_s\rangle$ with $\mathbb{Z}_2$ charge $+1$.[11] If we would only have used the trivial boundary condition, $i = 1$, the MPO would only map from the $\mathbb{Z}_2$ sector with charge $+1$. Analogous considerations on the fermion side show that this $U^1$ would also only target parity even fermion states. We conclude that we need the four $U^i$ to arrive at a complete map between the full spin and fermion Hilbert spaces. In Table 1 we summarize how the different charge sectors are related by these MPO's.

Not only charge sectors are redistributed, also boundary conditions are mapped in a nontrivial way by $U^i$. To see this, consider the translation operator $T_f$ on the fermionic side, defined as

$$T_f a_n T_f^{-1} = a_{n+1}, \tag{205}$$

for $j < N$ and $T_f a_N T_f^{-1} = \pm a_1$ for (anti-)periodic boundary conditions so that $T_f^N = \pm 1$. Diagrammatically $T_f$ can be represented as

$$T_f = \qquad (\sigma_f^z) \qquad \tag{206}$$

with the so-called twist $\sigma_f^z = P$ only present in the case of anti-periodic boundary conditions. Analogously, a translation operator $T_s$ can be defined on the spin side with a $\sigma^x$ twist for anti-periodic boundary conditions. Now suppose we start from a translation-invariant spin state $|\psi_s\rangle$ with periodic boundary conditions, i.e., $T_s |\psi_s\rangle = e^{ik} |\psi_s\rangle$ with $k = \frac{2\pi}{N} n$ and $n \in \{1, .., N\}$. Translating the corresponding fermion state $|\psi_f\rangle$ then yields

$$T_f |\psi_f\rangle = T_f U^i T_s^{-1} T_s |\psi_s\rangle$$

$$= e^{ik} \quad \text{(207)}$$

---

[11]Note that in this calculation bosonizing the virtual legs was quite helpful as it allowed to work with multiple, otherwise odd, $\sigma^x$ tensors. Alternatively, one could retain the fermionic degrees of freedom and introduce some consistent form of ordering. This is indispensable for a 2D extension and explained in more detail in Ref. [26].

Table 1: Summary of how different topological sectors are mapped by $U^i$, the composition of a Jordan-Wigner transformation and a Kramers-Wannier duality (Fermions and Spins columns), and by $U^{\phi(i)}_{\mathrm{KW}}$, a matched Kramers-Wannier duality (Spins and Dual Spins columns). The mapping of topological sectors by their composition, i.e., the usual Jordan-Wigner transformation, is hence given by the Fermions and Dual Spins columns with the combined bosonic boundary condition (BC) in the final column.

| $i$ | Bosonized BC | Fermions $\bigotimes_n \sigma^z_{f_n}$ | Twist | Spins $\bigotimes_n \sigma^x_n$ | Twist | Dual Spins $\bigotimes_n \sigma^z_n$ | Twist | Unified BC |
|---|---|---|---|---|---|---|---|---|
| 1 | $\sigma^z$ | $+1$ | $I$ | $-1$ | $I$ | $+1$ | $\sigma^z$ | $I$ |
| $z$ | $I$ | $+1$ | $\sigma^z_f$ | $+1$ | $I$ | $+1$ | $I$ | $I$ |
| $x$ | $\sigma^x$ | $-1$ | $I$ | $+1$ | $\sigma^x$ | $-1$ | $I$ | $\sigma^x$ |
| $y$ | $\sigma^x\sigma^z$ | $-1$ | $\sigma^z_f$ | $-1$ | $\sigma^x$ | $-1$ | $\sigma^z$ | $\sigma^x$ |

Here, the composition of $U^i$ with the translation operators can be related to the original MPO by moving the boundary condition to the new final site. Based on the form of this boundary condition, this procedure can generate twists on the spin and/or fermion side. If $i = 1$ for instance, the bosonized boundary condition is $\sigma^z$ and moving it through the bosonized GHZ and CNOT tensors, we obtain a $\sigma^z_f$ twist on the fermion side. However, we also reorder the fermion legs and as mentioned before, this generates an additional fermionic parity factor, $(-1)^{|f_N|\sum_{i=1}^{N-1}|f_i|}$. Because the total fermion charge is even for $i = 1$, this factor reduces to $(-1)^{|f_N|}$, i.e., yet another $\sigma^z_f$ on the new first fermion leg. Hence, the twist is removed. Similarly, there is no twist by moving the bosonized BC for $i = z$ but the reordering of the fermion legs introduces a $\sigma^z_f$. This shows that $|\psi_f\rangle$ is not an eigenvector of the periodic $T_f$ then but rather of the anti-periodic version as the latter compensates the twist. We conclude that $U^1$ will map the periodic sector on the spin side to the periodic sector on the fermion side while $U^z$ targets the anti-periodic sector on the fermion side. Similar considerations amount to the twists summarized in Table 1. Note again that we need all four MPO's to fully map the complete spaces.

The usual JW transformation is obtained by doing an extra Kramers-Wannier transformation on the spin side. This KW transformation is of the same form as Eq. (198),

$$U^i_{\mathrm{KW}} = \qquad\qquad\qquad\qquad \boxed{\sigma^i} \qquad\qquad (208)$$

where

$$\sum_{\alpha\beta s}\delta_{\alpha s}\delta_{\beta s}|\alpha\rangle|s\rangle\langle\beta| = \quad {}^{\alpha}\!-\!\!\blacksquare\!\!-\!{}^{\beta} \quad \text{and} \quad \sum_{\alpha+\beta=s \ \mathrm{mod}\ 2}|\alpha\rangle\langle s|\langle\beta| = \quad {}_{\alpha}\!-\!\boxplus\!-\!{}_{\beta} \qquad (209)$$

are fully bosonic now but have the same properties as in Eq. (204). One can again work out how charge sectors on the top and bottom side of $U^i_{\mathrm{KW}}$ are mapped onto each other with the $\mathbb{Z}_2$ charge on top, i.e., for the dual spins, being the eigenvalue of $\bigotimes_n \sigma^z_n$ and the non-trivial dual twists being $\sigma^z$. The results are summarized in Table 1. Note that the KW duality essentially has the effect of swapping the charge and twist columns. Furthermore, note that we constructed the table for $U^{\phi(i)}_{\mathrm{KW}}$ rather than for $U^i_{\mathrm{KW}}$. Indeed, this matching of the boundary conditions makes sure that when gluing the KW on top of $U^i$, the spin charge sectors are the same and the composite transformation, i.e., the JW, will not reduce to zero. Moreover, one

can immediately read off the mapping of the topological sectors for the composite JW MPO. Regarding the tensors in this MPO, we have that

$$\tag{210}$$

where

$$\alpha - \!\!\!\!\!\!\begin{array}{c} s \\ \bigcirc \\ f \end{array}\!\!\!\!\!\! - \beta = \sum_s \sum_{\substack{\alpha+\beta=f \!\!\mod 2}} \delta_{sf} \, |\alpha\rangle \, |f\rangle \, \langle s| \, \langle \beta| \, . \tag{211}$$

The bond dimension thus reduces to 2. For the combined boundary conditions we obtain

$$\tag{212}$$

and

$$\tag{213}$$

so that

$$U_{\text{JW}}^i = \,\overbrace{\quad\bigcirc\!\!-\!\!\bigcirc\!\!-\cdots-\!\!\bigcirc\!\!-\!\!\boxed{\sigma^{\chi(i)}}\quad}$$

$$= \sum_{f_1, f_2, \dots} \text{Tr}\!\left( \tilde{G}^{f_1} \tilde{G}^{f_2} \cdots \tilde{G}^{f_N} \sigma^{\chi(i)} \right) |f_1\rangle |f_2\rangle \cdots |f_N\rangle \, \langle f_N| \cdots \langle f_2| \langle f_1| \, , \tag{214}$$

with $\chi(1) = \chi(z) = 1$ and $\chi(x) = \chi(y) = x$. As opposed to the combination of JW and KW, exactly the same MPO (i.e., with the same BC) deals with both the periodic and anti-periodic case in each charge sector. On the other hand, putting a $\sigma^z$ ($\sigma^y$) BC yields zero in the even (odd) sector. Therefore, we could just as well take sum, returning an MPO with open boundary conditions $|0\rangle \langle 0|$ in the even sector and $|1\rangle \langle 0|$ in the odd. Then the JW transformations, both for the even and odd case, can be unified in a single MPO with boundary term $(|0\rangle + |1\rangle) \langle 0|$. This is the reason that one typically just talks about one unique JW transformation.

Clearly, the $\mathbb{Z}_2$-graded tensor network language yields a pretty natural framework for formulating and understanding the intricacies of the JW transformation. What is more, it can also be naturally extended to higher dimensions by explicitly retaining the fermionic nature of the virtual legs. Again, we refer to Ref. [26] for more details. The question whether it is better to use graded tensor networks or to use standard tensor network methods on a JW transformed Hamiltonian also imposes itself now. The answer is certainly not unequivocal, but will depend on the questions that one wishes to ask. For example, the nature of the excitations might change drastically under duality transformations: local excitations can become topological domain-wall like excitations and vice-versa. Also, local symmetries in one representation can become non-local in the other one. Equally so, one has to be very careful in labelling the different entanglement spectra, as duality transformations permute the different labels (as examplified in the table above).

# 6 Fermionic PEPS

Fermionic PEPS (fPEPS) represent the 2D analogue of the fermionic MPS discussed in the previous sections. Here, we explain how one constructs fermionic PEPS within the framework of $\mathbb{Z}_2$-graded Hilbert spaces. Next, contraction of these tensor networks is considered with the aim to calculate expectation values. Both the boundary MPS (also referred to as MPS-MPO) approach [19,20] and CTMRG [16–18] are considered. Finally, we also discuss the variational optimization of fermionic PEPS.

## 6.1 General form

The conceptual construction of a 2D fermionic PEPS strongly resembles that of the 1D fermionic MPS. To each lattice site we associate virtual fermionic degrees of freedom but now in both spatial directions. For a simple square lattice, we thus have a pair of virtual degrees of freedom per lattice site in both the horizontal and the vertical direction. Next, neighboring virtual fermions are brought into a maximally entangled state,

$$|\Psi_{\text{virt}}\rangle = \prod_{e_h}\left(1 + a^\dagger_{e_h,l}a^\dagger_{e_h,r}\right)\prod_{e_v}\left(1 + a^\dagger_{e_v,d}a^\dagger_{e_v,u}\right)|\Omega_{\text{virt}}\rangle = \qquad (215)$$

after which we introduce a linear, parity-preserving map, locally projecting the four virtual modes per site onto a single physical mode,

$$\mathcal{A}_j = \left(A_j\right)_{\alpha_1\alpha_2\sigma\alpha_3\alpha_4}(a_{j,l})^{\alpha_1}(a_{j,d})^{\alpha_2}(a^\dagger_j)^\sigma(a_{j,r})^{\alpha_3}(a_{j,u})^{\alpha_4}. \qquad (216)$$

As was the case in 1D, each virtual link amounts to a Kronecker delta connecting neighboring virtual indices. However, as the projectors are no longer placed along a line, fermionic parity factors will appear as well. Still, the final result can be reinterpreted as a contraction of local $\mathbb{Z}_2$-graded tensors,

$$|\Psi(A)\rangle = \mathcal{C}_{\text{virt}}\left(\prod_j\left(\left(A_j\right)_{\alpha_j\beta_j\gamma_j\varepsilon_j\delta_j}|\alpha_j\rangle|\beta_j\rangle|\gamma_j\rangle\langle\varepsilon_j|\langle\delta_j|\right)\right), \qquad (217)$$

where the fPEPS tensors are defined as

$$\alpha \leftarrow \boxed{A_j} \leftarrow \varepsilon \quad = \left(A_j\right)_{\alpha\beta\gamma\delta\varepsilon}|\alpha\rangle|\beta\rangle|\gamma\rangle\langle\varepsilon|\langle\delta|, \qquad (218)$$

with $|\gamma\rangle$ labeling the physical leg. Assuming translation invariance, the fPEPS construction can thus be summarized as

$$|\Psi(A)\rangle = \qquad (219)$$

As was the case for fMPS, both the physical and virtual Hilbert spaces can be enlarged. Taking $f$ fermions on the physical level for instance amounts to a local Hilbert space of dimension $d = 2^f$ whereas for the virtual spaces arbitrary natural bond dimensions $D$ are possible, resulting in a fPEPS object living in a $dD^4$-dimensional, graded Hilbert space. Also note that upon contraction of these objects, fermionic minus signs due to supertraces are inevitable in contrast to what we observed in 1D for fMPS.

Analogous to the 1D case, fermionic PEPS have gauge freedom due to their specific virtual structure, i.e., an operation

$$\text{(220)}$$

leaves the fPEPS unchanged but now different gauge transforms can be applied in both directions. However, unlike in 1D, it is not possible to use this gauge transforms to impose canonical forms that significantly simplify the calculation of expectation values and ensure unit normalisation. Just like their bosonic counterparts, fermionic PEPS thus require approximate methods for their contraction.

## 6.2 Boundary MPS

One possible approach to contract an fPEPS relies on the determination of boundary MPS. We will first illustrate this for the calculation of the norm of a fPEPS, which can be expressed as

$$\langle \Psi(A)|\Psi(A)\rangle = \qquad \text{(221)}$$

where the double-layer tensor $O$ is defined by

$$\text{(222)}$$

and where

$$\text{(223)}$$

is just an internally permuted version of the conventional $\bar{A}$. Again, all the physical legs in $|\Psi(A)\rangle$ point outward so that no $P$ tensors appear when taking the norm by contracting with

its conjugate. In order to contract this tensor network, we calculate the leading top and bottom eigenvector, also called fixed points, of the horizontal linear transfer matrix,

$$T(O) = \qquad\qquad\qquad\qquad\qquad\qquad\qquad\qquad (224)$$

Once computed, both can be contracted, yielding the norm of $|\Psi(A)\rangle$.

To obtain the fixed points, we approximate them by a uniform MPS with a certain boundary bond dimension $\chi$ and with two "physical" legs instead of one. I.e., we propose

$$|\zeta(M)\rangle = \qquad\qquad\qquad\qquad\qquad\qquad\qquad\qquad (225)$$

for the top boundary state. The relevant eigenvalue equation then becomes

$$\approx \Lambda_t \qquad\qquad\qquad\qquad\qquad\qquad\qquad\qquad (226)$$

Note that this is not equivalent to applying the corresponding operator, $T |\zeta(M)\rangle = \Lambda_t |\Psi(M)\rangle$. Indeed, in the latter $P$ tensors are placed on all of the upward pointing legs of $|\zeta(M)\rangle$. Therefore, we redefine our transfer operator by

$$\tilde{T}(\tilde{O}) = \qquad = \qquad\qquad\qquad\qquad\qquad\qquad (227)$$

where the circular tensors correspond to $P$. Application of this slightly altered operator to $|\zeta(M)\rangle$ then amounts to the desired eigenvalue equation. Similarly, the solution of the bottom fixed-point equation could be approximated by an MPS having the structure of the conjugate of $|\zeta(M)\rangle$, i.e.,

$$\langle\zeta(N)| = \qquad\qquad\qquad\qquad\qquad\qquad\qquad\qquad (228)$$

The appropriate eigenvalue equation then becomes $\langle\zeta(N)| \tilde{T} = \Lambda_b \langle\zeta(N)|$, corresponding to

$$= \Lambda_b \qquad\qquad\qquad\qquad\qquad\qquad\qquad\qquad (229)$$

A useful feature of these definitions of the fixed-points is that the bottom eigenvalue equation can be rewritten in the form of the top one. Taking the adjoint of the bottom equation, we obtain $\tilde{T}^\dagger |\zeta(N)\rangle = \Lambda_b^* |\zeta(N)\rangle$, i.e., the top equation but now for $\tilde{T}^\dagger$. Indeed, $|\zeta(M)\rangle$ is the right eigenvector of $\tilde{T}$ (as an operator) while $\langle\zeta(N)|$ is the left eigenvector. Writing out the resulting equation in a diagram, we obtain

$$= \Lambda_b^* \qquad\qquad\qquad\qquad\qquad\qquad\qquad\qquad (230)$$

We conclude that the bottom fixed-point equation can be transformed to a top fixed-point equation as in Eq. (226) but with $M \to N$, $\Lambda_t \to \Lambda_b^*$ and

$$\begin{array}{ccc} \boxed{O} & \to & \boxed{\bar{O}} \end{array} \tag{231}$$

Alternatively, we could absorb the parities on the vertical legs in the $N$ tensors. This does remove quite some parities from the following equations for the general case of non-Hermitian $\tilde{T}$ where $M \not\approx N$. However, for Hermitian $\tilde{T}$, one would have to add these $P$ tensors again instead of just approximating the $N$ by $M$ tensors (see Eq. (245)). Therefore, we do not absorb the parities here. With $\Lambda_t$ being the dominant eigenvalue of $\tilde{T}$ and $\Lambda_b^*$ of $\tilde{T}^\dagger$, we also immediately have that $\Lambda_t \approx \Lambda_b$ and denote both with $\Lambda$. To solve the top fixed-point equations for $M$ and $N$, the VUMPS algorithm can be applied again but now for MPS tensors with two physical legs. This introduces some slight alterations to the method that we work out in Appendix A.3.

After computing the fixed-points in their center-site gauge with the VUMPS algorithm, the norm and local expectation values of fermionic PEPS can easily be computed. For instance, sequential application of the eigenvalue equation to the overlap between the boundary states yields

$$\langle \Psi(A)|\Psi(A)\rangle = \Lambda^{|\mathbb{Z}|} \langle \zeta(N)|\zeta(M)\rangle = \Lambda^{|\mathbb{Z}|} \begin{array}{c} \boxed{M^l} \; \boxed{M^c} \; \boxed{M^r} \\ \boxed{\bar{N}^l} \; \boxed{\bar{N}^c} \; \boxed{\bar{N}^r} \end{array} \tag{232}$$

Indeed, with every application of Eq. (226) or Eq. (229), an additional $T(O)$ layer appears, finally yielding the expression for the norm up to a (possibly infinite) factor. Essentially, this factor is equal to $\Lambda^{N_v}$ with $N_v$ the infinite number of vertical layers. Furthermore, Appendix A.3 learns that $\Lambda = \lambda^{N_h}$ with $N_h$ the (infinite) number of sites in the horizontal direction and $\lambda$ the dominant eigenvalue obtained in the determination of the environments (Eq. (A.31)). We thus find that $\langle \Psi(A)|\Psi(A)\rangle = \lambda^{N_h N_v} \langle \zeta(N)|\zeta(M)\rangle$. For the remaining MPS overlap, we determine left and right environments,

$$\begin{array}{c} \boxed{G^l} \boxed{M^l} \\ \boxed{\bar{N}^l} \end{array} = \mu \; \boxed{G^l} \qquad \text{and} \qquad \begin{array}{c} \boxed{M^r} \boxed{G^r} \\ \boxed{\bar{N}^r} \end{array} = \mu \; \boxed{G^r} \tag{233}$$

that we normalize as

$$\begin{array}{c} \boxed{G^l} \; \widehat{C_M} \; \boxed{G^r} \\ \; \widehat{\bar{C}_N} \end{array} = 1 \,, \tag{234}$$

with $C_M$ ($C_N$) the canonical $C$ tensor for the top (bottom) boundary MPS so that for the norm $\langle \Psi(A)|\Psi(A)\rangle = \lambda^{N_h N_v} \mu^{N_h}$. This shows that normalizing an arbitrary PEPS is not as straightforward as for MPS. However, one typically does not require a normalized state for the calculation

of local expectation values. Indeed, consider a strictly local operator $W$, then

$$\langle\Psi(A)|W|\Psi(A)\rangle = \tag{235}$$

with

$$\tag{236}$$

Using the fixed points, this reduces to

$$\langle\Psi(A)|W|\Psi(A)\rangle = \lambda^{N_h(N_v-1)} \tag{237}$$

where environments can be defined as

$$= \mu \quad \text{and} \quad = \mu \tag{238}$$

with

$$= 1, \tag{239}$$

so that

$$\langle\Psi(A)|W|\Psi(A)\rangle = \lambda^{N_h(N_v-1)}\mu^{N_h-1} \tag{240}$$

However, as we did not explicitly normalize the PEPS, we are rather interested in

$$\frac{\langle\Psi(A)|W|\Psi(A)\rangle}{\langle\Psi(A)|\Psi(A)\rangle}, \tag{241}$$

where the norm can also be expressed as

$$\langle \Psi(A)|\Psi(A)\rangle = \lambda^{N_h(N_v-1)} \quad \begin{array}{c}\includegraphics{}\end{array} \quad = \lambda^{N_h(N_v-1)}\mu^{N_h}, \tag{242}$$

so that

$$\frac{\langle \Psi(A)|W|\Psi(A)\rangle}{\langle \Psi(A)|\Psi(A)\rangle} = \frac{1}{\mu} \quad \begin{array}{c}\includegraphics{}\end{array} \tag{243}$$

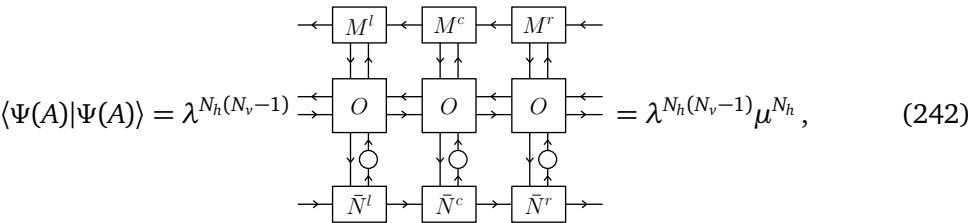

In this way, all the possibly infinite prefactors have been eliminated. We conclude that calculating local expectation values poses no problem for fermionic PEPS. Similarly, two-point correlation functions in the horizontal and vertical direction can be calculated efficiently in this way (see Sec. 7).

The contraction of PEPS becomes particularly manageable when the linear transfer operator is Hermitian. Indeed, when $\tilde{T} = \tilde{T}^\dagger$, the bottom fixed point is equal to the adjoint of the top fixed point and both have the same real eigenvalue. As a result only a single VUMPS run has to be performed to obtain both. Furthermore, the norm can be expressed as

$$\langle \Psi(A)|\Psi(A)\rangle = \lambda^{N_h N_v} \langle \Psi(M)|\Psi(M)\rangle$$
$$= \lambda^{N_h N_v} \quad \begin{array}{c}\includegraphics{}\end{array} \quad = \lambda^{N_h N_v}, \tag{244}$$

with $\lambda \in \mathbb{R}$ and positive so that it can be regarded as a norm density. Hence, dividing each $A$ tensor with $\sqrt{\lambda}$ immediately yields a normalized PEPS. To impose the hermiticity on an arbitrary fPEPS, one could use Eq. (231). When

$$\begin{array}{c}\includegraphics{}\end{array} \quad = \quad \begin{array}{c}\includegraphics{}\end{array} \tag{245}$$

$\tilde{T}$ will be Hermitian. Writing out this equality in terms of the PEPS tensor and its conjugate, we get

$$\begin{array}{c}\includegraphics{}\end{array} \quad = \quad \begin{array}{c}\includegraphics{}\end{array} \tag{246}$$

Note that on the left-hand side of the equation, the upper left legs belong to a single tensor whereas this is not the case on the right-hand side. Therefore, we introduce the following

unitary operations in the latter,

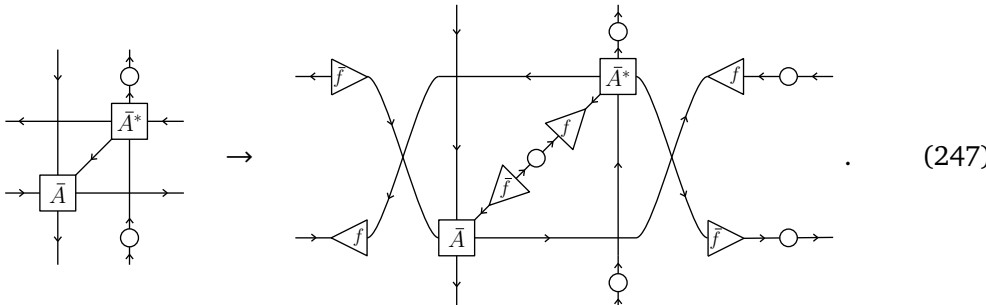

$$\hspace{11cm} . \hspace{1cm} (247)$$

Indeed, making a chain, the swaps and flippers are undone so that the sufficient condition remains satisfied. Furthermore, we can express it on the level the PEPS tensor as the combination of

$$\hspace{11cm} , \hspace{1cm} (248)$$

and

$$\hspace{11cm} . \hspace{1cm} (249)$$

Rewriting $A^*$ in terms of $\bar{A}$, one can easily show that both equations are equivalent. Imposing this constraint Eq. (248), we are thus guaranteed to have a Hermitian $\tilde{T}$.

## 6.3 Corner transfer matrix renormalization group

The corner transfer matrix renormalization group algorithm [16–18] represents a complementary alternative to the boundary MPS approach for contracting PEPS. At its core is the assumption that the double layer of a PEPS and its conjugate,

$$\langle \Psi(A) | \Psi(A) \rangle = \hspace{6cm} (250)$$

can be represented as

$$\hspace{11cm} (251)$$

where the corner transfer matrices (CTMs) $\{C_1, C_2, C_3, C_4\}$ approximate the infinite corners of the network while the half rows and columns are approximated by $\{T_1, T_2, T_3, T_4\}$. The

virtual legs regulating the size of these transfer matrices correspond to spaces of dimension $\chi$. As a result, a local expectation value can be computed by performing the same contraction as in Eq. (251) but with $O$ replaced by $O_W$. To determine the $C$ and $T$ tensors, the CTMRG algorithm in its most general form consists of three steps respectively applied in the left, right, upward and downward direction:

- *insertion*: the network from Eq. (251) is enlarged, here in the left direction (i.e., a "left move") by inserting a new column containing $T_1$, $O$ and $T_3$,

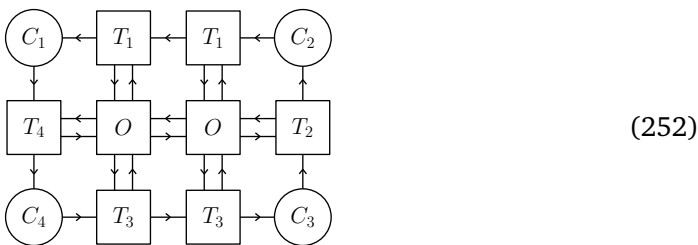

$$(252)$$

- *absorption*: the inserted tensors are contracted with their neighboring $C$ and $T$ yielding the bigger

$$(253)$$

$$(254)$$

and

$$(255)$$

- *renormalization*: $\tilde{C}_1$, $\tilde{T}_4$ and $\tilde{C}_4$ are truncated back to a virtual space of the original size by applying the projectors

$$(256)$$

yielding the updated transfer matrices

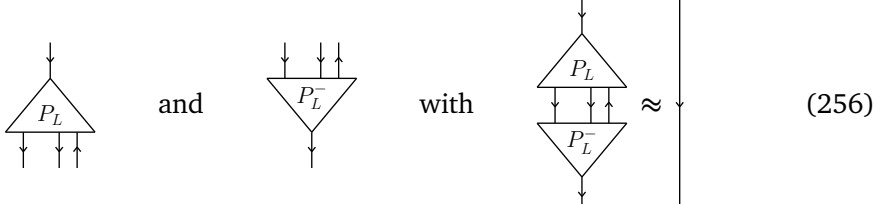

$$(257)$$

Cycling through the four directions while repeating these steps and normalizing the transfer matrices, the CTMRG algorithm will converge towards a fixed point, invariant under new insertions. As a result, the norm of the PEPS as expressed in Eq. (251) can equally well be expressed as

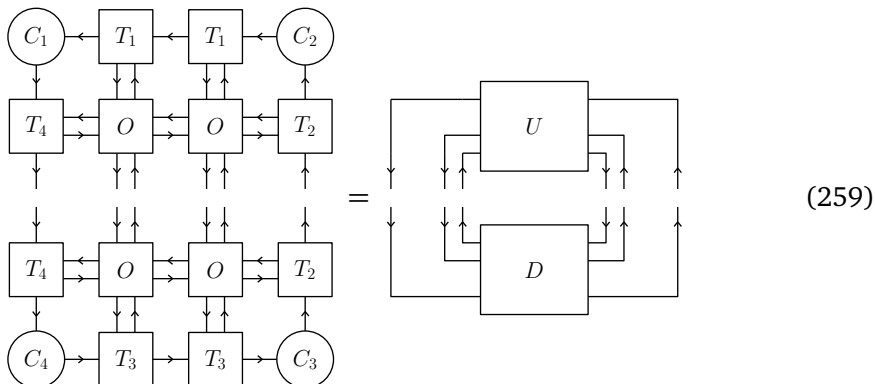 (258)

Multiplying the former and Eq. (251) and dividing by the latter two then yields a quantity whose difference between consecutive iterations can be used as an error measure.

Determination of a proper choice for the projector in the renormalization step is crucial for the stability and efficiency of the algorithm. Many different methods for obtaining these projectors have been proposed. Here, we focus on the one introduced in Ref. [42] and reformulated in Ref. [19], a recurring choice in many state of the art CTMRG codes. For the left move it is constructed as follows.

- First, define the half-system transfer matrices $U$ and $D$,

$$= \qquad (259)$$

and perform an SVD of their left-side contraction,

$$\approx \qquad (260)$$

where the approximate equality indicates that we only keep the highest singular values to truncate back to the desired bond dimension.

- Construct the projectors

$$= \qquad (261)$$

and

$$= \qquad (262)$$

where $\Sigma_L^+$ is the pseudoinverse of $\Sigma_L$ and where the circular tensors are parity tensors. Indeed, these are necessary to make use of the left (right) isometry of $U_L$ ($V_L$) so that Eq. (256) holds. The only alteration to the CTMRG algorithm for fermions thus consists of adding $P$ tensors to the SVD decompositions in the renormalization step.

## 6.4 Variational optimization

To perform a direct variational optimization within the (fermionic) PEPS manifold, one combines the energy evaluation, based on one of the contraction algorithms above, with a calculation of the gradient w.r.t. the variational parameters. Together, these ingredients can be used in a gradient search like LBFGS to minimize the energy. As mentioned before, PEPS gradients can be determined by means of corner environments but recently gradients calculated with automatic differentiation (AD) [74] have become more popular. In this case, one runs through the computation of the objective function (in our case the energy density) in a backwards fashion and by specifying the Jacobians of each of the computational steps, an AD engine can combine all these in a systematic and efficient way to compute the gradient in a time on par with the energy evaluation itself. This has become the state of the art method to optimize PEPS, both by means of boundary MPS contraction [75] and CTMRG [47, 76].

# 7 Applications in 2D

To benchmark the fermionic TNS methods in 2D, we reproduce correlation functions of Gaussian fermionic TNS (GfTNS) [33] approximating the ground state of a chiral model. Other studies have found that in chiral phases, variationally optimized PEPS tend to reproduce the exponentially decaying correlation functions up to a certain bulk correlation length, regulated by the PEPS bond dimension $D$, after which a, possibly polynomial, long-range tail takes over [77]. Here, we intend to verify this behavior with GfTNS for the p-wave superconductor. Subsequently, generic fPEPS algorithms are used to reproduce the Gaussian correlation functions. For the smaller bond dimensions, we also compare the Gaussian energies with those obtained for variationally optimized generic fPEPS.

## 7.1 Long-range correlations in chiral superconductors

Consider the p-wave superconductor,

$$H = -t \sum_{\mathbf{n}} \left( a_{\mathbf{n}}^\dagger a_{\mathbf{n}_\rightarrow} + a_{\mathbf{n}}^\dagger a_{\mathbf{n}_\uparrow} + h.c. \right) - \mu \sum_{\mathbf{n}} a_{\mathbf{n}}^\dagger a_{\mathbf{n}} - \Delta \sum_{\mathbf{n}} \left( a_{\mathbf{n}}^\dagger a_{\mathbf{n}_\rightarrow}^\dagger + i a_{\mathbf{n}}^\dagger a_{\mathbf{n}_\uparrow}^\dagger + h.c. \right),$$

where $\mathbf{n}_{\rightarrow(\uparrow)}$ corresponds to the nearest neighbor of site $\mathbf{n}$ on the right (upper) side in a square lattice. More specifically, we examine the point where $\Delta/t = 0.2$ and $\mu/t = -2$, i.e., a gapped and chiral topological superconductor with Chern number $C = 1$. As this model is free it can be solved by a Bogoliubov transformation. The quadratic nature also justifies the use of GfTNS to efficiently approximate the ground state, e.g. by minimizing their energy density with a gradient descent method. Indeed, for GfTNS these local observables can be calculated analytically using correlation matrices rather than via approximate environments, allowing for a simpler and faster optimization [78]. However, the restriction to GfTNS also comes at a cost. Calculating local observables for a 2D translation invariant GfTNS in the thermodynamic limit for instance requires the computation of continuous $\mathbf{k}$-space integrals. In more than one spatial dimension this can no longer be done analytically and therefore one has to work with finite lattices containing $N_s$ sites and (anti-)periodic boundary conditions, thus replacing the integrals with finite sums. More importantly, the no-go theorem of Dubail and Read

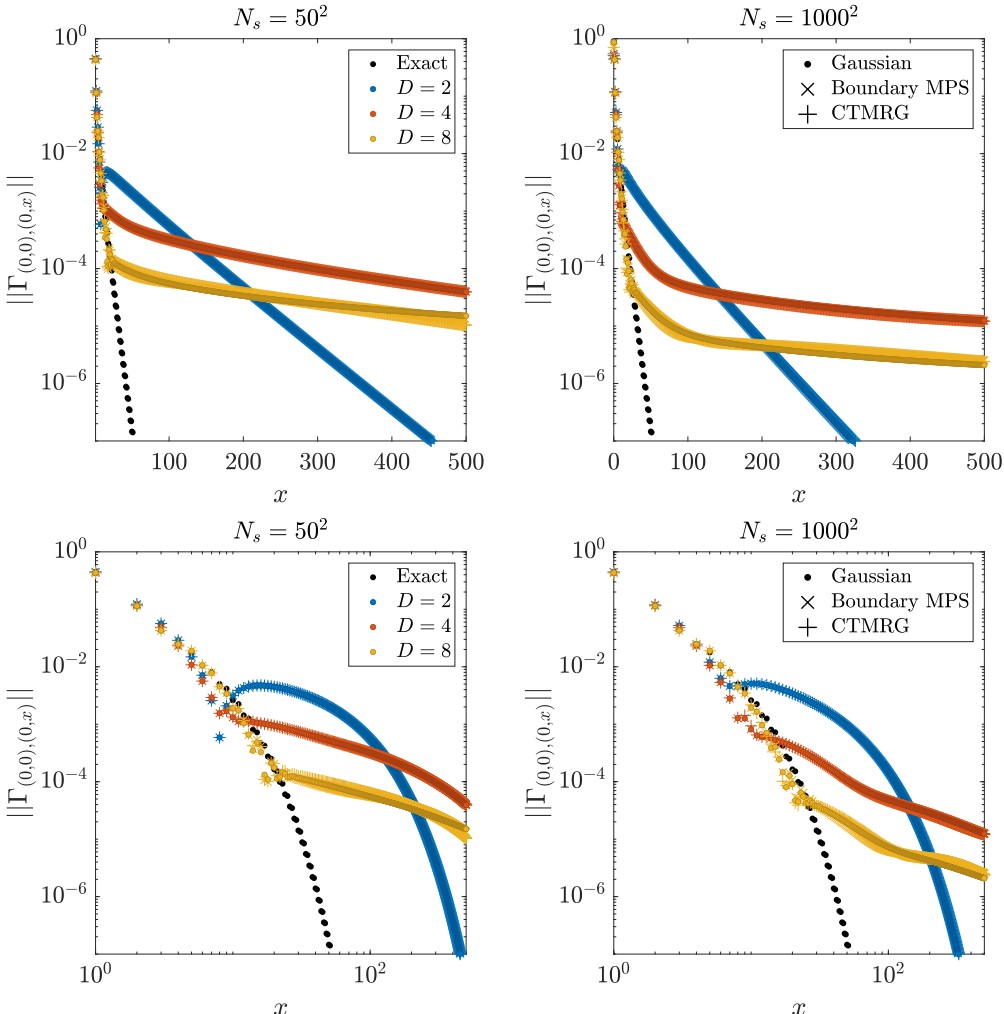

Figure 6: Semi-logarithmic (top panels) and double logarithmic (bottom panels) plots of the real-space correlations, more specifically $||\Gamma_{(0,0),(0,x)}||$, the Frobenius norm of the real-space correlation matrix between sites $(0,0)$ and $(x,0))$ for the chiral p-wave superconductor at $\Delta/t = 0.2$ and $\mu/t = -2$. Exact results obtained by means of a Bogoliubov transformation are compared to those for GfTNS (circular markers) evaluated on $3000 \times 3000$ lattice but optimized on both small ($N_s = 50^2$, left panels) and large lattices ($N_s = 1000^2$, right panels) and with bond dimensions $D = 2, 4, 8$. For the latter, polynomial tails are only resolved at elevated $D$ and $N_s$. These Gaussian results were reproduced with generic fPEPS methods and the $\times$ (+) sign markers correspond to values obtained with boundary MPS (CTMRG) where $\chi = 80, 200, 400$ to contract the infinite-size fPEPS with $D = 2, 4, 8$ obtained from the optimized GfTNS tensors.

applies, stating that non-singular GfTNS cannot realize chiral features and therefore have a trivial Chern number [30]. As explained in Ref. [79], optimizing GfTNS while starting from a random and thus non-singular initial guess, these states will never become truly singular since this requires a strict fine-tuning of their parameters. The approximate GfTNS will therefore never become truly chiral, and ultimately exponential decay will take over again after the polynomial decay at intermediate distances. Indeed, a more detailed examination shows that the Berry curvature generated all through the Brillouin zone is compensated in a very small region around $\mathbf{k} = 0$. In real space, this results in a polynomial tail for the correlations up to

a corresponding (large) length scale after which the final exponential decay takes over. The compensating region in **k**-space was found to shrink by increasing the bond dimension as well as the finite system size $N_s$ used during the GfTNS optimization. Indeed, a higher $N_s$ samples more points close to the zone center so that the cost function tries to shrink this region further. Increasing $N_s$ and $D$, the length scale for the onset of the exponential tail hence increases. All of these features can be confirmed in Fig. 6 for $||\Gamma_{(0,0),(x,0)}||$, the Frobenius norm of the real-space correlation matrix between sites $(0,0)$ and $(x,0)$,

$$\Gamma^{kl}_{(0,0),(x,0)} = \frac{i}{2} \langle \Psi | \left[ c^k_{(0,0)}, c^l_{(x,0)} \right] | \Psi \rangle \,, \tag{263}$$

with $c^1_{\mathbf{n}} = a^\dagger_{\mathbf{n}} + a_{\mathbf{n}}$ and $c^2_{\mathbf{n}} = -i(a^\dagger_{\mathbf{n}} - a_{\mathbf{n}})$ the Majorana operators. At elevated $D$ and $N_s$ the exact, exponentially decaying region is followed by a polynomial regime. However, Fig. 6 also shows that this polynomial regime is followed again by a second exponential decay. When $D$ and $N_s$ are too low, the polynomial region shrinks or completely disappears. If $D = 2$, for instance, the two exponential regimes immediately follow each other, independent of $N_s$. For $D = 4$ on the other hand, $N_s = 50^2$ results in a limited polynomial regime quickly followed by the second exponential decay while $N_s = 1000^2$ results in a significantly longer polynomial tail. If $D = 8$, the second exponential region is never even probed for separations $x$ ranging up to 500 sites. The notion that chiral features in PEPS require long-range tails in their correlation functions is thus sustained and even explained in the sense that it is a consequence of the no-go theorem. However, for approximate, non-fine-tuned GfTNS, this polynomial region also has a finite length scale after which a second exponential regime sets in.

The peculiar correlation functions described above represent an interesting test case for the contraction methods outlined in the previous sections. Indeed, after casting the approximate GfTNS in a generic fPEPS format as described in Appendix B, these methods should reproduce the Gaussian results. Normalized correlations in the horizontal direction like $\langle a^\dagger_{(0,0)} a_{(x,0)} \rangle$ and $\langle a_{(0,0)} a_{(x,0)} \rangle$ can be calculated using the boundary MPS by contracting

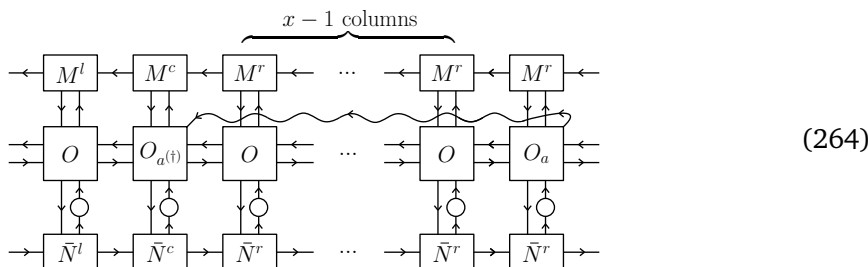

$$\tag{264}$$

where

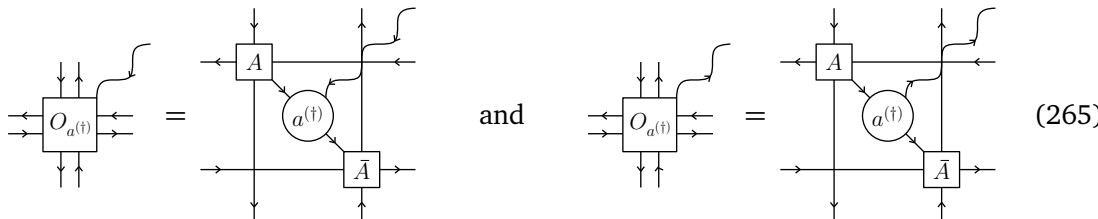

$$\tag{265}$$

contain the physical creation and annihilation operators from Eq. (96) and Eq. (97) (also note the difference in the auxiliary arrows). For the normalized correlations, we thus obtain

$$\langle a^{(\dagger)}_{(0,0)} a_{(x,0)} \rangle = \frac{1}{\mu^{x+1}} \quad$$ 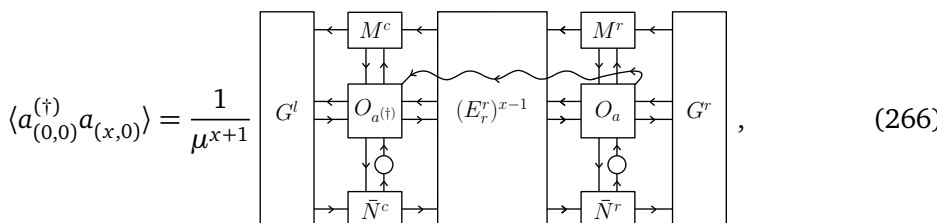 $$, \tag{266}$$

with

$$\text{(267)}$$

and $\mu$, $G^l$ and $G^r$ as in Eq. (238).

Similarly, the converged CTMs and half row/column transfer matrices yield the unnormalized correlations

$$\langle a^{(\dagger)}_{(0,0)} a_{(x,0)} \rangle = \text{(268)}$$

with

$$\text{(269)}$$

the horizontal transfer matrix. An efficient and stable way to calculate these correlations proceeds by setting

$$\text{(270)}$$

and

$$\text{(271)}$$

Table 2: Comparison of ground state energy densities for the p-wave superconductor model with $\Delta/t = 0.2$ and $\mu/t = -2$. The exact energy density $e_{\text{exact}} = -0.209431$ was approximated by fully converged GfTNS at $D = 2$ and $D = 4$. Starting from a random initial guess, generic fPEPS with these $D$ were optimized as well and demonstrated to improve on the Gaussian results.

|  | $D = 2$ | $D = 4$ |
|---|---|---|
| $e_{\text{GfTNS}}$ | -0.17872 | -0.208598 |
| $e_{\text{fPEPS}}$ | -0.17874 | -0.209019 |

Thus assuming that $G^l$ ($G^r$) approximates the dominant left (right) eigenvector upon contraction with $E_h$ for the converged CTMRG environments, the normalized correlations can be expressed as

$$\langle a_{(0,0)}^{(\dagger)} a_{(x,0)} \rangle = \frac{1}{\mu^{x+1}} \quad \boxed{\begin{array}{ccc} & T_1 & T_1 \\ G^l & O_{a^{(\dagger)}} \; E_h^{x-1} \; O_a & G^r \\ & T_3 & T_3 \end{array}} \tag{272}$$

From the $\langle a_{(0,0)}^{\dagger} a_{(x,0)} \rangle$ and $\langle a_{(0,0)} a_{(x,0)} \rangle$ correlation functions we compute $||\Gamma_{(0,0),(0,x)}||$. Comparing the results to the Gaussian correlation functions in Fig. 6 confirms that both generic contraction methods yield a satisfactory reproduction and this with manageable environment bond dimensions $\chi = 80, 200, 400$ for $D = 2, 4, 8$.

As an additional benchmark, we optimized generic fPEPS using CTMRG contraction in combination with an AD gradient calculation. For the latter, we made use of the CTMRG fixed-point equation with gauge-fixed environments [80] to speed up the computation. Table 2 compares the energies at the Gaussian minimum for the lowest bond dimensions with those that were attained by generic fPEPS. We note that the latter are not fully converged but rather demonstrate that one can decrease energies below the Gaussian optimimum.

# 8 Conclusions

We have shown that fermions can be directly built into tensor network states without a need to keep track of leg crossings explicitly. Indeed, constructing fermionic tensors as elements of graded Hilbert spaces, we derived a methodology that takes into account their anti-commuting nature within the tensor structure itself, as well as in other tensor operations such as contractions and decompositions. The resulting formalism can be diagrammatically represented, with fermionic tensors having a specific orientation for each leg, in line with common practices to incorporate global symmetries. As a result, it can be readily integrated in standard tensor network toolboxes with symmetries.

Once built up, we have used the formalism to construct fermionic versions of MPS and PEPS, and discussed standard algorithms like DMRG, tangent-space methods and CTMRG. Highlighting the differences with their bosonic counterparts in the form of additional parity tensors, we have fermionized important elements of the traditionally bosonic TNS toolbox. We have benchmarked these algorithms against interesting test cases like the Kitaev chain where we resolve the gapless edge modes, the 1D Hubbard model exhibiting spin-charge separation and the chiral p-wave superconductor in 2D with characteristic long-range tails in the TNS approximations of real-space correlations.

Though this work establishes the main rationale behind a consistent fermionic formalism, it only serves as a starting point. Numerous concepts and algorithms within the extensive tensor network literature have yet to be explored. RG-like procedures that alter the lattice geometry, such as the tensor renormalization group (TRG), and their connection to spin structures, as a condition for consistent spinful fermionic theories in the continuum, remain unexplored. The $\mathbb{Z}_2$ grading underlying our formalism can also be extended to $\mathbb{Z}_p$, effectively incorporating parafermions [81]. This approach has already been followed to explore potential gapped phases but one could again aim to extend the TNS toolbox based on this more complicated grading [82, 83]. Another direction for future research involves a detailed comparison of the advantages of working directly with fermions versus employing a Jordan-Wigner transformation first [26]. Especially in higher dimensions, the interplay between more involved global symmetries and periodic as well as more exotic boundary conditions remains to be explored. Nonetheless, the graded formalism presented here appears to be the most natural language to address these questions.

## Acknowledgments

We acknowledge useful discussions with Maarten Van Damme and Laurens Lootens.

**Funding information** This research is supported by the European Research Council (ERC) under the European Union's Horizon 2020 research and innovation program Grant agreements No. 101076597 - SIESS (NB), the Research Foundation Flanders, EOS (grant No. 40007526), IBOF (grant No. IBOF23/064), and BOF-GOA (grant No. BOF23/GOA/021). LB is supported by a PhD fellowship 11H7223N from the Research Foundation Flanders. LV is supported by the FNRS (Belgium).

## A  Fermionic tangent space methods

### A.1  Tangent-space projector for fermionic MPS

We derive an expression for $P_{|\Psi(A)\rangle}$, the tangent space projector in and orthogonal to a point $|\Psi(A)\rangle$ on the uniform fMPS manifold. Therefore, consider an arbitrary translation invariant state $|\chi\rangle$. Its projection onto the tangent space, $|\Phi(B(X);A)\rangle = P_{|\Psi(A)\rangle}|\chi\rangle$, can be parametrized by a tensor $X$ as in Eq. (146) where

$$X = \min_X \||\chi\rangle - |\Phi(B(X);A)\rangle\|^2 \tag{A.1}$$
$$= \min_X \left(\langle\Phi(B(X);A)|\Phi(B(X);A)\rangle - \langle\Phi(B(X);A)|\chi\rangle - \langle\chi|\Phi(B(X);A)\rangle\right).$$

Differentiating w.r.t. the complex conjugate of the entries of $X$, this minimum is characterized by

$$0 = \frac{\partial}{\partial\bar{X}} \||\chi\rangle - |\Phi(B(X);A)\rangle\|^2, \tag{A.2}$$

i.e.,



$$= 0, \tag{A.3}$$

so that

$$|\Phi(B(X);A)\rangle = \sum_j$$ 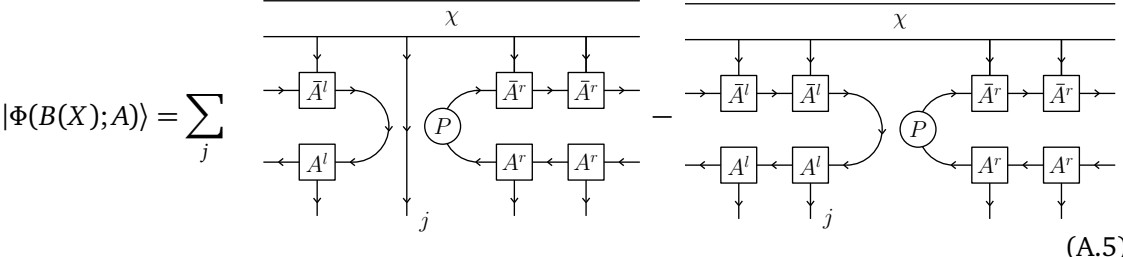 $$(A.4)$$

Using the completeness relation from Eq. (147), we can rewrite this as

$$(A.5)$$

yielding the final result for the tangent space projector

$$P_{|\Psi(A)\rangle} = \sum_j \qquad - \qquad (A.6)$$

## A.2 Vumps for fermionic MPOs

Consider a 1D fermionic Hamiltonian represented as an MPO,

$$H = T(O) = \qquad (A.7)$$

Its ground state minimizes the real-valued Rayleigh quotient,

$$\frac{\langle\Psi|H|\Psi\rangle}{\langle\Psi|\Psi\rangle}, \qquad (A.8)$$

and so does its optimal MPS approximation $|\Psi(\tilde{A})\rangle$ when restricting $|\Psi\rangle$ to the MPS manifold. Put differently, the derivative of Eq. (A.8) w.r.t. to the complex conjugate of the MPS parameters equals zero, i.e.,

$$\frac{\partial}{\partial\bar{A}} (\langle\Psi(A)|) \left( H - \frac{\langle\Psi(A)|H|\Psi(A)\rangle}{\langle\Psi(A)|\Psi(A)\rangle} \right) |\Psi(A)\rangle = 0, \qquad (A.9)$$

when $A = \tilde{A}$. This expresses that the tangent space projection of the original eigenvalue equation equals zero in the optimal MPS, i.e.,

$$P_{|\Psi(\tilde{A})\rangle} (H - E) |\Psi(\tilde{A})\rangle = 0. \qquad (A.10)$$

Using the tangent space projector derived in Appendix A.1, this condition gives rise to the fixed-point equations, $A^{c\prime} \propto A^c$ and $C' \propto C$, where we dropped the tilde and

$$\text{(A.11)}$$

The proportionality factor in the fixed-point equations is equal to the (infinite) eigenvalue $E$. Invoking the MPO structure of $H$ then leads to the set of eigenvalue and linear problems Eqs. (158), (159), (160) and (161), as detailed in Sec. 3.5, that can be iteratively solved to arrive at the fixed-point solution (Eq. (162)).

Note that in our derivation of the VUMPS algorithm (Eq. (A.9)) we have made use of the fact that $H$ is Hermitian. However, all of the techniques involved, based on the determination of dominant eigenvalues (typically involving Krylov methods), also apply when searching the dominant eigenvector for a more general, non-Hermitian MPO. However, the method is no longer variational then. In this regard, one could wonder if there exists a local way to express that a fermionic MPO is Hermitian. I.e., is there a condition on $O$ under which we know that $T$ is Hermitian? Building on Sec. 2.5, we know that hermiticity of such an operator comes down to

$$T = \quad\cdots\quad = \quad\cdots\quad = T^\dagger. \quad \text{(A.12)}$$

Hence, one could infer a sufficient condition by relating the local tensors on both sides of the equality. However, as the virtual arrows point in opposite directions, we need to flip them. Furthermore, both MPOs might be related via a gauge transform. Hence, if there exists some arbitrary flipper satisfying

$$\text{(A.13)}$$

such that the relation

$$\text{(A.14)}$$

is satisfied, then $T(O)$ will be Hermitian.

## A.3 Vumps for a double-layer transfer matrix

This type of VUMPS algorithm is typically encountered when computing PEPS expectation values as the contraction of a 2D bra and ket layer with possibly a local operator in between. Hence, we will determine the leading eigenvector of the transfer operator

$$T(O) = \quad\cdots\quad , \quad \text{(A.15)}$$

where $O$ is a double-layer tensor as in Eq. (222). For the fixed-point, we propose an MPS Ansatz,

$$|\Psi(M)\rangle = \quad\text{———}\boxed{M}\text{—}\boxed{M}\text{—}\boxed{M}\text{———} \tag{A.16}$$

The optimal $M$ tensors are then determined by projecting the eigenvalue equation for $T(O)$ onto the MPS tangent space, yielding

$$P_{|\Psi(M)\rangle}\left(T(O)-\Lambda\right)|\Psi(M)\rangle = 0. \tag{A.17}$$

To do so, we have to determine the tangent space projector $P_{|\Psi(M)\rangle}$ in and orthogonal to an MPS with two physical legs. Note therefore that all the concepts mentioned in Appendix A.1 can readily be extended to multiple physical legs, with $P$ tensors on the upward physical legs as the only exception. In particular, we can bring the MPS in the mixed gauge

$$|\Psi(M)\rangle = \quad\text{———}\boxed{M^l}\text{—}\boxed{M^c}\text{—}\boxed{M^r}\text{———} \tag{A.18}$$

where

$$\boxed{M^l}\text{—}\!\!\bigcirc\!\!C \quad = \quad \boxed{M^c} \quad = \quad \bigcirc\!\!C\text{—}\boxed{M^r} \tag{A.19}$$

with

$$\left(\begin{array}{c}\boxed{M^l}\\ \bigcirc\\ \boxed{\bar{M}^l}\end{array}\right) = \left(\quad\right) \quad \text{and} \quad \left(\begin{array}{c}\boxed{M^r}\\ \bigcirc\,\,P\\ \boxed{\bar{M}^r}\end{array}\right) = \left(P\right) \tag{A.20}$$

Here, the circular tensors in the physical contractions again indicate a parity tensor. A tangent vector for the $|\Psi(M)\rangle$ states can always be expressed as

$$|\Phi(B;M)\rangle = \sum_j \quad\text{———}\boxed{M^l}\text{—}\boxed{B}\text{—}\boxed{M^r}\text{———} \tag{A.21}$$

When orthogonal to the MPS, we can parametrize the $B$ tensor as

$$\boxed{V^l}\text{—}\!\!\bigcirc\!\!X \quad \text{or} \quad \bigcirc\!\!X\text{—}\boxed{V^r} \tag{A.22}$$

i.e., the left, respectively, right canonical form where $V^l$ spans the right null-space of $\bar{M}^l$ while $V^r$ describes the left null-space of $\bar{M}^r$ so that

$$\left(\begin{array}{c}\boxed{B}\\ \bigcirc\\ \boxed{\bar{M}^l}\end{array}\right) = 0 \quad \text{respectively} \quad \left(\begin{array}{c}\boxed{B}\\ \bigcirc\,\,P\\ \boxed{\bar{M}^r}\end{array}\right) = 0, \tag{A.23}$$

and the $X$ tensor thus essentially contains all the parametric freedom in $|\Phi(B;M)\rangle$. Furthermore, both $V^l$ and $V^r$ are normalized in the same way as their corresponding MPS tensor such that

$$\left(\begin{array}{c}\boxed{\bar{V}^l}\\ \boxed{V^l}\end{array}\right) + \left(\begin{array}{c}\boxed{\bar{M}^l}\\ \boxed{M^l}\end{array}\right) = \bigcirc \quad \text{and} \quad \left(\begin{array}{c}\boxed{\bar{V}^r}\\ \boxed{V^r}\end{array}\right) + \left(\begin{array}{c}\boxed{\bar{M}^r}\\ \boxed{M^r}\end{array}\right) = \bigcirc\,\,P \tag{A.24}$$

due to completeness. In both gauges the overlap between two tangent vectors reduces to

$$\langle \Phi(B_2; M) | \Phi(B_1; M) \rangle = |\mathbb{Z}| \quad \boxed{B_1} \ \bigcirc \ \boxed{P} \ = |\mathbb{Z}| \quad \boxed{X_1} \ \boxed{P} \ \boxed{\bar{X}_2} \ . \tag{A.25}$$

The tangent space projection in and orthogonal to $|\Psi(M)\rangle$ of a general translation-invariant state $|\chi\rangle$ is the tangent vector parametrized by $X$ for which

$$X = \min_X \big\| |\chi\rangle - |\Phi(B(X); M)\rangle \big\|^2 \ . \tag{A.26}$$

Again differentiating with respect to the complex conjugate of the entries of $X$, this minimum is characterized by

$$\boxed{X} \ \boxed{P} \ - \ \boxed{\bar{M}^l} \ \boxed{\bar{V}^l} \ \boxed{\bar{M}^r} \ = 0 \, , \tag{A.27}$$

so that

$$|\Phi(B(X); M)\rangle = P_{|\Psi(M)\rangle} |\chi\rangle = \sum_j \ \boxed{\bar{M}^l} \ \boxed{\bar{V}^l} \ \boxed{P} \ \boxed{\bar{M}^r} \ \boxed{M^l} \ \boxed{V^l} \ \boxed{M^r} \tag{A.28}$$

Using the aforementioned completeness relations, we obtain

$$P_{|\Psi(M)\rangle} = \sum_j \ \boxed{\bar{M}^l} \ \boxed{P} \ \boxed{\bar{M}^r} \ - \ \boxed{\bar{M}^l} \ \boxed{\bar{V}^l} \ \boxed{P} \ \boxed{\bar{M}^r} \tag{A.29}$$

for the "two-leg" tangent space projector as the $P$ tensors appear due to the operator application. The MPS approximation for the fixed-point of $T(O)$ thus satisfies the typical VUMPS fixed-point equations, $M^{c'} \propto M^c$ and $C' \propto C$, where now

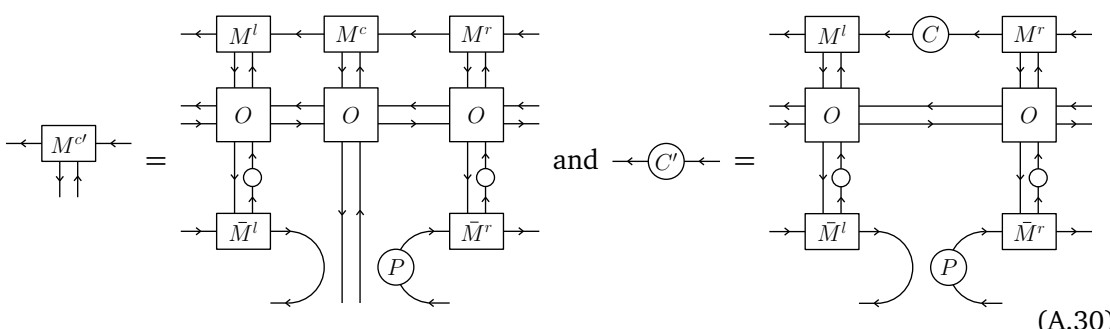

$$\tag{A.30}$$

I.e., $M^c$ and $C$ are the solutions to an eigenvalue problem which we can solve by determining the environments $G^l$ and $G^r$, defined by

$$\text{(A.31)}$$

The VUMPS fixed-point equations then reduce to

$$\text{(A.32)}$$

and

$$\text{(A.33)}$$

when the environments are normalized as

$$\text{(A.34)}$$

## B    Reformulation of GfTNS as generic fTNS

To define Gaussian fermionic tensor network states, the authors of Ref. [79] explore two approaches: the Kraus-Schuch (KS) method, which maximally exploits the Gaussian property and constructs GfTNS by applying a local Gaussian map to to a Gaussian state of maximally entangled pairs, and the Gu-Verstraete-Wen (GVW) construction, which primarily relies on the tensor network structure and utilizes Grassmann variables to contract local Gaussian tensors. Under straightforward constraints, these are equivalent and can be transformed into each other.

The GfTNS in this work (Sec. 7) are obtained by minimizing their energy density within the KS formalism. Subsequently, they are cast in their respective GVW form. As this definition resembles the $\mathbb{Z}_2$-graded formalism the most, we will use it as our starting point to transform GfTNS into generic fTNS. Therefore, consider an arbitrary GVW Gaussian fermionic tensor at site $\mathbf{n}$, defined as

$$\hat{T}_{\mathbf{n}} = \exp\left(\frac{1}{2}\chi_{\mathbf{n}}^T A \chi_{\mathbf{n}}\right) = \quad \theta_{\mathbf{n},\alpha}^h - \boxed{\hat{T}_{\mathbf{n}}} - \bar\theta_{\mathbf{n},\beta}^h \;, \tag{B.1}$$

where the column vector $\chi_{\mathbf{n}} \equiv \left(a_{\mathbf{n}}^{\dagger}, \theta_{\mathbf{n}}^{h}, \bar{\theta}_{\mathbf{n}}^{h}, \theta_{\mathbf{n}}^{v}, \bar{\theta}_{\mathbf{n}}^{v}\right)^{T}$ collects the $N$ physical creation operators $a_{\mathbf{n},i}^{\dagger}$ $(i = 1, \dots, N)$ as well as $4M$ virtual Grassmann variables $\theta_{\mathbf{n},\alpha}^{h}, \bar{\theta}_{\mathbf{n},\beta}^{h}, \theta_{\mathbf{n},\lambda}^{v}$ and $\bar{\theta}_{\mathbf{n},\sigma}^{v}$ $(\alpha, \beta, \lambda, \sigma = 1, \dots, M)$, all assigned to site $\mathbf{n}$. The latter are mutually anti-commuting and square to zero. Moreover, we make them anti-commute with the physical creation operators. The matrix $A \in \mathbb{C}^{(N+4M) \times (N+4M)}$, hence anti-symmetric, contains the variational parameters determined by the energy minimization and is assumed to be independent of $\mathbf{n}$, i.e., we are restricting ourselves to translation invariant states. With this definition of Gaussian tensors in place, we can define an (unnormalized) contracted GfTNS via the Berezin integral,

$$|\psi\rangle = \int [D\theta] \int [D\bar{\theta}] \prod_{\mathbf{n}} e^{\bar{\theta}_{\mathbf{n}}^{h T} \theta_{\mathbf{n}+\mathbf{e}_x}^{h}} e^{\bar{\theta}_{\mathbf{n}}^{v T} \theta_{\mathbf{n}+\mathbf{e}_y}^{v}} \hat{T}_{\mathbf{n}}|0\rangle, \tag{B.2}$$

where $|0\rangle$ is the physical Fock vacuum and $\mathbf{e}_{x/y}$ are unit vectors along the $x/y$-direction.

To understand how the Grassmann variables can incorporate the correct fermionic features on the virtual level, we show that there exists a straightforward isomorphism between $\mathcal{G}(M)$, the Grassmann algebra of $M$ Grassmann variables and the Clifford algebra of $M$ fermionic creation and annihilation operators, represented on a $\mathbb{Z}_2$-graded Hilbert space of dimension $2^M$. Therefore, consider the case of a single Grassmann number $\theta$. To every monomial in $\theta$ we associate a basis state of the super vector space as follows,

$$\theta^n \cong |n\rangle, \quad n \in \{0, 1\}. \tag{B.3}$$

The dual space is isomorphic to polynomials of another Grassmann number $\bar{\theta}$,

$$\bar{\theta}^n \cong \langle n|, \quad n \in \{0, 1\}. \tag{B.4}$$

The evaluation/contraction map is then given by the following Berezin integral,

$$\mathcal{C} : \langle n| \otimes |m\rangle \cong \bar{\theta}^n \theta^m \rightarrow \int d\theta \int d\bar{\theta} \, e^{\bar{\theta}\theta} \bar{\theta}^n \theta^m = \langle n|m\rangle = \delta_{nm}. \tag{B.5}$$

This mapping of monomials of Grassmann numbers to basis states of a super vector space generalizes straightforwardly to the case with more than one Grassmann number. Keeping this in mind, we can interpret the definition of the local tensor in Eq. (B.1) as a coherent state that couples the $N$ physical fermions to $4M$ virtual fermions. Moreover, we can regard it as generic fermionic tensor of the form

$$M \left\{ \begin{array}{c} \overbrace{\qquad}^{M} \quad \diagup N \\ \boxed{T_{\mathbf{n}}} \\ \underbrace{\qquad}_{M} \end{array} \right\} M. \tag{B.6}$$

Indeed, $\theta$ Grassmann variables correspond to standard $\mathbb{Z}_2$-graded Hilbert spaces, while $\bar{\theta}$ variables yield a dual space, i.e., this distinction is equivalent to placing arrows on the tensor legs. The Berezin integral performs the contraction of the local tensors by means of the $e^{\bar{\theta}_{\mathbf{n}} \theta_{\mathbf{n}+\mathbf{e}_{x/y}}}$ factors. Note that the resulting composite bond dimension of the GfTNS is indeed $D = 2^M$ in each direction.

The relation between the virtual Grassmann variables in the GVW construction and $\mathbb{Z}_2$-graded Hilbert spaces allows us to rephrase the Gaussian states as generic fTNS and thus to determine their tensor entries in Eq. (B.6) based on the parametric $A$ matrix in Eq. (B.1).

Indeed, also replacing the physical creation operators with Grassmann variables in $\chi$, we can expand the exponential in the resulting $T$ tensor as

$$T = \sum_{\{n_i\}=0}^{1} T_{n_1,n_2,\ldots,n_N} \theta_1^{n_1} \theta_2^{n_2} \cdots \theta_N^{n_N}, \tag{B.7}$$

where for simplicity of notation, we have denoted all the different Grassmann variables as $\theta_i$, where the the index $i$ enumerates the Grassmann variables in the order they appear in $\chi$. Given a bit string of $n_i's$, the corresponding tensor element can be obtained as follows: take the expansion in Eq. (B.7), take all Grassmann variables for which $n_i = 0$ and put them equal to zero by hand, and integrate over those Grassmann variables for which $n_i = 1$. We now apply the same procedure to the exponential expression in Eq. (B.1). This gives

$$T_{\{n_i\}} = \prod_{i:n_i=1} \int \mathrm{d}\theta_i \exp\left(\frac{1}{2} \chi_{\{n_i\}}^T A[n_i] \chi_{\{n_i\}}\right), \tag{B.8}$$

where the product over integrals is descending order, and $\chi_{\{n_i\}}$ is the vector of Grassmann variables for which $n_i = 1$, and $A[n_i]$ is the corresponding submatrix of $A$ where only the rows and columns corresponding to the Grassmann variables with $n_i = 1$ are kept. The result of the integral is

$$T_{\{n_i\}} = \mathrm{Pf}(A[n_i]). \tag{B.9}$$

The generic tensor entries can thus be obtained by calculating Pfaffians of submatrices of the parametric matrix $A$.

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
