# Peer review of "Fermionic tensor network methods"

_SciPost Physics, doi:SciPost Phys. 18, 012 (2025)_

## Round 1 · Referee Report · Anonymous (Referee 1) · 2024-7-19

Strengths

1) a systematic way of dealing with fermion signs is developed, where most fermion signs are treated automatically, independent of specific algorithms.

2) The results on excitations in the 1D Hubbard model are impressive. The plane-wave ansatz seems to capture deconfined excitations and multiparticle continuum in 1D.

Weaknesses

1) The rule of inserting additional parity tensor should be clarified. It is currently shown in examples, and it is not clear how to implement the method for a new algorithm.

2) The lack of 2D numerical results in variationally finding ground states.

3) The discussion on fermionic matrix product operators should be more explicit.

Report

This submission provides a new way to deal with fermion signs in tensor network algorithms, an important subject first studied 15 years ago. Simulating strongly interacting fermionic systems is one of the central topics in condensed matter physics, and PEPS has recently become a reliable method. A simple way to deal with the non-locality of fermionic statistics in various tensor network algorithms is desirable.

Many methods have been developed before, for example, explicitly attaching fermion creation/annihilation operators to the tensors or adding an extra swap gate whenever fermionic legs cross. The method introduced here is based on these ideas but with significant improvements. Most fermion signs are automatically dealt with at the level of elementary tensor operations, independent of the algorithm. Additional parity tensors are introduced when enforcing isometric form or hermiticity. The 1D Hubbard model, Kitaev chain, and correlation functions of p+ip superconductors are analyzed to demonstrate the method. The numerical results confirm the correctness of the formalism, but it would be good to have more results on 2D fermionic systems, especially in solving ground states. For future applications, it is essential to know whether fermionic signs affect the stability and efficiency of variational algorithms.

The draft is generally well-written but should be revised to clarify a few subtle issues. I recommend publication if the authors address the issues below.

Requested changes

1) On page 3, the discussion on the swap gate should be improved. In most applications, like double-layer contraction and sampling, swap gates only occur between neighboring tensors and can be incorporated locally, although one needs to draw figures and count crossings. The difference between the graded Hilbert space used in this draft and the fermion creation/annihilation operator used in previous methods is slightly over-emphasized.

2) On page 3, when discussing previous works on excitations in 2D, authors could mention real-time evolution in https://arxiv.org/abs/2211.00043.

3) On page 6, I find the example given in Eq. 8-10 confusing. No matter what order we place the tensor, we can always add a twist to any leg and get an extra minus sign. For readers not familiar with the NCON syntax, it would be even more confusing. I am not sure whether it is helpful.

4) On page 9, after the sentence 'We will use the notation F very loosely and will refer to objects related by this isomorphism as being the same.' it's probably better to add an equation |i>1 |j>2 = (-1)^|i||j| |j>2 |i>1 to clarify the statement, and then write C(<i||j>) = delta_ij, C(|j><i|) = (-1)^|i| delta_ij and avoiding using \tilde{C}. Currently, \tilde{C} is introduced but never used. I initially confused it with the additional parity tensor.

5) On page 12, after Eq. 41, 'Note that the order in which the legs are drawn on the diagram does not matter.'. I find this sentence confusing. Is it about the counterclockwise order? There is no way to know which leg is the first leg from the drawing. Maybe the authors should state clearly that fermion signs are treated according to algebraic rules that depend on the order of indices and their direction in an automatic way and are not related to how they are drawn.

5) The insertion of additional parity tensor P and the fuser is the most tricky aspect of this method and should be better explained. First, if one does not truncate, there is no need to insert P for the double-layer contraction of finite MPS or PEPS. Is that true? I understand that the insertion of P is always related to the definition of inner product. For example, in Eq. 109, the orthogonality condition is defined such that <psi|psi> = inner product of Ac and itself, which has a P on the right. In Eq. 215, the insertion of P is to make <psi|psi> = <N|M>. And P is inserted in the definition of linear maps such that the unitarity or hermiticity of the map respects the inner product. The logic of inserting P in each example should be clearly stated. Most importantly, is there a simple, generic rule for adapting a bosonic algorithm to fermions? Is detailed knowledge of an algorithm required to know where to insert extra parity tensors?

6) I found the discussion after Eq 65 hard to understand when I first read it. Perhaps the authors could change |beta> into <beta| in the example of tensor contraction in Eq 27 (and change the delta accordingly) such that there is an extra sign (-1)^|beta|. Otherwise, it is pretty mysterious what cancels the P tensors inserted.

7) On page 21, I think it is better to introduce the fermion creation and annihilation operator without the complexity of SU(2) symmetry and add symmetries afterward. In general, fermionic MPO should be more explicitly introduced. Currently, the process of contracting the auxiliary legs of a and a^deg and svd the result into L and R to make the mpo is briefly sketched, but hard to follow. I think it's good to add a section discussing mpo, perhaps between 3.2 and 3.3 . To explicitly write down the mpo tensor with physical and virtual legs, not the auxiliary leg, for a simple model (e.g. Kitaev chain) without the complication of non-abelian symmetry.

8) On page 53, does imposing Eq 234 and 235 reduce the representability of PEPS? It's not clear what 'integrating gauge transformation' means

9) On page 53, it is claimed one could combine automatic differentiation and the fermionic methods introduced here to find ground states variationally. Why not demonstrate that numerically? It would be interesting to see whether the general algorithm finds a better ground state than GfTNS for free-fermion models. GfTNS restricts representability, but it is easy to optimize.

10) On page 58, figure 6, it is not clear where the second exponential regime is. I cannot see any qualitative difference between D = 4 and D = 8

11) Missing A_{alpha beta gamma} in Eq 89

12) The right half of Eq 251 should be Oa instead of Oa^deg

Recommendation

Ask for minor revision

---

## Round 1 · Referee Report · Anonymous (Referee 2) · 2024-8-12

Strengths

1- Clearly explains the challenges associated with dealing with fermions, and proposes a useful framework for handling them with Z2 graded Hilbert spaces.

2- Provides good overview of many important tensor network algorithms.

3- Provides many numerical results demonstrating the power of fermionic tensor network algorithms.

Weaknesses

1- I think the paper can benefit from a table summarizing the contraction rules.

Report

This paper presents a method to perform fermionic tensor network computations using Z2 graded Hilbert spaces, and associated tensor contraction rules.
The method can be readily applied to tensor networks in arbitrary dimensions and can be numerically implemented to perform most standard tensor network algorithms.
The paper is clearly written and addresses an important technical challenge.

There are a few minor points I would like to see addressed before recommending the publication of this manuscript, which I detail below.

Requested changes

  1. There are many rules being introduced for contraction of fermionic tensor networks. I think I managed to absorb all of them, since I finished reading the manuscript in a single sitting. However, I expect it might be difficult to use it as a reference for contraction rules in the future. Would it be possible to have a table summarizing all the contraction rules, conjugation rules etc. maybe with appropriate reference to equations?

  2. Most of the paper is self-contained, but some parts of the discussion early on was a little confusing. In particular, as the other referee also pointed out, I found the discussion about "self-crossing" around Eq. (10) a bit confusing. This is probably because I am not familiar with how to handle tensor networks using swap gate based method, but I expect there are many other readers like me. I think a bit more explanation on this point might be helpful.

  3. I found the discussion around Eq.(60) a bit confusing. This is probably because I found it difficult to keep track of "mirror" rule for $\bar{T}$. I think an easy work around is to make $T$ appear mirror-asymmetric, such that the number of legs from left and right appear different.

  4. Around Eq. (186), the manuscript mentions that closing a loop introduces an additional factor of $z$ for $i=1, z$. Can you be more explicit about how this comes about?

  5. In the discussion about DMRG for Hubbard model, the manuscript proposes using shifted U(1) charge so that the charge zero sector corresponds to the sector of interest. I was wondering how it should be modified to the case with electron doping. Should the charge be shifted non-uniformly to account for the correct charge sector?

More minor comments below: 1. Equation (27) has two types of C's, and I found this confusing. This can be avoided by using D in place of C for the tensor.

  1. Typo above Eq. (48)

  2. Above equation (61), the manuscript says $C(\bar{T}P(TP(v)))$. Shouldn't it be $C(\bar{T}P(C(TP(v))))$ instead?

  3. When reporting agreement of ground state energy between exact solution and numerical computation, I would like to know the number of digits of agreement.

  4. Typo (Fig. Fig. 1) above Eq. (169)

Recommendation

Ask for minor revision

---

## Round 2 · Referee Report · Anonymous (Referee 1) · 2024-12-1

Report

The authors have addressed all of my questions and comments. The resubmission clearly stated the difference between this method and existing methods and better summarized the set of rules implemented for fermionic tensors. I recommend accepting this manuscript.

Recommendation

Publish (meets expectations and criteria for this Journal)

---

## Round 2 · Referee Report · Anonymous (Referee 2) · 2024-12-2

Report

The authors adequately addressed my concerns, and the distinction with the swap gate method has been made clearer. I fully recommend the publication of this manuscript.

Recommendation

Publish (easily meets expectations and criteria for this Journal; among top 50%)

---

## Round 2 · Author Response

We thank the referees for taking the time to evaluate our manuscript.

In response to the feedback, we have tried to incorporate all requested details and clarifications regarding the conceptual framework. Additionally, we have added a summary at the end of Section 2 for readers that want to use our formalism, e.g. for numerical implementation. Furthermore, we performed an additional optimization of fermionic PEPS using CTMRG contraction in combination with automatic differentiation (AD), demonstrating the capabilities of our framework.

---

## Round 2 · List of Changes

Report 1

1) On page 3, the discussion on the swap gate should be improved. In most applications, like double-layer contraction and sampling, swap gates only occur between neighboring tensors and can be incorporated locally, although one needs to draw figures and count crossings. The difference between the graded Hilbert space used in this draft and the fermion creation/annihilation operator used in previous methods is slightly over-emphasized.

We improved the discussion of the swap-gate approach, aimed at a clearer and more balanced comparison with our formalism (also continued in Section 2.1).

2) On page 3, when discussing previous works on excitations in 2D, authors could mention real-time evolution in https://arxiv.org/abs/2211.00043.

We now cite this reference.

3) On page 6, I find the example given in Eq. 8-10 confusing. No matter what order we place the tensor, we can always add a twist to any leg and get an extra minus sign. For readers not familiar with the NCON syntax, it would be even more confusing. I am not sure whether it is helpful.

We would like to thank the refer for pointing this out. Section 2.1 was rewritten to give a more concrete comparison between our formalism and the swap-gate approach, based on the example that was already present. To highlight the specifics of our method, we also return to this example both in Sec. 2.3 and 2.4. We kept the NCON syntax but put it in a footnote. What the commands mean in terms of contracted legs was written out in full so that readers unfamiliar with the NCON syntax can equally well follow the discussion. In response to the comment about the possibility to always add a twist: this is not possible by continuously moving the tensors around, i.e., while we allow for tensors to be moved around freely, moving only legs and creating new twists in the process is not what we meant here.

4) On page 9, after the sentence 'We will use the notation F very loosely and will refer to objects related by this isomorphism as being the same.' it's probably better to add an equation |i>1 |j>2 = (-1)^|i||j| |j>2 |i>1 to clarify the statement, and then write C(<i||j>) = delta_ij, C(|j><i|) = (-1)^|i| delta_ij and avoiding using \tilde{C}. Currently, \tilde{C} is introduced but never used. I initially confused it with the additional parity tensor.

\tilde{C} was not used anymore and we added the requested clarification.

5) On page 12, after Eq. 41, 'Note that the order in which the legs are drawn on the diagram does not matter.'. I find this sentence confusing. Is it about the counterclockwise order? There is no way to know which leg is the first leg from the drawing. Maybe the authors should state clearly that fermion signs are treated according to algebraic rules that depend on the order of indices and their direction in an automatic way and are not related to how they are drawn.

We rephrased this to avoid confusion.

5) The insertion of additional parity tensor P and the fuser is the most tricky aspect of this method and should be better explained. First, if one does not truncate, there is no need to insert P for the double-layer contraction of finite MPS or PEPS. Is that true? I understand that the insertion of P is always related to the definition of inner product. For example, in Eq. 109, the orthogonality condition is defined such that <psi|psi> = inner product of Ac and itself, which has a P on the right. In Eq. 215, the insertion of P is to make <psi|psi> = <N|M>. And P is inserted in the definition of linear maps such that the unitarity or hermiticity of the map respects the inner product. The logic of inserting P in each example should be clearly stated. Most importantly, is there a simple, generic rule for adapting a bosonic algorithm to fermions? Is detailed knowledge of an algorithm required to know where to insert extra parity tensors?

Since it was also suggested by the second referee, a summary containing the most important operations from Sec. 2 was added at its end. Together with a reminder of the conventions we abide by, and the important note that for fermions contraction and inner products are not the same in our formalism (hence the P's), the main elemets of our formalism are highlighted here.

6) I found the discussion after Eq 65 hard to understand when I first read it. Perhaps the authors could change |beta> into <beta| in the example of tensor contraction in Eq 27 (and change the delta accordingly) such that there is an extra sign (-1)^|beta|. Otherwise, it is pretty mysterious what cancels the P tensors inserted.

We changed the manuscript according to the referee's suggestion.

7) On page 21, I think it is better to introduce the fermion creation and annihilation operator without the complexity of SU(2) symmetry and add symmetries afterward. In general, fermionic MPO should be more explicitly introduced. Currently, the process of contracting the auxiliary legs of a and a^deg and svd the result into L and R to make the mpo is briefly sketched, but hard to follow. I think it's good to add a section discussing mpo, perhaps between 3.2 and 3.3. To explicitly write down the mpo tensor with physical and virtual legs, not the auxiliary leg, for a simple model (e.g. Kitaev chain) without the complication of non-abelian symmetry.

We agree that it is more instructive to start with spinless operators and therefore discuss this case first. In particular, we emphasize the need for the auxiliary leg carrying the odd fermionic charge. We then turn to the more involved SU(2)-symmetric case and have also improved the explanation here. For a full discussion on how to cast the resulting Hamiltonian contributions in an MPO, we refer the reader to more dedicated papers, as it is not the focus of this work. For the specific models used in Sec. 4, we explictly write out the MPO's and again cite the relevant references.

8) On page 53, does imposing Eq 234 and 235 reduce the representability of PEPS? It's not clear what 'integrating gauge transformation' means

We agree with the referee that this does reduce the PEPS representability and hence explictly refer to Eq. (248) as a constraint. Furthermore, we removed the incorrect remark about adding gauge transformations. The constraint is a sufficient condition for a Hermitian transfer operator.

9) On page 53, it is claimed one could combine automatic differentiation and the fermionic methods introduced here to find ground states variationally. Why not demonstrate that numerically? It would be interesting to see whether the general algorithm finds a better ground state than GfTNS for free-fermion models. GfTNS restricts representability, but it is easy to optimize.

For the smaller bond dimensions (D = 2, 4), randomly initialized fermionic PEPS were optimized for this model via an AD-assisted, CTMRG LBFGS procedure. We confirm what the referee was expecting: generic PEPS energies can do better than GfTNS at the cost of a more difficult optimization algorithm. Results were added at the end of Sec. 7.1, a discussion of the AD procedure in Sec. 6.4.

10) On page 58, figure 6, it is not clear where the second exponential regime is. I cannot see any qualitative difference between D = 4 and D = 8

One should really focus on the circle markers for the Gaussian result here. In the double log plot for Ns=50^2 and for D=4, these first have the exact exponential decay, then a polynomial decay (straight line) but after ~200 sites these again level off to an exponential decay. At the same system size, we do not see this for D=8. At higher system sizes however, both D=4 and D=8 have a polynomial tail for all the separations in the plot. We conclude that the second expontial decay is a feature that has to be there as the state is not fully chiral (no-go theorem of Read and Dubail) but it is surpressed when the state is optimized at a higher Ns and/or when using a higher bond dimension.

11) Missing A_{alpha beta gamma} in Eq 89

Indeed, we added this.

12) The right half of Eq 251 should be Oa instead of Oa^deg

We added a remark pointing out that the difference is the auxiliary arrow.

Report 2

1) There are many rules being introduced for contraction of fermionic tensor networks. I think I managed to absorb all of them, since I finished reading the manuscript in a single sitting. However, I expect it might be difficult to use it as a reference for contraction rules in the future. Would it be possible to have a table summarizing all the contraction rules, conjugation rules etc. maybe with appropriate reference to equations?

We thank the referee for this suggestion and added a summary of the relevant operations at the end of Section 2.

2) Most of the paper is self-contained, but some parts of the discussion early on was a little confusing. In particular, as the other referee also pointed out, I found the discussion about "self-crossing" around Eq. (10) a bit confusing. This is probably because I am not familiar with how to handle tensor networks using swap gate based method, but I expect there are many other readers like me. I think a bit more explanation on this point might be helpful.

As mentioned before, this Section was thoroughly rewritten to explain the swap gates in more detail and to constrast them with our approach based on the example that was already present.

3) I found the discussion around Eq.(60) a bit confusing. This is probably because I found it difficult to keep track of "mirror" rule for T. I think an easy work around is to make T appear mirror-asymmetric, such that the number of legs from left and right appear different.

We implemented the suggestion of the reader, though in a slightly different way by just changing the shape of the T (and thus \bar{T}) tensor.

4) Around Eq. (186), the manuscript mentions that closing a loop introduces an additional factor of z for i=1,z. Can you be more explicit about how this comes about?

We explained this in more detail in a footnote and refer to the seminal paper of Bultinck et al. where this was also encountered.

5) In the discussion about DMRG for Hubbard model, the manuscript proposes using shifted U(1) charge so that the charge zero sector corresponds to the sector of interest. I was wondering how it should be modified to the case with electron doping. Should the charge be shifted non-uniformly to account for the correct charge sector?

Indeed, a certain filling can be targeted by choosing a commensurate unit cell.

6) Equation (27) has two types of C's, and I found this confusing. This can be avoided by using D in place of C for the tensor.

We decided to keep this as was to not introduce too many different tensors.

7) Typo above Eq. (48)

Indeed, fixed.

8) Above equation (61), the manuscript says C(TP(TP(v))). Shouldn't it be C(TP(C(TP(v)))) instead?

Indeed, fixed.

9) When reporting agreement of ground state energy between exact solution and numerical computation, I would like to know the number of digits of agreement.

We added the number of correct digits for the accuracy (e.g. in Sec. 4.1).

10) Typo (Fig. Fig. 1) above Eq. (169)

Fixed.

---

## Editorial Decision

published